# *Gtf2i*-encoded transcription factor Tfii-i regulates myelination via *Sox10* and *Mbp* regulatory elements

Gilad Levy [1], May Rokach[1], Inbar Fischer[1], Omri Kimchi-Feldhorn[1], Shiri Shoob[2], Ela Bar[3,4], Tali Rosenberg[5], Joanna Bartman[5], Hadar Parnas[5], Meitar Grad[1], Ifat Israel-Elgali[1,2], Galit E. Sfadia[3], Sari S. Trangle[3], Anna Vainshtein[6], Yael Eshed Eisenbach[6], Olaf Jahn [7,8], Sophie B. Siems[9], Hauke B. Werner [9,10], Noam Shomron[1,2], Yaniv Assaf [1,4], Elior Peles [6], Inna Slutsky [1,2,11], Asaf Marco [5] & Boaz Barak [1,3,4] ✉

The transcriptional regulatory network governing the differentiation and functionality of oligodendrocytes (OLs) is essential for the formation and maintenance of the myelin sheath, and hence for the proper function of the nervous system. Perturbations in the intricate interplay of transcriptional effectors within this network can lead to a variety of nervous system pathologies. In this study, we identify *Gtf2i*-encoded general transcription factor II-I (Tfii-i) as a regulator of key myelination-related genes. *Gtf2i* deletion from myelinating glial cells in male mice leads to functional alterations in central nervous system (CNS) myelin, including elevated mRNA and protein expression levels of myelin basic protein (Mbp), the central myelin component, enhanced connectivity properties, and thicker myelin wrapping axons with increased diameters. These changes resulted in faster axonal conduction across the corpus callosum (CC), and improved motor coordination. Furthermore, we show that in mature OLs (mOLs), Tfii-i directly binds to regulatory elements of *Sox10* and *Mbp*. In the peripheral nervous system (PNS), *Gtf2i* deletion from Schwann cells (SCs) leads to hypermyelination of the tibial branch of the sciatic nerve (SN). These findings add to our understanding of myelination regulation and specifically elucidate a cell-autonomous mechanism for Tfii-i in myelinating glia transcriptional network.

The myelin sheath is an insulating multi-lamellar membrane surrounding axons that is produced by OLs[1] in the CNS and by SCs in the PNS[2]. Crucial for proper axonal integrity and conductivity, myelination of the CNS mainly occurs postnatally[3,4], and continues throughout life[3–7]. Neurodevelopmental[8,9], neuropsychiatric[10,11], and neurodegenerative disorders[12,13] may lead to a lack of proper myelination during development or to de-myelination. This can result in impaired signal conduction, affecting multiple brain functions such as motor coordination and

behavior modulation[2,14,15]. It can even lead to shorter life expectancy compared to unaffected individuals[13,16,17]. The limited number of current therapies for white matter (WM) pathologies focus on increasing the number of myelin-producing cells[18–21], rather than enhancing myelin production per cell. Since WM is composed primarily of myelinated axons, boosting myelin synthesis directly could improve WM integrity and function, potentially leading to better outcomes in conditions affecting myelination. Therefore, there is considerable need for basic

and clinical research aimed at identifying factors that regulate myelination and then determining whether modulating the actions of such factors can increase myelin production[22].

During late embryonic and early postnatal stages, OL progenitor cells (OPCs) appear in the ventricular zone of the neural tube[23,24]. These then proliferate and migrate brain-wide, accounting for 5–8% of the adult CNS cell population[25]. OPC maturation into OLs begins with the establishment of axon-OPC contacts. Once initiated by committed OPCs, such maturation triggers process outgrowth and OPC polarization towards the axon, supported by various OPC intracellular modulators[26–29]. Following these initial steps, additional layers of new membrane generated by the newly matured OL are added at the interaction site, wrapping the axon. Compaction of these membranes is critical for proper insulation and is mainly attributed to the MBP, which acts as an adhesive molecule[30]. Compact myelin greatly reduces the radial leakage of ions flowing along the axon. The presence of non-myelinated regions along the axon termed 'Nodes of Ranvier' (NoRs) that contain voltage-gated sodium channels, enable saltatory signal propagation, thus supporting rapid signal conduction along the length of the axon[31].

During development, various extracellular signals, such as growth factors and morphogens, activate specific signaling pathways, leading to the expression of lineage-specific transcription factors (TFs) that orchestrate OL differentiation and myelination[32–36]. The major TFs that contribute to regulating myelination include OLIG1/2[37], MRF[38], YY1[39], ZFP24[40], NKX2.2[41], and SOX10[34,42,43]. Mutation[44,45] or deletion[46–48] of the genes encoding these TFs may result in adverse phenotypes, including damage to the myelin sheath[35]. These intracellular factors regulate both cell-intrinsic[40,49–53] and axon-glial signaling pathways[54–61]. For example, Sox10 is a cell-intrinsic activator of myelination in myelinating OLs, directly and positively regulating the expression of key myelination components, such as *Plp1*[62] and *Mbp*[34], with the expression level of Mbp having been shown to be a rate-limiting step in CNS myelination[63].

While studies have found that TFs can increase the expression of myelination-related genes[34,62], TFs can also inhibit the transcription of myelination-related genes[64,65], or do both actions[66]. One such dual-action TF is GTFII-I, a highly conserved and ubiquitously expressed multi-functional TF[67,68]. Encoded by *GTF2I* in humans, GTFII-I is a widely expressed TF, shown to interact with promoter initiator elements and DNA binding motifs such as the E-box element[69]. GTFII-I is activated in response to a variety of extracellular signals that can trigger its translocation into the nucleus. In the nucleus, GTFII-I regulates gene expression through interactions with tissue-specific TFs and complexes related to chromatin remodeling[70]. GTFII-I interacts with various components of transcriptional complexes to mediate transcription[70], such as Suz12 of the polycomb repressive complex 2[71], but not with the transcriptional core complex itself[72]. GTFII-I is also active outside of the nucleus and acts as an inhibitor of agonist-induced calcium entry in the cytoplasm[73]. Additionally, GTFII-I is essential in embryonic development[68,74], critical for cell-type specific immune functions in B[75,76] and T cells[77], and is involved in the regulation of the endoplasmic reticulum stress response pathway[78].

*GTF2I* is part of the 7q11.23 human chromosomal region, which is prone to de novo alternations in a gene dosage-dependent manner[79]. Micro-duplication of this region results in speech delay, autism spectrum disorder (ASD)-related behaviors[80–86], and schizophrenia[87], while a micro-deletion in this region is the cause of Williams syndrome (WS)[88,89], a condition associated with a unique cognitive[90–93] and hyper-social phenotype[94]. Copy number variation studies of this chromosomal region specifically associate *GTF2I* levels with the cognitive and behavioral manifestation[95–101]. For example, *GTF2I* duplications were shown to result in increased anxiety[96], and ASD-like phenotype[102]. Furthermore, atypical deletions of *GTF2I* results in social disinhibition[95,98,100,103–105] and intellectual disability[95,97,101,104].

Additionally, single nucleotide polymorphisms (SNPs) of *GTF2I* were associated with increased anxiety[106,107] and ASD[102]. Finally, SNPs of *GTF2I* are also associated with autoimmune disorders[108,109] and thymic epithelial cancer[110,111].

We recently characterized a mouse model with a conditional homozygous deletion of *Gtf2i*, the murine homolog of *GTF2i*, in forebrain excitatory neurons[20]. These mice unexpectedly demonstrated multi-layered myelination defects underlying WS-associated increased sociability, motor deficit behaviors, and axonal conductivity deficits. While these findings indicate that neuronal *Gtf2i* expression affects OL properties[112], the role of *Gtf2i* in OLs is unknown and was, therefore, the focus of the current study.

Previous comprehensive genome-wide analysis conducted on brain tissues of wild-type mice embryos demonstrated that Tfii-i (the murine protein product of *Gtf2i*) has the capacity to bind to regulatory elements (REs) associated with essential myelination genes, such as *Sox10* and *Mbp*[113]. These genes are known to be expressed in OPCs, newly-formed OLs, mOLs, and SCs. While this suggests that Tfii-i plays a regulatory role in mOLs, the precise effect of Tfii-i binding and its implications for myelination properties remains unclear. Therefore, to study Tfii-i-mediated transcriptional regulation specifically in myelin-producing cells (i.e., OLs and SCs), we homozygously deleted *Gtf2i* in myelinating glia using the Cre-loxP system, by crossing *Cnp-Cre* driver mice[114] and *Gtf2i* loxP mice[115]. The resulting strain (termed *Gtf2i* knockout [*Gtf2i*-KO] mice) exhibited altered myelin properties, including elevated mRNA and protein levels of the key myelin component Mbp, as compared to control littermates (*Gtf2i^f/f^;Cnp-Cre^-/-^*, hereby termed control mice) (Supplementary Fig. 1A). Furthermore, *Gtf2i*-KO mice showed enhanced connectivity properties, thicker myelin sheath wrapping axons with increased diameters in the CNS, faster axonal conduction along the CC, and hypermyelination of the tibial branch of the SN. Behavioral assessment of *Gtf2i*-KO mice revealed enhanced motor coordination, increased anxiety-like behavior, and heightened sociability compared to controls.

In seeking a mechanistic explanation for the enhanced myelination properties following *Gtf2i* deletion in myelinating glia, we demonstrated that, in cultured mOLs, Tfii-i binds directly to REs of *Mbp* and *Sox10*, the levels of which were also elevated in *Gtf2i*-KO mice. Furthermore, we showed that induced deletion of *Gtf2i* in an mOL-enriched primary cell culture increased Mbp and Sox10 expression levels, leading to cell surface outgrowth. Together, these data suggest that Tfii-i serves an important cell-autonomous role in mOLs, acting as a regulator of myelination, with its reduction resulting in functional alterations of CNS myelin and hypermyelination in the PNS.

## Results

### *Gtf2i* deletion in myelinating glia does not affect gross anatomical properties but results in brain-wide alterations in tract number

To validate the genetic deletion of *Gtf2i* selectively from mOLs and the reduction of the protein product, Tfii-i, in mOLs (Fig. 1A), we performed immunohistochemical staining, showing decrease in Tfii-i expression levels in CC1+ cells in *Gtf2i*-KO mice, as compared to controls (Fig. 1B). Tfii-i was previously shown to be expressed and have different roles in the nucleus and in the cytoplasm[73,116]. However, its expression level and roles were never directly assessed in the myelin fraction. Thus, we characterized its expression in the myelin fraction purified from whole cortex to study whether Tfii-i is specifically localized to myelin. While we found no evidence for Tfii-i expression in the purified myelin fraction (Supplementary Fig. 1B), Tfii-i levels in the crude myelin fraction, which may be partly contaminated with cellular and nuclear debris from both OLs and axons[117], were significantly decreased in *Gtf2i*-KO mice, as compared to controls (Supplementary Fig. 1B-C). As such, we showed that Tfii-i is expressed in mOLs and validated its deletion from mOLs in *Gtf2i*-KO mice.

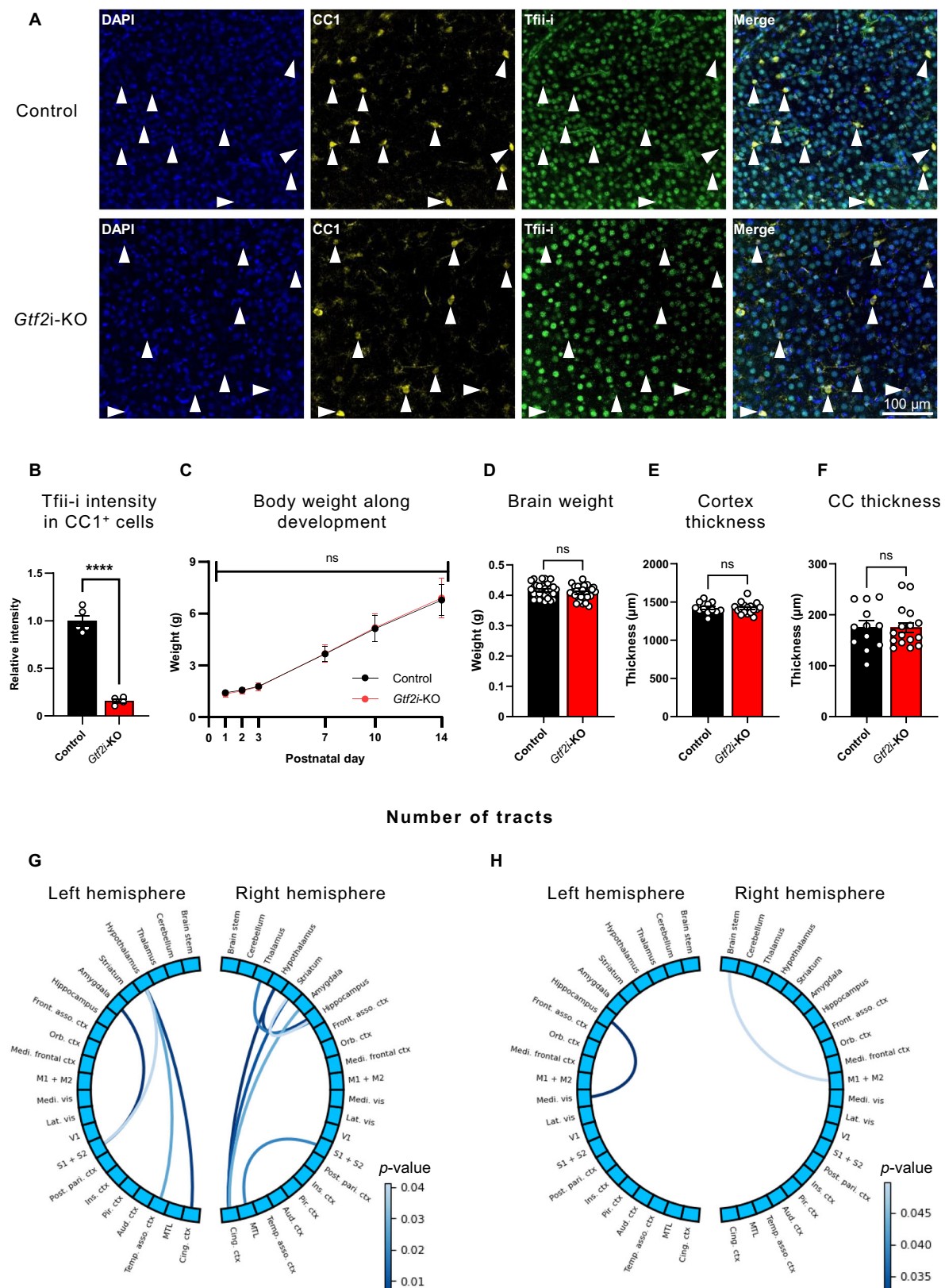

**Number of tracts**

As Tfii-i was found to be essential for neurotypical development[9,20,68,95,118], we assessed gross anatomical properties and found no significant differences between *Gtf2i*-KO mice and controls in terms of body weight during early post-natal stage (Fig. 1C), and brain weight (mean control 0.4157 ± 0.005 g, mean *Gtf2i*-KO 0.4085 ± 0.004 g) (Fig. 1D), cortical thickness (mean control 1420 ± 21.24 μm, mean *Gtf2i*-

KO 1418 ± 17.8 μm) (Fig. 1E), or CC thickness (mean control 175.7 ± 12.4 μm, mean *Gtf2i*-KO 175 ± 9.732 μm) (Fig. 1F) at the age of one-month (postnatal day 30, hereby termed P30).

To define whether *Gtf2i* deletion in mOLs affected brain connectivity properties, we utilized a magnetic resonance imaging and diffusion tensor imaging (MRI-DTI) protocol, able to detect alterations

**Fig. 1 | *Gtf2i* deletion from myelinating glia does not alter gross developmental and neuroanatomical properties but results in brain-wide connectivity pattern alterations. A** Representative images of immunofluorescence assay showing intact Tfii-i expression in mOLs (CC1+ cells) of the cortex of control mice (upper row), and a substantial reduction of Tfii-i expression in mOLs of the cortex of *Gtf2i*-KO mice (lower row). **B** Tfii-i is significantly decreased in CC1+ cells in the cortex of *Gtf2i*-KO mice, compared to controls (n = 5 control, n = 4 *Gtf2i*-KO, 13–20 cells analyzed per mouse, P = 4.24*10⁻⁶). Control normalized to 1. **C–F** Unchanged developmental and neuroanatomical properties in *Gtf2i*-KO mice, compared to controls; **C** body weight along early postnatal development (control n = 10, *Gtf2i*-KO n = 12, P = 0.91, two-way ANOVA, Sidak's multiple comparisons: P > 0.99 for P1, P2, P3, P7, P10, and P = 0.998 for P14), **D** brain weight at P30 (n = 29 control, n=29 *Gtf2i*-KO, P = 0.24), **E** cortex (n = 12 control, n = 17 *Gtf2i*-KO, P = 0.93) and **F** CC thickness (n = 12 control, n = 16 *Gtf2i*-KO, P = 0.96). **G, H** Connectograms mapping increase in the number of tracts in *Gtf2i*-KO mice, compared to controls (n = 16 controls, n = 14 *Gtf2i*-KO). Each arc represents projectiles between two brain regions in which there is a significant increase in the number of tracts in (**G**) *Gtf2i*-KO mice compared to littermate controls and in (**H**) littermate controls compared to *Gtf2i*-KO mice. **B, D-F** Two-sided t-test. **G, H** The scale bar indicates the level of significance (two-sided t-test). Light lines represent significance levels of P < 0.05 and close to this threshold, while dark lines indicate higher levels of significance. Notice the different scale bar values for (**G**) and (**H**). **B–F** Data are presented as mean values ± SEM. **** P < 0.0001, ns – non-significant. Source data are provided as a Source Data file.

in fiber systems and tissue micro-structures[119]. While DTI underlines connectivity patterns rather than directly measuring myelination or WM properties, there is a correlation between the number of reconstructed streamlines (tracts) and myelin-related properties[120,121]. P30 *Gtf2i*-KO mice demonstrated a significantly increased number of tracts (Fig. 1G) in multiple brain regions, as compared to controls, suggesting organizational and micro-structural alterations in the WM due to the gene deletion. Increased numbers of tracts in control mice, relative to *Gtf2i*-KO mice, was only observed in two WM fiber systems, with less significant alterations compared to those observed in *Gtf2i*-KO mice (Fig. 1H, see the different p-value legend compared to the one presented in Fig. 1G).

## Axonal, myelin, and conductivity properties are altered in the CC of *Gtf2i*-KO mice

The brain-wide connectivity changes observed in *Gtf2i*-KO mice prompted us to further study WM properties. Because the *Cnp* promoter is highly active in the CC[114], which also mediates interhemispheric communication and corresponds to the largest WM tract in the brain, we focused on the CC midline of P30 mice to study myelination-related ultrastructural properties by transmission electron microscopy (TEM) (Fig. 2A, Supplementary Fig. 3).

One extrinsic cue that affects myelination is the diameter of the axon[122–126], whereby the larger the axon diameter, the thicker is the surrounding myelin sheath. The interplay between axon diameter and myelin sheath diameter is termed the *g*-ratio[127], the value of which reflects axonal signal conduction properties[128]. *g*-ratio values are commonly used as an index of myelin sheath thickness[129], with higher values possibly underlining myelination deficits[130]. While the optimal theoretical *g*-ratio in the PNS is approximately 0.65[127], the optimal theoretical *g*-ratio in the CNS of rats is approximately 0.77[131].

While *g*-ratio values in the overall axonal population of the CC were not significantly different, the interplay between axonal diameter and *g*-ratio values differed significantly between *Gtf2i*-KO mice and controls (Fig. 2B). Specifically, the slope representing this interplay is more gradual in *Gtf2i*-KO mice axons, indicating that the *g*-ratio values of *Gtf2i*-KO mice axons are less affected by axonal diameter, compared to controls. In addition, the diameters of *Gtf2i*-KO mice axons were significantly increased, as compared to controls (Fig. 2C, dashed vertical line indicates the median axon diameter of the control group, 0.53 μm). Axons with larger diameters, as seen in the CC of *Gtf2i*-KO mice, are typically wrapped with a thicker myelin sheath[132]. Accordingly, *Gtf2i*-KO mice axons demonstrated thicker myelin sheath, compared to controls (Fig. 2D). The ratio of myelinated to unmyelinated axons was similar between *Gtf2i*-KO mice and controls (Fig. 2E, mean control 0.266 ± 0.02, mean *Gtf2i*-KO 0.26 ± 0.029).

To better capture the heterogeneity of axonal and WM properties across the CC and investigate structural and myelination abnormalities in *Gtf2i*-KO mice, we performed MRI-DTI and quantified WM tracts across the entire CC using tractography analysis (Fig. 2F). Our results show a significantly lower number of WM tracts in the CC of *Gtf2i*-KO mice, as compared to controls (Fig. 2G, mean control 1159.75 ± 83.637,

mean *Gtf2i*-KO 942.214 ± 50.016). Taken together with our finding of unchanged CC thickness in *Gtf2i*-KO mice (Fig. 1F), these results are in accordance with our finding of increased axonal diameter in the CC of *Gtf2i*-KO mice (Fig. 2C). Given that WM tract integrity and axonal properties are closely linked to myelination and conductivity, we further examined NoRs in the midline CC (Fig. 2H, Supplementary Fig. 3). Alterations in NoR number and length have been associated with myelination dynamics and conductivity efficiency[133,134]. While NoR length in the midline CC was similar between *Gtf2i*-KO and control mice (Fig. 2I, mean control 1.609 ± 0.056 μm, mean *Gtf2i*-KO 1.536 ± 0.064 μm), the number of NoRs per field of view (FOV) was significantly reduced in *Gtf2i*-KO mice compared to control mice (Fig. 2J, mean control 20.812 ± 1.213, mean *Gtf2i*-KO 15.25 ± 0.7).

Axonal conductivity is largely affected by myelin thickness and axon diameter[128,132], as well as by the length and number of NoRs[133]. The increased axonal diameters, thicker myelin, and decreased NoR numbers in the CC of *Gtf2i*-KO mice prompted us to investigate whether these changes impacted electrical conductivity in the CC. To achieve this, we stimulated the CC axonal bundle in one hemisphere of an anesthetized mouse and recorded the evoked field excitatory postsynaptic potentials (fEPSP) in the other hemisphere (Fig. 2K). In accordance with the increased diameter and myelin thickness of axons in the CC, *Gtf2i*-KO mice exhibited significantly shorter fEPSP latency (Fig. 2L, mean control 4.821 ± 0.147 ms, mean *Gtf2i*-KO 3.913 ± 0.244 ms) and significantly increased fEPSP slope (mV/ms) values of the maximal response, compared to controls (Fig. 2M, mean control 0.171 ± 0.0282 mV/ms, mean *Gtf2i*-KO 0.284 ± 0.0434 mV/ms). In addition, the input (stimulation current) - output (fEPSP slope) curve revealed significantly increased slope values (mV/ms) for the different stimulation intensities in *Gtf2i*-KO mice, compared to controls (Fig. 2N), suggesting an enhanced response in the CC of the *Gtf2i*-KO mice.

Together, these data demonstrate that the deletion of *Gtf2i* from mOLs results in significant alterations in myelination. These alterations are reflected in the modified myelin ultrastructure observed in the deletion strain and persist into adulthood (Supplementary Fig. 4). Importantly, these alterations have physiological implications, leading to enhanced conductivity in the CC.

## Hypermyelination of the SN in *Gtf2i*-KO mice

Since the *Cnp* promoter is also active in SCs of the PNS, we examined the myelin ultrastructure in the tibial branch of the SN of P30 mice by TEM (Fig. 3A). The *g*-ratio values of *Gtf2i*-KO mice axons were significantly lower compared to controls (Fig. 3B-C), indicating a hypermyelination phenotype in the PNS of *Gtf2i*-KO mice. We specifically observed that small and medium caliber axons present with significantly smaller *g*-ratio values in *Gtf2i*-KO mice, as compared to controls (Fig. 3D-E, mean control 0.618 ± 0.01, 0.635 ± 0.01, 0.664 ± 0.007, mean *Gtf2i*-KO 0.568 ± 0.004, 0.6 ± 0.006, 0.658 ± 0.011, for axon calibers of 0-2, 2-4, and 4-6 μm, respectively). *Gtf2i*-KO mice axons in the tibial branch of the SN also exhibited significantly increased myelin thickness compared to control mice (Fig. 3F). Specifically, in accordance with *g*-ratio values,

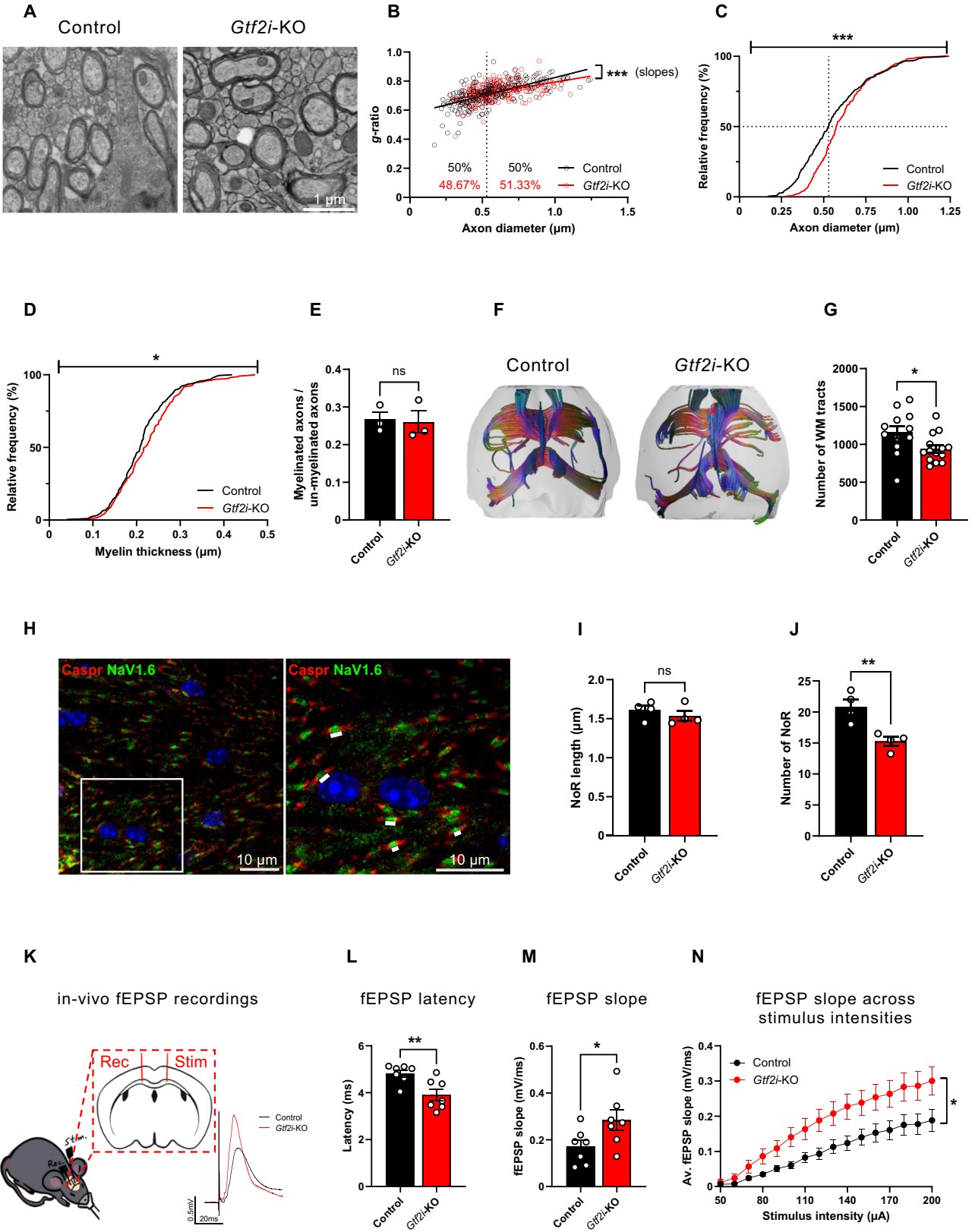

myelin thickness was significantly increased in small and medium caliber axons of *Gtf2i*-KO mice, as compared to controls (Fig. 3G, mean control $0.508 \pm 0.022\,\mu m$, $0.837 \pm 0.033\,\mu m$, $1.226 \pm 0.029\,\mu m$, mean *Gtf2i*-KO $0.61 \pm 0.014\,\mu m$, $0.988 \pm 0.036\,\mu m$, $1.239 \pm 0.065\,\mu m$, for axon calibers of 0-2, 2-4, and 4-6 μm, respectively). Axonal diameters remained unchanged between *Gtf2i*-KO and control mice (Fig. 3H). Decreased *g*-

ratio values have previously been shown to be associated with abnormalities in myelin ultrastructure[135], often resulting from deficits in myelin compaction. To assess whether myelin ultrastructural properties in the SN are affected by *Gtf2i* deletion we quantified the number of myelin deformations in myelinated axons of the SN, revealing unchanged prevalence of deformations between *Gtf2i*-KO and control mice

**Fig. 2 | Altered axonal and myelin properties with enhanced conduction characteristics in the CC of *Gtf2i*-KO mice. A** Representative TEM images of axons from the CC of control and *Gtf2i*-KO mice. **B** Scatter plot of *g*-ratio values and their respective axon diameters. The dashed line indicates the median diameter of control mice axons (0.53 μm), while the numbers on each side of the line represent the percentage of axons below (left) and above (right) this value for each genotype (two-sided simple linear regression, slopes $P = 0.0045$). **C** Myelinated axon diameter distribution demonstrates significantly larger axonal diameters in *Gtf2i*-KO mice ($P = 0.0004$). **D** Myelin thickness of *Gtf2i*-KO axons is significantly increased, as compared to controls ($P = 0.0231$). **E** Myelinated to unmyelinated axons ratio is unchanged in *Gtf2i*-KO mice, compared to controls ($n = 3$, $P = 0.86$). **F** Representative images of tractography analysis from the entire CC of control and *Gtf2i*-KO mice. **G** The CC of *Gtf2i*-KO mice presents with decreased number of WM tracts, as compared to controls ($n = 12$ control, $n = 14$ *Gtf2i*-KO, $P = 0.03$). **H** Representative image of NoR staining at the midline CC. NoRs are marked on the zoomed-in (right) image (white lines). Cells nuclei are stained with DAPI. **I** The midline CC of *Gtf2i*-KO mice presents with unchanged NoR length ($P = 0.428$) and **J** decreased number of NoRs per FOV, as compared to controls ($P = 0.0074$). **K** Illustration of in-vivo EPSP recordings experiment, electrodes placement in the CC and representative traces from recordings of both control and *Gtf2i*-KO mice. **L-N** Electrophysiological recordings from the CC ($n = 7$ control, $n = 7$ *Gtf2i*-KO), in-vivo. **L** Significantly shorter fEPSP latencies across the CC in *Gtf2i*-KO mice ($P = 0.0078$), with (**M**) significantly higher slope (mV/ms) values ($P = 0.0497$) compared to controls. **N** Average slope (mV/ms) values are significantly steeper in *Gtf2i*-KO mice across different stimulus intensities (two-sided mixed-effects analysis, $P = 0.042$). **B–D** $n = 3$ control, 231 axons. $n = 3$ *Gtf2i*-KO, 226 axons. **I, J** $n = 4$ control, 315 NoRs. $n = 4$ *Gtf2i*-KO, 301 NoRs. **C, D** Two-sided Kolmogorov-Smirnov test. **E, G, I, J, L, M** Two-sided t-test. **E, G, I, J, L–N** Data are presented as mean values ± SEM. ns−non-significant, * $P < 0.05$, ** $P < 0.01$, *** $P < 0.001$. Source data are provided as a Source Data file.

(Fig. 3I, mean control $24.121 \pm 2.193\%$, mean *Gtf2i*-KO $25.971 \pm 3.466\%$). These findings indicate that *Gtf2i* deletion in SCs results in hypermyelination of small- and medium-caliber axons in the tibial branch of the SN without affecting axonal diameter or myelin ultrastructure.

While this highlights the role of *Gtf2i* in regulating myelination in myelinating SCs, its influence on non-myelinating SCs remains unknown. The PNS comprises various types of non-myelinating SCs, including Remak SCs, which ensheath multiple small-caliber axons to form 'Remak's bundles'[136] (Fig. 3J). To assess whether *Gtf2i* deletion affects these unmyelinated axons, we quantified the number of the unmyelinated axons in the SN of *Gtf2i*-KO and control mice, and the ratio between myelinated and unmyelinated axons, revealing unaltered properties (Fig. 3K, mean myelinated control $670.33 \pm 37.922$, mean myelinated *Gtf2i*-KO $629.33 \pm 55.167$, mean unmyelinated control $1474 \pm 172.2$, mean unmyelinated *Gtf2i*-KO $1422.33 \pm 122.265$).

These results indicate that *Gtf2i* deletion from SCs results in hypermyelination of the SN, without affecting axonal properties, or altering myelin integrity and compaction properties.

### *Gtf2i*-KO mice exhibit improved motor coordination, moderately increased anxiety, and increased sociability

Motor coordination was shown to be dependent on myelin condition[137–139]. Key regions involved in motor coordination include the cerebellum in the CNS, a brain region responsible for the control, timing and coordination of motor functions[140], and the SN in the PNS[141,142]. In addition, deficits in CC myelination have been linked to abnormal anxiety behavior[143,144]. Given the observed alterations in CNS and PNS myelin ultrastructure in *Gtf2i*-KO mice, we next assessed whether these alterations had behavioral consequences.

To determine whether motor coordination was affected in our test mice, both *Gtf2i*-KO and control mice were assessed in a rotarod test. The results showed that the time that passed before falling off the rotating rod was significantly longer for *Gtf2i*-KO mice than for controls (Fig. 4A), indicative of improved balance and motor coordination following the reduction of Tfii-i levels in myelinating cells.

To determine whether the improved performance of *Gtf2i*-KO mice in the rotarod test was due to altered locomotor activity, we tested mice in an open-field exploration test. *Gtf2i*-KO mice exhibited a higher level of anxiety-like behavior, spending significantly more time in the margins of the arena than did controls (Fig. 4B). Yet, the distance traveled and the number of entries to the center of the arena were not significantly different between *Gtf2i*-KO and control mice (Supplementary Fig. 4). To further examine anxiety-like behavior, control and *Gtf2i*-KO mice were subjected to the elevated zero maze (EZM) test. The results showed a trend towards higher anxiety levels in *Gtf2i*-KO mice, indicated by their spending less time in the open arms compared to controls (Fig. 4C, mean control $249.1 \pm 19.62$ s, mean *Gtf2i*-KO $200.3 \pm 16.59$ s).

Among the genes deleted in WS, *Gtf2i* haploinsufficiency has been directly linked to the distinctive hypersocial phenotype[95,98,100,104,105]. To investigate whether the deletion of *Gtf2i* from myelinating cells affects social behavior, we employed the three-chambers paradigm to evaluate the sociability of *Gtf2i*-KO mice. Both control and *Gtf2i*-KO mice exhibited a preference for interacting with the stranger mouse over the inanimate object (Fig. 4D). Although the total interaction time with the stranger mouse and inanimate object did not differ significantly between the two strains (Fig. 4D, mean control $302.428 \pm 18.798$ s, mean *Gtf2i*-KO $337.282 \pm 14.99$ s), the social index, calculated as the ratio of time spent interacting with the stranger mouse to the time spent interacting with the inanimate object, was significantly higher in *Gtf2i*-KO mice compared to control mice (Fig. 4E, mean control $1.49 \pm 0.103$, mean *Gtf2i*-KO $1.817 \pm 0.1$). This increase in the social index suggests that the deletion of *Gtf2i* from myelinating cells enhances sociability in mice.

Together, these data suggest that *Gtf2i*-KO mice possess improved motor coordination, moderately increased anxiety-like behavior, and demonstrate increased sociability, as compared to controls.

### Tfii-i reduction in mOLs results in elevated expression levels of myelin-related proteins and transcripts

To assess how Tfii-i reduction in mOLs alters myelin properties on a molecular level, we examined the biochemically-enriched myelin fraction[145] purified from a single hemisphere of P30 mice. Using an entire hemisphere provided us a broader representation of whole-brain myelin composition, allowing us to comprehensively assess the impact of *Gtf2i* deletion from mOLs on myelin molecular properties. The protein composition of the purified myelin fraction was assessed by label-free quantitative mass spectrometry[146]. Proteome analysis revealed altered protein abundance of several key myelin proteins (Fig. 5A, Supplementary Data 1, and Supplementary Fig. 6). Specifically, the relative abundance of myelin oligodendrocyte glycoprotein (Mog) and Mbp was significantly elevated in myelin purified from *Gtf2i*-KO mice (Fig. 5A), while the abundance of Gpr37 was reduced. Mbp is an abundant constituent of compact myelin, its expression being a rate-limiting step in myelination[63]. Mog is a marker for the abaxonal myelin membrane[147]. Gpr37 has been demonstrated as a negative regulator of myelination[148]. Overall, the altered myelin proteome observed in *Gtf2i*-KO mice suggests a molecular enhanced phenotype following Tfii-i reduction in mOLs.

Next, to determine whether the altered physiological and behavioral properties observed in *Gtf2i*-KO mice are accompanied by molecular alterations specifically in cortical myelin, we assessed Mbp protein and transcript expression levels in the pure myelin fraction derived from the cortices of P30 mice. *Mbp* mRNA is known to be transported from the nucleus to the plasma membrane, where it is

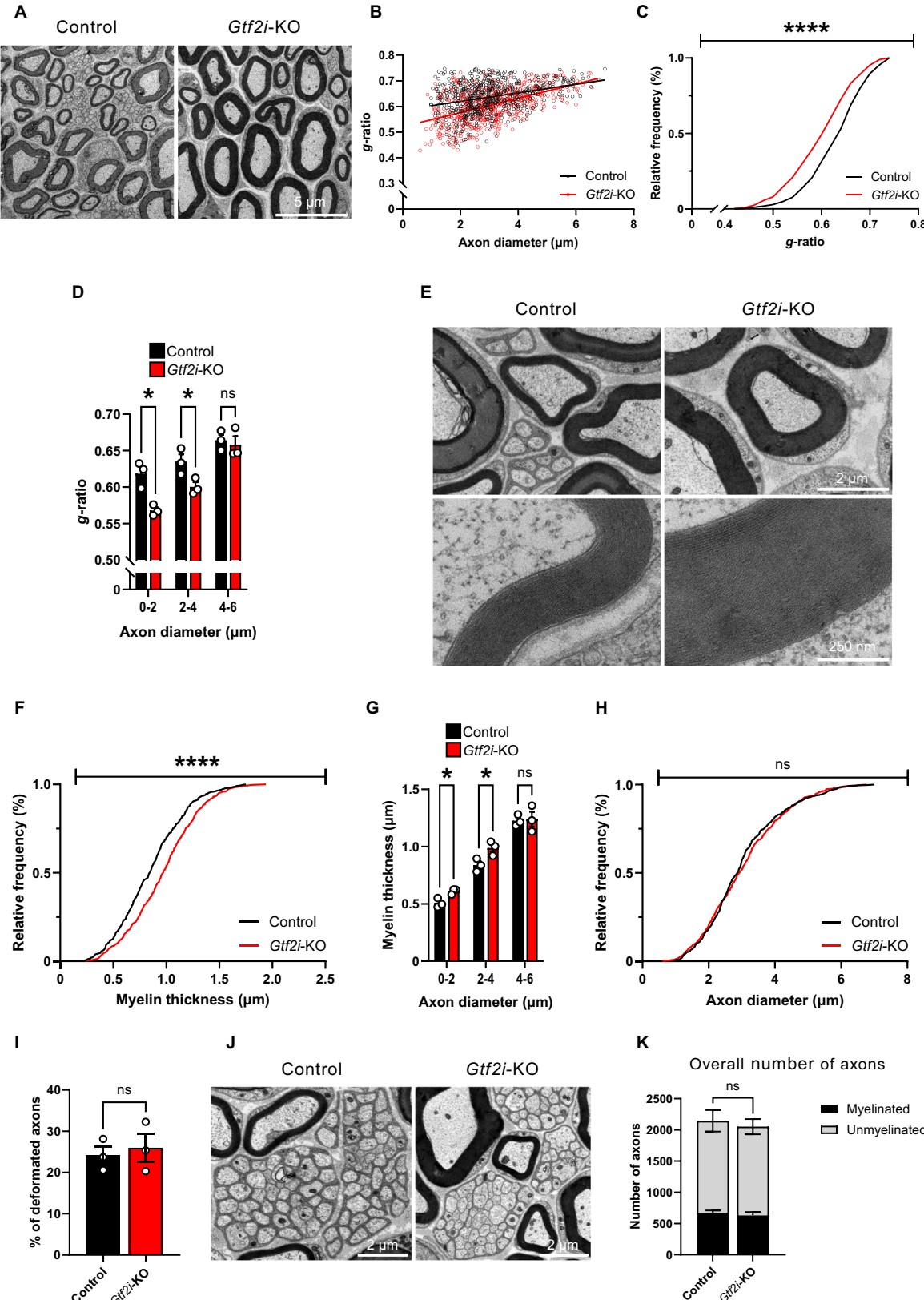

translated at the axon-glia contact site[149–151]. This fraction of *Mbp* transcripts is first in line to be translated into active protein. Additionally, *Mbp* transcript levels were also shown to affect myelin sheath properties[152]. Consistent with the proteome results, Mbp protein (Fig. 5B, mean control 1 ± 0.054, mean *Gtf2i*-KO 1.164 ± 0.058) and transcript (Fig. 5C, mean control 1 ± 0.132, mean *Gtf2i*-KO 1.744 ± 0.218)

expression levels in cortical myelin were elevated in *Gtf2i*-KO mice, as compared to controls.

In searching for additional altered molecular properties in *Gtf2i*-KO mice, we examined the transcript cortical expression levels of the OL and myelin-relevant TF, Sox10. Sox10 is a major TF in the oligodendroglial lineage, and has been shown to positively regulate the

**Fig. 3 | Hypermyelination of the SN of *Gtf2i*-KO mice. A** Representative TEM images of axons from the SN of control and *Gtf2i*-KO mice. **B** Scatter plot of *g*-ratio values and their respective axon diameters. **C** *Gtf2i*-KO mice axons present with significantly lower *g*-ratio values, compared to controls ($P = 1.12*10^{-10}$). **D** Specifically small and medium caliber axons present with lower *g*-ratio values in *Gtf2i*-KO mice, as compared to controls ($P = 0.01, 0.046, 0.7$, for axons with diameters of 0–2 μm, 2–4 μm, and 4–6 μm, respectively). **E** *Gtf2i*-KO small caliber axons present increased myelin thickness, representative images. **F** Myelin thickness of *Gtf2i*-KO axons is significantly increased, as compared to controls ($P = 3.72*10^{-8}$). **G** Small and medium caliber axons specifically present with increased myelin thickness in *Gtf2i*-KO mice, as compared to controls ($P = 0.019, 0.037, 0.863$, for axons with

diameters of 0–2 μm, 2–4 μm, and 4–6 μm, respectively). **H** Axonal diameter is unchanged in *Gtf2i*-KO mice, compared to controls ($P = 0.153$). **I** Similar number of myelin abnormalities in the SN of control and *Gtf2i*-KO mice ($P = 0.675$). **J** Representative TEM images of Remak's bundles from the SN of control and *Gtf2i*-KO mice. **K** Unchanged ratio of myelinated, unmyelinated, and overall number of axons in the SN of control and *Gtf2i*-KO mice ($n = 3$, $P = 0.573$, $P = 0.819$, $P = 0.71$ for myelinated, unmyelinated, and overall number of axons accordingly). **B–D, F–I** $n = 3$ control, 597 axons. $n = 3$ *Gtf2i*-KO, 599 axons. **C, F, H** Two-sided Kolmogorov-Smirnov test. **D, G, I, K** Two-sided t-test. Data are presented as mean values ± SEM. ns – non-significant, * $P < 0.05$, **** $P < 0.0001$. Source data are provided as a Source Data file.

expression of *Mbp*[34]. Additionally, we examined *Mbp* and *Mog* mRNA expression levels in whole cortex tissue to further validate the proteome data. Our analysis revealed significantly higher expression of *Sox10* (Fig. 5D, mean control $1 ± 0.064$, mean *Gtf2i*-KO $1.375 ± 0.059$), *Mbp* (Fig. 5E, mean control $1 ± 0.06$, mean *Gtf2i*-KO $1.483 ± 0.0107$), and *Mog* (Fig. 5F, mean control $1 ± 0.041$, mean *Gtf2i*-KO $1.281 ± 0.083$) mRNA in *Gtf2i*-KO mice, as compared to controls. These findings suggest that Tfii-i potentially regulates the expression of these genes in mOLs.

*Gtf2i* and Tfii-i's expression levels change across different developmental stages[153]. To assess whether the observed molecular alterations persist into adulthood, we examined the pure myelin fraction derived from the cortices of P90 mice. Our analysis showed increased protein expression levels of Mbp (Fig. 5G, mean control $1 ± 0.027$, mean *Gtf2i*-KO $1.164 ± 0.066$) and Mog (Fig. 5H, mean control $1 ± 0.035$, mean *Gtf2i*-KO $1.217 ± 0.067$). Additionally, we examined mRNA expression levels of *Sox10* (Fig. 5I, mean control $1 ± 0.102$, mean *Gtf2i*-KO $1.279 ± 0.063$), *Mbp* (Fig. 5J, mean control $1 ± 0.09$, mean *Gtf2i*-KO $1.34 ± 0.071$), and *Mog* (Fig. 5K, mean control $1 ± 0.047$, mean *Gtf2i*-KO $1.328 ± 0.051$) in the cortex of P90 *Gtf2i*-KO mice, revealing increased transcript expression levels of these genes compared to controls.

Overall, these molecular alterations detected in *Gtf2i*-KO mice suggest that Tfii-i plays a crucial role in regulating the myelination process in mOLs across different developmental stages. Its reduction leads to elevated mRNA and protein expression levels of key myelin components.

### Exogenous viral-mediated deletion of *Gtf2i* in mOL-enriched primary cell culture results in increased expression of key myelin proteins and cell surface area

To assess whether the regulation of Tfii-i on *Sox10* and *Mbp* transcription occurs in a cell-specific manner, we prepared mOL-enriched primary cell cultures from *Gtf2i* conditional KO mice (*Gtf2i*f/f;Cre-/-)[115], followed by the introduction of an iCre-expressing adeno-associated virus (AAV) (Fig. 6A) to allow for exogenous *Gtf2i* deletion in mOLs. Upon iCre recombination, a significant reduction in Tfii-i expression was measured, as compared to cells infected with mCherry-expressing control AAV that does not induce iCre recombination (Fig. 6B-C, mean control 1, mean iCre $0.278 ± 0.099$). Furthermore, both nuclear Sox10 (Fig. 6D-E, mean control 1, mean iCre $1.713 ± 0.153$) and cellular Mbp (Fig. 6F-G, mean control 1, mean *Gtf2i*-KO $1.314 ± 0.069$) levels were significantly increased following the exogenous deletion of *Gtf2i* in mOLs. In addition, overall cell surface area was increased in mOLs infected with iCre-expressing AAV, as compared to mOLs infected with the control mCherry vector (Fig. 6H, mean control $4930 ± 288.9$ μm, mean iCre $8145 ± 361.2$ μm). While not a direct marker of myelination, the increased cell surface area may suggest that, in mOLs, *Gtf2i* regulates process extension and outgrowth, essential for myelination[154].

These findings suggest that, following Tfii-i reduction in mOLs, the observed increased expression levels of Mbp and Sox10 are a result of a cell-autonomous mechanism in mOLs, unrelated to other cell types (Supplementary Fig. 8).

### Tfii-i binds to *Mbp* and *Sox10* RE in mOLs

In searching for a cellular mechanism to explain the altered myelination properties in *Gtf2i*-KO mice, we characterized the general OL population and the specific mOL pool in the motor cortex and CC using immunohistochemistry (Supplementary Fig. 9, 10). Our analysis revealed that the total number of oligodendroglial lineage cells (Olig2+ cells) and mOLs (Olig2+ + CC1+ cells) in the motor cortex and CC from both strains did not differ significantly (Supplementary Fig. 10). To ensure that differentiation properties remain unchanged throughout development, we performed similar experiments at postnatal ages of 14 and 90 days (Supplementary Fig. 10), as well as in-vitro using a primary mOL-enriched culture (Supplementary Fig. 11). Overall, no significant differences in differentiation were found between *Gtf2i*-KO and control mice.

As such, changes in the number of mOLs cannot account for the observed myelination and axonal alterations in *Gtf2i*-KO mice, suggesting that these alterations are due to a cell-autonomous mechanism triggered by the deletion of *Gtf2i* in mOLs. However, the precise mechanisms by which Tfii-i modulates the expression of myelination-related genes remain inadequately understood. To unravel the nature of its regulation, it is crucial to identify the specific DNA loci to which Tfii-i directly binds in mOLs. By pinpointing these binding sites, we can gain valuable insights into Tfii-i's mode of action in mOLs. Consequently, in the next phase of our research, we aimed to comprehensively map the genome-wide Tfii-i binding sites and enrichment levels, specifically in mOLs.

To achieve this, fully differentiated mOLs, derived from primary mOL-enriched cell cultures, were subjected to chromatin immunoprecipitation followed by sequencing (ChIP-seq), targeting Tfii-i. As expected, Tfii-i peaks were highly abundant in control compared to *Gtf2i*-KO samples (Fig. 7A). Overall, 10,467 unique genomic loci were identified as high-confidence Tfii-i binding sites in control samples (i.e., consensus peaks), with minimal overlap in *Gtf2i*-KO samples (Fig. 7A, Supplementary Data 2, Supplementary Fig. 12A-C). These findings confirm the robustness of the ChIP-seq protocol and validate the specificity of Tfii-i binding. Given the near absence of binding in *Gtf2i*-KO samples, subsequent analyses focus exclusively on Tfii-i binding properties in control samples. This approach allows us to comprehensively characterize its genomic distribution and transcriptional regulatory potential without interference from the knockout condition, ensuring a clearer interpretation of its functional role. Functional annotation analysis revealed that Tfii-i binding sites in control samples were predominantly enriched in intergenic regions and introns; intergenic (6237 peaks), introns (3776 peaks), exons (252 peaks), transcription termination sites (TTS) (114 peaks), and promoters (88 peaks) (Fig. 7B). These data correlate well with our previous findings, where we identified cell-type-specific aberrant methylation profiles on enhancers in post-mortem brain tissue of individuals with WS[155]. Motif analysis of Tfii-i consensus peaks in mOLs within introns, exons, TTS, and promoters (i.e., across the gene body) revealed enrichment of Olig2 and Sox10 motifs, with the Olig2 motif being particularly prominent, alongside known Tfii-i binding motifs[113,156] (Fig. 7C,

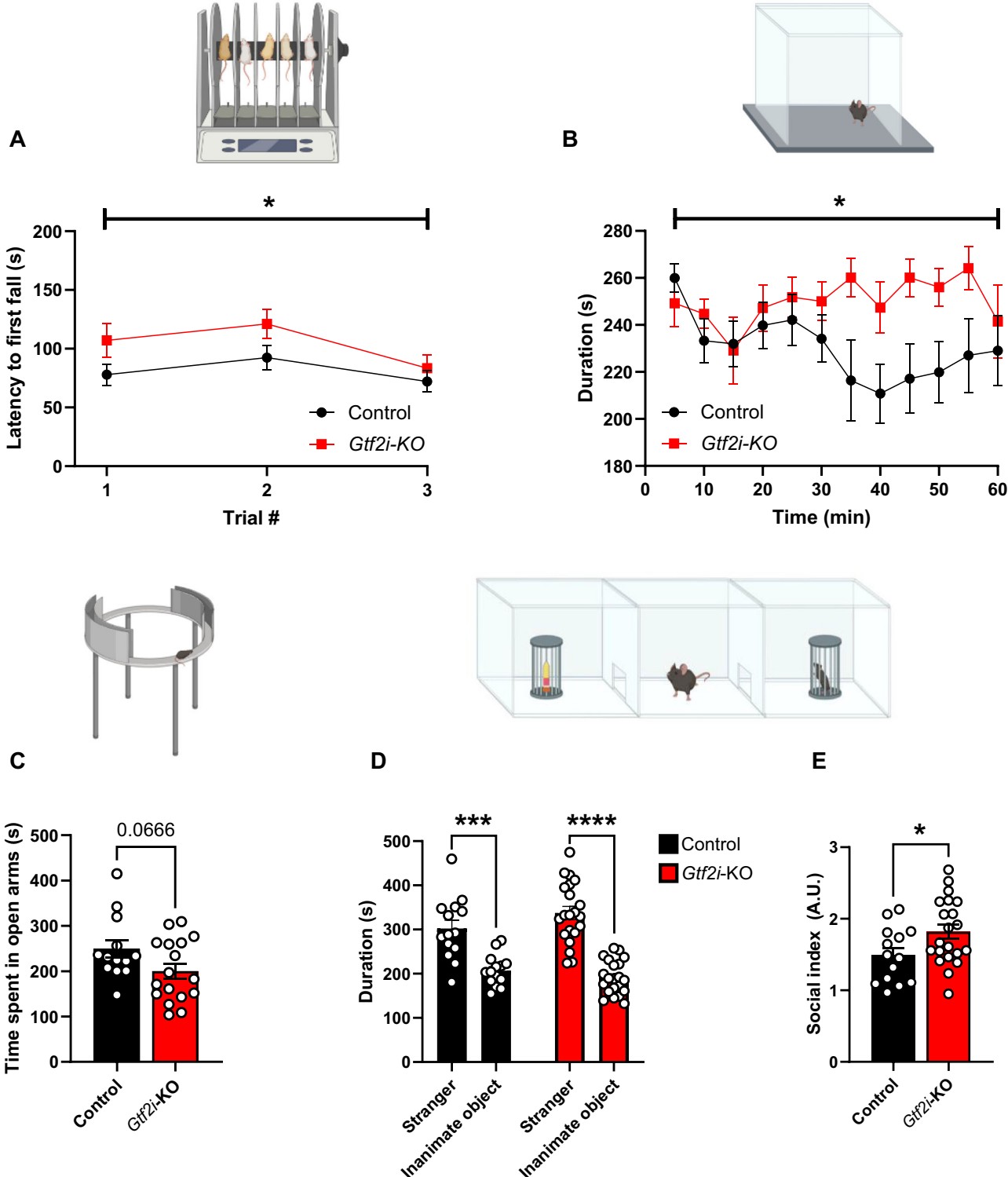

**Fig. 4 | The behavioral phenotype of *Gtf2i*-KO mice reveals improved motor coordination, increased anxiety-like behavior, and enhanced sociability.**
**A** Rotarod test. *Gtf2i*-KO mice show improved motor coordination as their latency to the first fall is significantly higher compared to controls along three trials of the rotarod test ($n = 19$ control, $n = 15$ *Gtf2i*-KO, two-sided mixed-effects analysis, $P = 0.047$). **B** Open field test. *Gtf2i*-KO mice show increased anxiety-like behavior as they spend significantly more time in the margins of the arena compared to controls ($n = 15$ control, $n = 14$ *Gtf2i*-KO, two-way ANOVA, $P = 0.0325$). **C** EZM test ($n = 13$ control, $n = 17$ *Gtf2i*-KO). *Gtf2i*-KO mice show a trend towards increased anxiety as they spend less time in the open arms of the EZM compared to controls ($P = 0.066$). **D, E** Three-chambers sociability test ($n = 14$ control, $n = 21$ *Gtf2i*-KO). **D** While both control and *Gtf2i*-KO mice show preference towards interacting with the social stimulus rather than with the inanimate object (control - $P = 0.0001$, *Gtf2i*-KO - $P = 1.71*10^{-10}$), *Gtf2i*-KO mice demonstrate increased sociability as their **E** social index is higher, as compared to controls ($P = 0.035$). **C–E** Two-sided t-test. **A–E** Data are presented as mean values ± SEM. * $P < 0.05$, *** $P < 0.001$, **** $P < 0.0001$. Source data are provided as a Source Data file. Created with BioRender. Rokach, M. (2025) https://BioRender.com/o51w712.

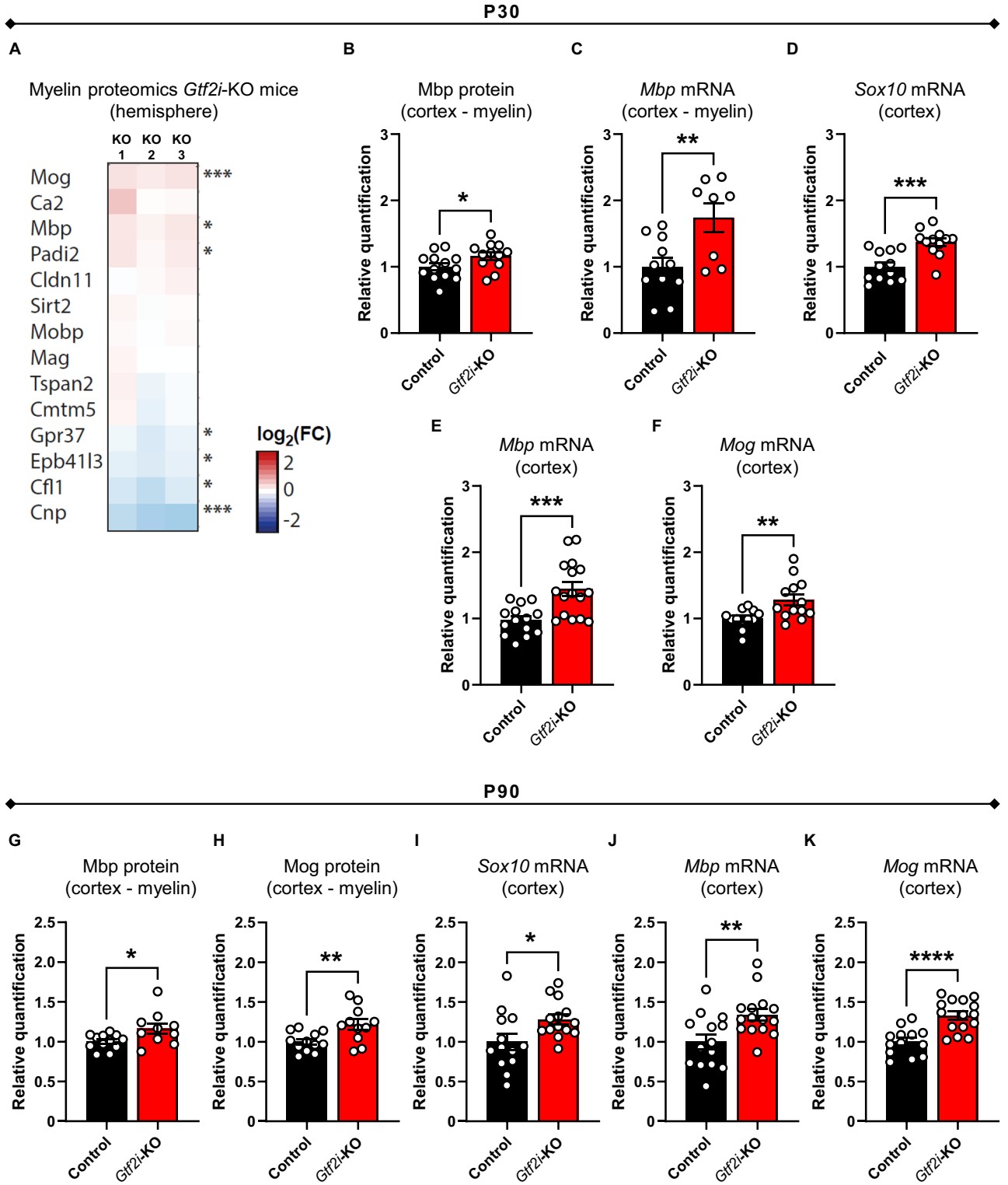

Supplementary Fig. 12D-F). To further investigate this potential interaction, we performed colocalization analysis of Tfii-i peaks in mOLs using publicly available ChIP-seq datasets. This analysis demonstrated substantial overlap between Tfii-i and Olig2 binding sites, as well as with Srf, another TF expressed in OLs[157], whereas overlap with the neuronal TF Neurod1 was minimal (Fig. 7D). Olig2 is a core regulator of OL differentiation and myelination, functioning through a variety of cofactors and binding partners[36,48,158–163]. Collectively, these findings

suggest that Tfii-i may play a broad role in a genome-wide transcriptional network coordinated by Olig2 in mOLs.

To dissect the specific contribution of Tfii-i to the transcriptional program of mOLs, we integrated our ChIP-seq data with a cell type-specific database of post-translational histone modifications (PTMH), which includes datasets derived specifically from mOLs[164]. This integrative approach enabled us to contextualize our findings within the broader epigenetic landscape, providing deeper insights into how Tfii-i

**Fig. 5 | The molecular myelin framework is altered in *Gtf2i*-KO mice, with increased expression levels of key myelin factors, along different developmental stages. A** Heatmap showing the relative abundance of selected myelin proteins in *Gtf2i*-KO compared to control myelin derived from the hemisphere of P30 mice. Data represent n = 3 mice per genotype analyzed as four technical replicates per mouse. Note that the relative abundance of several proteins was significantly increased (Mog, Mbp, Padi2) or decreased (Gpr37, Epb41l3, Cfl1) in *Gtf2i*-KO mice myelin. The abundance of Cnp was reduced in *Gtf2i*-KO myelin reflecting that the Cre driver line possesses only one *Cnp* allele. Exact q-values are given in Supplementary Data 1 and were calculated by moderated t-statistics and corrected for multiple comparisons using the Bioconductor R packages 'limma' and 'q-value', respectively. * q < 0.05, *** q < 0.001 (**B**) Mbp protein expression level is significantly elevated in the myelin fraction derived from the cortices of *Gtf2i*-KO mice, as compared to controls (n = 13 control, n = 12 *Gtf2i*-KO, P = 0.048). **C** *Mbp* mRNA expression level in the myelin fraction derived from the cortices of P30 mice is significantly elevated in *Gtf2i*-KO mice, as compared to controls (n = 11 control,

n = 8 *Gtf2i*-KO, P = 0.0067). **D**–**F** Quantitative PCR (qPCR), whole cortex mRNA, P30 mice. **D** *Sox10* (n = 12 control, n = 12 *Gtf2i*-KO, P = 0.0003) **E** *Mbp* (n = 14 control, n = 16 *Gtf2i*-KO, two-sided Welch's t-test, P = 0.0006), and **F** *Mog* (n = 12 control, n = 13 *Gtf2i*-KO, P = 0.0071) mRNA expression levels are significantly elevated in *Gtf2i*-KO mice, compared to controls. **G**, **H** Western blot assay on myelin fraction derived from the cortices of P90 mice. **G** Mbp (n = 12 control, n = 10 *Gtf2i*-KO, P = 0.0247) and **H** Mog (n = 12 control, n = 11 *Gtf2i*-KO, P = 0.008) protein expression levels are significantly elevated in *Gtf2i*-KO mice, compared to controls. **I**–**K** qPCR, whole cortex mRNA, P90 mice. **I** *Sox10* (n = 13 control, n = 14 *Gtf2i*-KO, P = 0.0269), **J** *Mbp* (n = 14 control, n = 15 *Gtf2i*-KO, P = 0.006), and **K** *Mog* (n = 13 control, n = 15 *Gtf2i*-KO, P = 8.3*10$^{-5}$) mRNA expression levels are significantly elevated in the cortex of P90 *Gtf2i*-KO mice, as compared to controls. **B**–**K** control normalized to 1. Protein levels were normalized to β-tubulin IV and mRNA levels were normalized to *Gapdh*. Data are presented as mean values ± SEM. **B**–**D**, **F**–**K** Two-sided t-test. * P < 0.05, ** P < 0.01, *** P < 0.001, **** P < 0.0001. Source data are provided as a Source Data file.

---

regulates gene expression in mOLs. Tfii-i peaks in mOLs were predominantly located at genomic regions enriched with the post-translational histone modification H3K36me3 (Histone H3 lysine 36 tri-methylations), a marker of actively transcribed euchromatin and transcriptional elongation[165,166] (Fig. 7E, Supplementary Fig. 12G). Among the high-confidence Tfii-i peaks, notable binding enrichment was observed at relatively evolutionarily conserved regions (Supplementary Fig. 12H) within the loci of *Mbp* (Fig. 7F, Supplementary Fig. 12I) and *Sox10* (Fig. 7G, Supplementary Fig. 12I), both of which are critical for mOL function. At the *Mbp* locus, the Tfii-i binding site co-localized with the H3K4me3 mark[164], indicative of active promoters, as well as the H3K36me3 marker (Fig. 7F, Supplementary Fig. 12J). These findings suggest that in mOLs, Tfii-i directly interacts with RE of *Mbp*, potentially contributing to its transcriptional regulation.

Although we did not observe direct Tfii-i binding at the *Sox10* genomic loci, at known *Sox10* enhancers[167], or at established Olig2 binding sites within the *Sox10* locus (Supplementary Fig. 12K), we did identify high-confidence Tfii-i peaks at two cis-REs (cREs) predicted to physically interact with the *Sox10* promoter region (Fig. 7F, peaks 44 and 69). These predictions were previously assessed in-silico by applying the Cicero[168] tool on mOL-derived single-cell CUT&Tag data[164]. To experimentally assess these promoter–cREs interactions, we performed a chromatin conformation capture assay followed by qPCR (3C-qPCR) on cells derived from the cortex of P30 control mice and from an mOL-enriched primary culture. Interaction frequencies between the *Sox10* promoter and candidate cREs, as well as other control loci within the *Sox10* region, were quantified and normalized to the interaction frequency at the *Ercc3* locus, a commonly used reference in 3C-qPCR assays[169] (Fig. 7H). Consistent with standard 3C-qPCR profiles, we observed high interaction frequencies at proximal genomic loci near the *Sox10* promoter (e.g., the 5 Kb region), reflecting efficient proximal ligation. These frequencies declined at more distal sites (e.g., 35 Kb), but showed a local increase at the positions of the putative cREs (peaks 69 and 44), before declining again at more distant regions (65 Kb and 85 Kb). Notably, this interaction pattern was observed in data derived from both whole cortex and mOL-enriched primary culture samples. We also detected elevated interaction frequency at D7, a previously characterized downstream enhancer of *Sox10*, further supporting the regulatory role of these cREs[167]. These findings demonstrate that the *Sox10* promoter physically interacts with the specific predicted cREs. While this alone does not constitute definitive evidence of enhancer activity, it highlights candidate regulatory loci and suggests that Tfii-i binding may, in some cases, coincide with active regulatory elements. As enhancer-promoter interactions were previously shown to play a crucial role in OL transcriptional program[170,171], our results support the notion that Tfii-i may contribute to *Sox10* transcriptional regulation.

Collectively, our findings suggest that in mOLs, Tfii-i is involved in transcriptional regulation, with prominent engagement at key loci such as *Sox10* and *Mbp*. This regulatory role may contribute to the observed altered interplay of myelin thickness and axonal diameter observed in the CC of *Gtf2i*-KO mice, further supporting a cell-autonomous effect of Tfii-i in mOLs.

## Discussion

In the present study, we identified Tfii-i as a regulator of the broad myelin transcriptome, affecting myelin biology by modulating the transcription of key genes such as *Mbp* and *Sox10*, specifically in mOLs. *Gtf2i*'s deletion from myelinating glial cells resulted in PNS hyper-myelination and disrupted myelin-axon dynamics in the CNS, accompanied by alterations in WM properties, enhanced axonal conductivity, and behavioral changes. Furthermore, cellular and molecular data in our study identify Tfii-i as a regulator in the transcriptional program specifically in mOLs, highlighting its cell-autonomous mode of action.

Support for this hypothesis comes from mOLs-specific ChIP-seq and 3C-qPCR data showing that Tfii-i binds to regulatory regions of *Mbp* and *Sox10*. Additionally, we show that *Gtf2i* deletion from mOLs disturbed this binding, resulting in increased levels of both *Sox10* and *Mbp*. Collectively, exogenous deletion of *Gtf2i* in vitro in an mOL-enriched primary cell culture that minimizes the effectiveness of exogenous neuronal factors, led to higher Sox10 and Mbp protein levels and an increase in process outgrowth and cell surface area. These findings support our notion that the observed myelination alterations primarily result from the loss of Tfii-i function in mOLs, making defects in secondary cell types a less likely explanation (Fig. 8).

While *Gtf2i* was shown to be present in the transcriptome of mouse brain myelin[172], the protein product, Tfii-i, was not present in the myelin fraction at detectable levels, consistent with previous proteomic characterization of mouse brain myelin[146]. Tfii-i binding to the REs of key myelin regulator *Sox10* and the key myelin component *Mbp* was previously shown[113]. However, its exact contribution to the myelin transcriptome has yet to be elucidated. Furthermore, previous data collected by ChIP-seq analysis relied on comparison of wild-type and a mouse model with complete, homozygous, deletion of *Gtf2i*, regardless of cell type, occurring at an early embryonic stage (E13.5)[113], when myelin production is low to non-existent. As such, the relevance of the reported finding in terms of the transcription-related role of Tfii-i in mOLs is limited. Specifically, as the expression pattern of many myelin-related genes changes in the different stages of the oligodendroglial lineage[173], it is possible that these data are not specific to Tfii-i binding targets in mOLs. Previous Tfii-i ChIP-seq data from E13.5 neural tissue revealed strong colocalization with the architectural protein CTCF[174], supporting a potential role in chromatin organization during early neural development. In contrast, our data show limited overlap

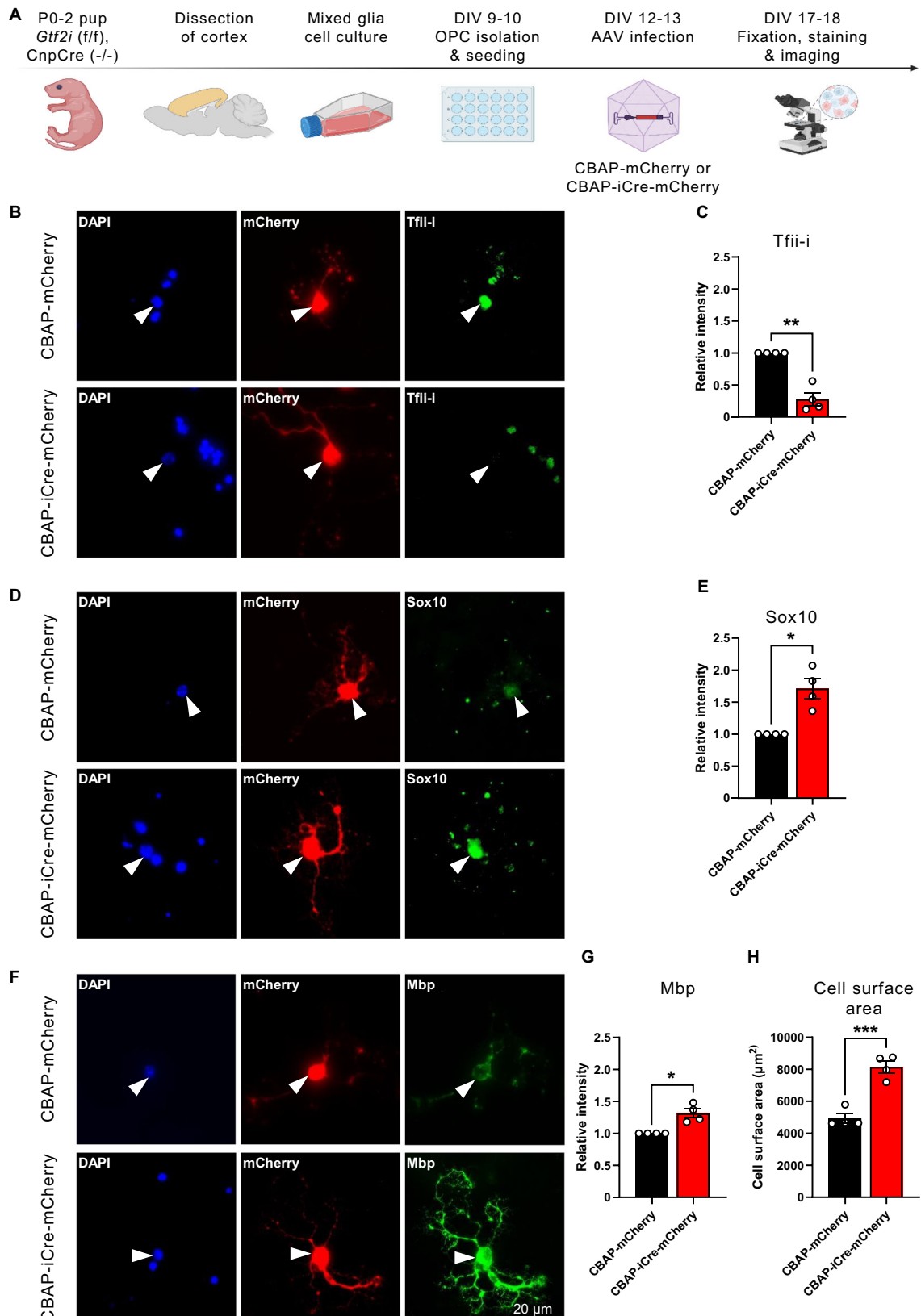

20 µm

with embryonic Tfii-i binding (Supplementary Fig. 12L-M), as well as with CTCF binding and the architectural protein Rad21 (Supplementary Fig. 12M). This discrepancy suggests a context-dependent role for Tfii-i that evolves during differentiation, potentially shifting from a chromatin architectural function during early development in embryonic neural progenitors to a more direct transcriptional

regulatory role in mOLs. Our model specifically delineates the role of *Gtf2i* in myelinating cells, as expression of Cre recombinase under the Cnp promoter in our system occurs at the post-mitotic stage, specifically targeting mOLs and mature SCs[114]. Importantly, we found that Tfii-i peaks in mOLs were predominantly localized to genomic regions enriched with the H3K36me3 PTMH modification. Collectively, our

**Fig. 6 | In-vitro deletion of Tfii-i from mOLs results in increased expression levels of nuclear Sox10, Mbp, and promotes cell growth. A** Illustration of in-vitro experimental design and workflow. Created with BioRender. Rokach, M. (2025) https://BioRender.com/o51w712. **B** Representative images of Tfii-i (green channel), staining in-vitro following viral infection (mCherry, red channel). Arrowheads point to nuclear localization. **C** Tfii-i expression level is significantly reduced in differentiated mOL-enriched primary cell culture as a result of viral infection with an iCre-expressing AAV (P = 0.0054). **D** Representative images of Sox10 (green channel), staining in-vitro following viral infection (mCherry, red channel). Arrowheads point to nuclear localization. **E** Sox10 expression level is significantly increased in differentiated OPC-enriched primary cell culture as a result of viral infection with an iCre-expressing AAV (P = 0.0187). **F** Representative images of Mbp (green channel), staining in-vitro following viral infection (mCherry, red channel). **G** Mbp expression level is significantly increased in differentiated OPC-enriched primary cell culture as a result of viral infection with an iCre-expressing AAV (P = 0.0196). **H** mOLs cell surface area is significantly increased following Tfii-i deletion in-vitro (two-sided t-test, P = 0.0004). **C, E, G, H** n = 4 CBAP-mCherry, n = 4 CBAP-iCre-mCherry. Data are presented as mean values ± SEM. **C, E, G** Two-sided one-sample t-test. * P < 0.05, ** P < 0.01, *** P < 0.001. Source data are provided as a Source Data file.

findings suggest that in mOLs, Tfii-i functions as an auxiliary regulator of *Mbp* and *Sox10* transcription. Our findings contribute to understanding the regulatory roles of Tfii-i in mOLs and provide insight into its potential influence on the myelin genes *Sox10* and *Mbp*, which we suggest to be repressive.

We previously showed that *Gtf2i* deletion from excitatory neurons led to myelination deficits[20]. These changes were also associated with various cognitive, social, and motor phenotypes of individuals with WS, and were shown to be corrected upon administration of an FDA-approved pro-myelinating drug[20]. In the present study, we instead observed an opposite effect of *Gtf2i* deletion from mOLs on myelination. Tfii-i was shown to have a dual influence on gene expression, being able to mediate both activation or repression of transcription, depending on the signal that activated Tfii-i and cell type[70]. Thus, we suggest that the mode of Tfii-i action may differ in neurons and mOLs, which may underline the differential effects on myelination seen upon deleting *Gtf2i* in different cell types. For example, while Tfii-i was shown to predominantly bind to promoters[70], here we show that in mOLs Tfii-i primarily binds to intergenic regions and introns. Furthermore, the genetic deletion performed in the current study was not restricted to a specific brain region and thus affected various nervous system elements. Finally, differences in the brain circuits affected by *Gtf2i* deletion in the two studies could explain the opposing effects on myelination and WM. While *Gtf2i* deletion from myelinating cells results in opposing effects on myelination compared to neuronal *Gtf2i* deletion[20], mice lacking *Gtf2i* expression in myelinating cells demonstrate increased sociability, similar to mice with neuronal deletion of *Gtf2i*[20]. Increased sociability is one of the most prominent behavioral phenotypes of individuals with WS[88,89], and *Gtf2i* haploinsufficiency was previously directly associated with said phenotype[95,98,100,103]. As such, our results suggest that reduction of *Gtf2i* levels, regardless of cell-type and myelin properties, results in increased sociability. Furthermore, the observed hypermyelination in the SN of *Gtf2i*-KO mice raises the possibility that PNS myelination changes contribute to motor coordination improvements, while CNS myelination abnormalities may play a role in altered anxiety and sociability behaviors. However, future studies are required to dissect the precise contribution of these alterations to behavioral phenotypes.

GTF2I has been implicated in the cellular response to DNA damage and in DNA repair pathways[175–177]. In parallel, H3K36me3, traditionally associated with transcriptionally active gene bodies[165,166], has also been shown to accumulate at sites of DNA damage and facilitate the recruitment of DNA repair machinery[178,179]. Our findings indicate that Tfii-i binding in mOLs substantially colocalizes with regions marked by H3K36me3. While DNA repair was not the primary focus of this study, this observation raises the possibility that Tfii-i may also contribute to DNA repair processes in mOLs.

Myelin, OLs, and axon health are tightly regulated by one another[124,180–184]. *Sox10* expression levels have been shown to decrease following exposure to cuprizone[185], a copper chelator known to induce demyelination[144,186]. Although postnatal *Sox10* overexpression has not been extensively studied in-vivo, current evidence suggests it can enhance remyelination and support myelin maintenance. For instance, overexpression of *Sox10* in the hippocampus following de-myelination

by cuprizone, improved behavioral deficits, myelin ultrastructural properties, and normalized myelin sheath-related protein expression levels[187]. Furthermore, in another study of a cuprizone demyelination model, upon the introduction of *Sox10* through exosomes, OPC differentiation to OLs was promoted and an increase in Mbp protein level was observed[185]. Additionally, in the embryonic chick spinal cord, overexpression of *Sox10* was shown to promote the expression of myelin-related genes[159]. In SCs derived from the rat SN, *Sox10* overexpression similarly enhanced myelination[188]. Furthermore, in the oligodendroglial cell line, Oli-neu, *Sox10* overexpression increased the expression of *Mbp* and other OL-related genes[189]. Collectively, these findings suggest that elevated *Sox10* expression may contribute to remyelination and support myelin maintenance. They also support the notion that increased *Sox10* levels could underline, at least in part, the myelination phenotype observed following *Gtf2i* deletion in myelinating glial cells.

*Gpr37* has been identified as a negative regulator of OL differentiation and myelination[148,190]. Its expression begins at the late OPC stage[148], which partially aligns with the timing of *Gtf2i* deletion using the CnpCre line[114]. Although reduced Gpr37 expression in the CNS myelin of *Gtf2i*-KO mice may contribute to the enhanced myelination phenotype, this change alone is unlikely to fully account for the observed effects. Notably, *Gpr37* depletion has been shown to result in precocious myelination through increased differentiation of pre-myelinating OLs into mature, myelin-producing OLs[148]. However, in our *Gtf2i*-KO model, we did not observe accelerated OL differentiation across multiple developmental time points (Supplementary Fig. 10). Moreover, Gpr37 expression is minimal in the SN[148], suggesting it cannot explain the hypermyelination observed in the PNS of *Gtf2i*-KO mice. Therefore, while both increased *Sox10* and decreased Gpr37 expression may contribute to the myelination phenotype, the overall effects likely reflect a combination of these transcriptional and molecular changes, alongside direct regulatory roles of Tfii-i in gene expression programs within myelinating glial cells.

In accordance with the increased axonal diameter[122–124,126], we showed that *Gtf2i* deletion from mOLs resulted in increased myelin thickness wrapping these axons in the CC. Notably, Mbp expression level is a rate-limiting consideration in CNS myelination[63]. Furthermore, *Mbp* mRNA levels were shown to affect myelin sheath properties[152]. Recently, Mbp expression was shown to support axonal regeneration in neural progenitor cells[191], and to improve neurites properties[192]. As such, excess *Mbp* transcripts and protein in *Gtf2i*-KO mice may underline the mechanism by which the myelin thickness of CC axons is increased in these mutant mice. Furthermore, in accordance with the increased axonal diameter and thicker myelin, we show enhanced signal conduction in *Gtf2i*-KO mice, as reflected in electrophysiological recordings of the CC.

Variations in NoRs number and length are associated with myelination dynamics and conductivity properties[133,134]. In a recent study, monocular deprivation was shown to result in reduced internode length and increased number of paranodes in the optic tract of mice, resulting in reduction of conduction velocity[193]. As such, the reduced number of NoRs may also contribute to the enhanced conduction properties observed in the CC of *Gtf2i*-KO mice. Full deletion of *Cnp*

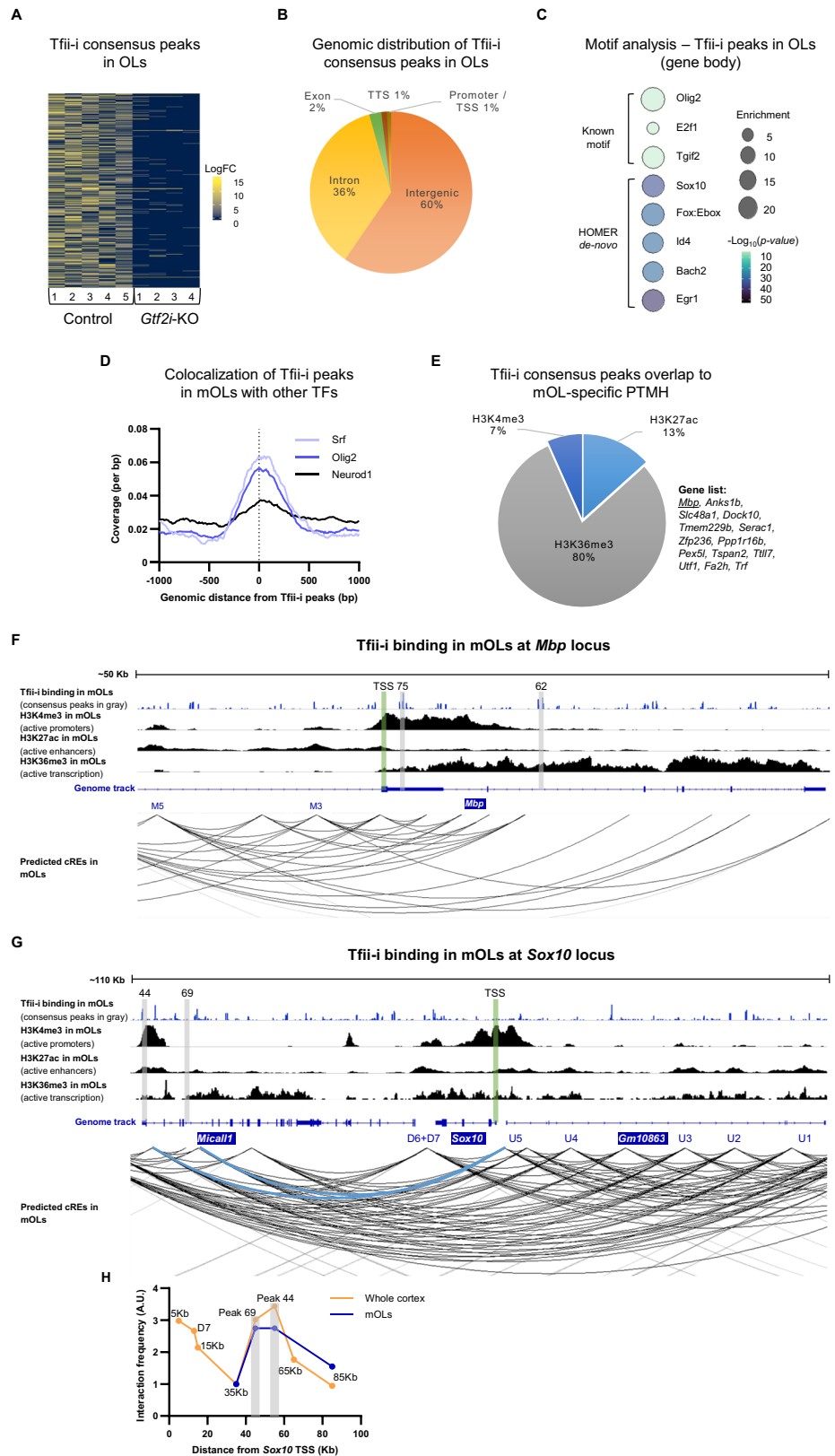

**A** Tfii-i consensus peaks in OLs

**B** Genomic distribution of Tfii-i consensus peaks in OLs

**C** Motif analysis – Tfii-i peaks in OLs (gene body)

**D** Colocalization of Tfii-i peaks in mOLs with other TFs

**E** Tfii-i consensus peaks overlap to mOL-specific PTMH

Gene list: *Mbp, Anks1b, Slc48a1, Dock10, Tmem229b, Serac1, Zfp236, Ppp1r16b, Pex5l, Tspan2, Ttll7, Utf1, Fa2h, Trf*

**F** Tfii-i binding in mOLs at *Mbp* locus

**G** Tfii-i binding in mOLs at *Sox10* locus

**H**

was previously shown to reduce the number of NoRs[194]. Therefore, we cannot entirely rule out the possibility that the reduced Cnp levels in *Gtf2i*-KO mice (resulting from the innate *Cnp* haploinsufficiency associated with the CnpCre line[114]) may contribute to the observed reduction in NoRs number. However, Rasband et al.[194] examined the effects of homozygous *Cnp* deletion on NoR numbers in the optic nerve, not in the CC. Moreover, they reported that NoR numbers in the optic nerve of 1-month-old mice were comparable between *Cnp* homozygous knockouts and controls. Thus, it is unlikely that the innate *Cnp* haploinsufficiency in our study underlies the reduced NoR numbers observed in the CC of *Gtf2i*-KO mice compared to control mice.

**Fig. 7 | In mOLs, Tfii-i binds RE of *Mbp* and *Sox10*. A** Heatmap showing genome-wide Tfii-i binding intensity from ChIP-seq data in mOLs derived from control and *Gtf2i*-KO mice. **B** Genomic distribution of Tfii-i consensus peaks in control mOLs. **C** Motif analysis within a 100 bp window centered on Tfii-i peaks in mOLs reveals enrichment of Olig2 and Sox10 binding motifs. **D** Colocalization analysis of Tfii-i consensus peaks in mOLs shows high overlap with Srf and Olig2 binding. **E** Overlap analysis of Tfii-i peaks with mOL-specific PTMH reveals predominant localization within genomic regions enriched for H3K36me3, a mark associated with active transcription. **F** Tfii-i binding at the *Mbp* genomic locus. The upper track displays Tfii-i binding in mOLs, with consensus peaks highlighted in gray (peaks 75 and 62), TSS is marked in green. The second, third, and fourth tracks show mOL-specific H3K4me3, H3K27ac, and H3K36me3 binding data, respectively. The genome annotation track indicates known *Mbp* enhancers M3 and M5. The bottom track presents mOL-specific predicted cREs and their putative interactions with target

genes, visualized as arcs. **G** Tfii-i binding at *Sox10* genomic locus. The upper track displays Tfii-i binding in mOLs, with consensus peaks highlighted in gray (peaks 44 and 69), the TSS is marked in green. The second, third, and fourth tracks show mOL-specific H3K4me3, H3K27ac, and H3K36me3 binding data, respectively. The genome track indicates known *Sox10* enhancers (U1-U5, D6-D7). The bottom track presents mOL-specific predicted cREs. Tfii-i peaks 44 and 69, located within intronic regions of the *Micall1* gene, align with these cREs and are predicted to physically interact with the *Sox10* promoter (highlighted in blue). **H** 3C-qPCR analysis ($n = 3/4$ for whole cortex and $n = 4$ pooled into a single sample for mOLs, control only). Elevated interaction frequencies were observed between *Sox10* promoter and the regions corresponding to consensus peaks 44 and 69 (highlighted in gray), compared to adjacent regions within the same locus. Interaction frequencies were normalized to those at the *Ercc3* region and further normalized to the 35 Kb data point for each dataset. Source data are provided as a Source Data file.

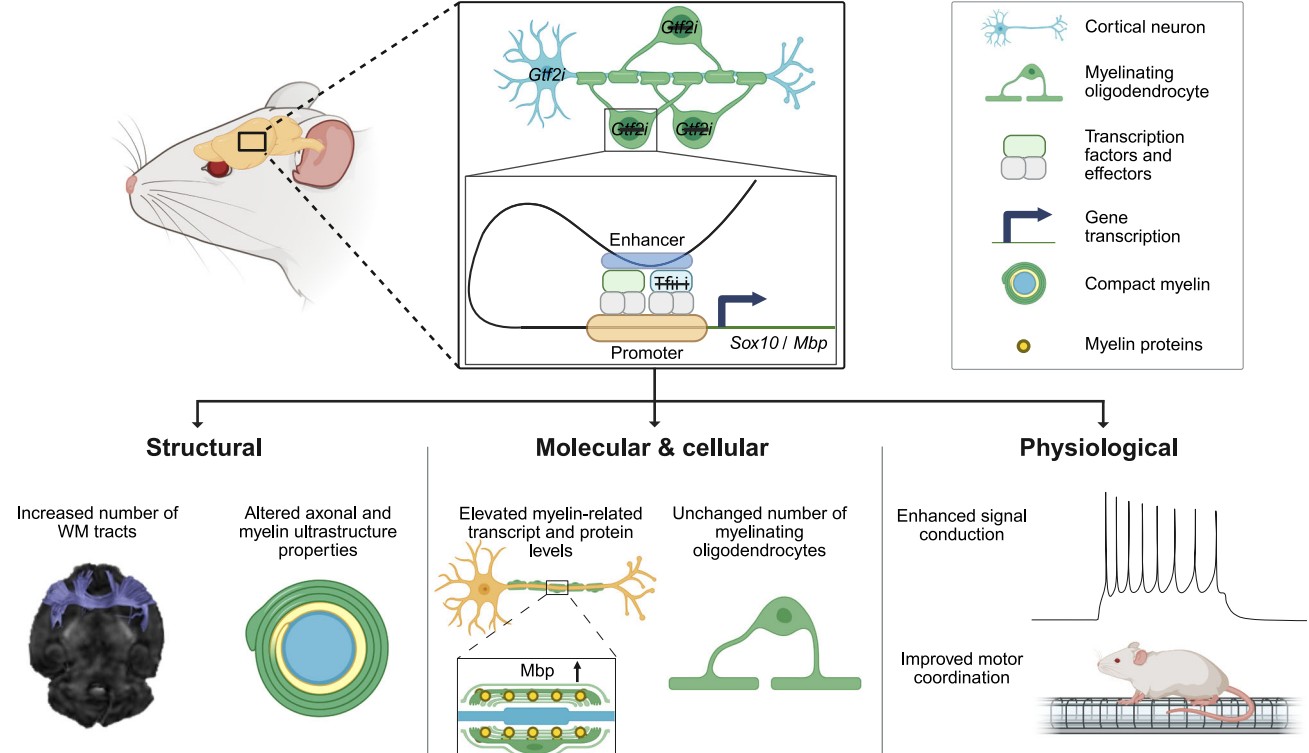

**Fig. 8 | Graphical summary.** In mOLs, Tfii-i binds REs of the key myelination genes, *Sox10* and *Mbp*. Conditional deletion of *Gtf2i* from mOLs alters CNS myelin sheath structure and composition, resulting in enhanced signal conduction and improved motor coordination. Created with BioRender. Rokach, M. (2025) https://BioRender.com/o51w712.

In summary, we identified *Gtf2i*-encoded Tfii-i as a regulator of myelination in mOLs, demonstrating its direct binding, specifically in mOLs, to *Mbp* and *Sox10* regulatory regions. *Gtf2i* deletion from mOLs promoted cellular, molecular, and structural changes, resulting in alterations of myelination in the CNS and PNS. Finally, we provide evidence that Tfii-i's mode of action in mOLs is cell-specific and shed light on its regulatory targets in mOLs.

## Methods
### Mouse handling
*Gtf2i* loxP mice[115] were back-crossed to pure C57Bl/6 J mice (stock no. 000664; Jackson Laboratory) for more than 15 generations. *Cnp-Cr* MGI:3051635, Gift from Dr. Klaus-Armin Nave[114] mice are in a C57Bl/6 background. Mouse lines used in this study are subject to MTA from the original investigator. Each cage contained 2–5 mice regardless of genotype. Mice were housed at a constant 20–24 °C on a 12 h light–dark cycle (lights on at 7:00, lights off at 19:00) with *ad libitum*

access to food and water. Experiments were approved by the institutional animal care and use committee of Tel Aviv University and the Israel Ministry of Health. All efforts were made to minimize animal suffering and the number of animals used.

### Diffusion MRI, fiber tracking, and analysis
Mice were anaesthetized with 5% isoflurane and then maintained anesthetized by continuous isoflurane (1.5%) inhalation throughout the procedure. MRI scanning was performed using a 7T MRI scanner (Bruker, Billerica, MA) with a 30 cm bore and a gradient strength of up to 400 mT/m. The MRI protocol included diffusion imaging acquisition with a diffusion-weighted spin-echo echo-planar imaging pulse sequence. Acquired volumes were 18 slices, each 0.6 mm thick, with the following parameters: resolution of 0.175 mm × 0.175 mm² (matrix size, 128 × 96), repetition time of 3000 ms, echo time of 25 ms, Δ/δ was 10/2.5 ms, 4 echo-planar imaging segments, and 15 non-collinear gradient directions with a single b-value shell at 1000 s/mm² and 3 images

with a *b*-value of 0 s/mm$^2$ (b0 image). The DTI acquisition took 10 min and 48 s. At the end of the experiment, mice were euthanized using $CO_2$.

All diffusion MRI analysis was performed in ExploreDTI[195], and included the following steps:

1. Motion and distortion correction to correct for possible motion- and susceptibility-induced artifacts.
2. Transformation into atlas space via non-linear registration and extraction of atlas space FA and MD per mouse brain.
3. Whole-brain fiber tracking with 0.175 mm × 0.175 mm × 0.175 mm seed voxel resolution; minimal FA and stropping criteria for tracking: FA > 0.1; maximal 30° tracking angle allowed. Tracking step size: 0.175 mm.
4. The reconstructed number of streamlines was taken for statistical analysis between groups and presented as the number of tracts.

## Myelin isolation

Myelin isolation was performed as previously described[145], with minor modifications. Briefly, mice were sacrificed by cervical dislocation, and their cortices were dissected and snap-frozen with liquid nitrogen and stored at −80 °C. Cortices were then pooled (three cortices from three different mice with the same genotype in one tube) and homogenized in 0.32 M sucrose solution supplemented with protease inhibitor (Catalog no. 539131, Sigma-Aldrich). Samples were then layered onto the top of a 0.85 M sucrose solution to create a sucrose step gradient. Following ultra-centrifugation (Optima XPN80, SW41-Ti rotor, Beckman-Coulter), the crude myelin fraction was visible in the interphase of the sucrose step gradient. Two washing steps were performed with filtered DDW, followed by centrifugation on a second sucrose step gradient to obtain the purified myelin fraction.

Purification of myelin for the assessment of transcript (mRNA) levels was performed with RNAse-free solutions.

## Myelin proteome analysis

In-solution digestion of myelin proteins by filter-aided sample preparation (FASP)[145] and LC-MS-analysis by data-independent acquisition mass spectrometry was performed as recently established for mouse CNS myelin[146]. Briefly, protein fractions corresponding to 10 μg myelin protein were dissolved in lysis buffer (1% ASB-14, 7 M urea, 2 M thiourea, 10 mM DTT, 0.1 M Tris pH 8.5) and processed according to a CHAPS-based FASP protocol in centrifugal filter units (30 kDa MWCO, Merck Millipore). After removal of the detergents, protein alkylation with iodoacetamide, and buffer exchange to digestion buffer (50 mM ammonium bicarbonate (ABC), 10 % acetonitrile), proteins were digested overnight at 37 °C with 400 ng trypsin. Tryptic peptides were recovered by centrifugation and extracted with 40 μl of 50 mM ABC and 40 μl of 1% trifluoroacetic acid (TFA), respectively. For quantification according to the TOP3 approach, combined flow-throughs were spiked with 10 fmol/μl of Hi3 E.coli standard (Waters Corporation; contains a set of quantified synthetic peptides derived from E. coli. chaperone protein ClpB) and directly subjected to LC-MS-analysis.

Tryptic peptides were separated by nanoscale reversed-phase UPLC and mass spectrometrically analyzed on a quadrupole time-of-flight mass spectrometer with ion mobility option (Synapt G2-S, Waters Corporation) as recently described in detail[146]. The samples were analyzed in the MSE data acquisition mode without ion mobility separation of peptides to ensure the correct quantification of exceptionally abundant myelin proteins. Continuum LC-MS data were processed using Waters ProteinLynx Global Server and searched against a custom database compiled by adding the sequence information for *E. coli.* chaperone protein ClpB and porcine trypsin to the UniProtKB/Swiss-Prot mouse proteome (release 2023-05, 17137 entries) and by appending the reversed sequence of each entry to enable the determination of false discovery rate (FDR) set to 1% threshold.

For post-identification analysis including TOP3 quantification of proteins and for the detection of significant changes in protein abundance by moderated t-statistics, the freely available software ISOQuant and the Bioconductor R packages "limma" and "q-value" were used as described[146]. Only proteins represented by at least two peptides (minimum length seven amino acids, score ≥5.5, identified in at least two runs) were quantified as parts per million, i.e., the relative amount (w/w) of each protein in respect to the sum over all detected proteins. FDR for both peptides and proteins was set to 1% threshold and at least one unique peptide was required. Proteins identified as contaminants from blood (albumin, hemoglobin) or skin/hair cells (keratins) were removed, and potential outlier proteins were revised by inspecting the quality of peptide identification, quantification and distribution between protein isoforms.

## Western blot

Cortices were dissected as described above and homogenized in solubilization buffer (50 mM HEPES, pH 7.5, 150 mM NaCl, 10% glycerol, 1% Triton X-100, 1 mM EDTA, pH 8, 1 mM EGTA, pH 8, 1.5 mM $MgCl_2$, 200 μM $Na_3VO_4$, and protease inhibitor [Catalog no. 539131, Sigma-Aldrich], diluted 1:100). Protein levels in the homogenized lysate or in the crude or pure myelin fractions were assessed by Quick Start Bradford 1X Dye reagent (Catalog no. 5000205, Bio-Rad) and equal amounts of protein were loaded and resolved by SDS-PAGE through a 4–20% gel. Proteins were electrophoretically transferred to a nitrocellulose membrane in transfer buffer (25 mM Tris-HCl, pH 8.5, 190 mM glycine, and 20% methanol absolute). Membranes were then blocked for 45 min in TBST buffer (0.05 M Tris-HCl, pH 7.5, 0.15 M NaCl, and 0.1% Tween-20) containing 6% skimmed milk at room temperature. Following three consecutive washes with TBST, primary antibodies were added to the membranes for overnight incubation at 4 °C, with gentle shaking. The next day, the membranes underwent three washes of 15 min each in TBST, followed by incubation with secondary antibodies linked to horseradish peroxidase for 45 min at room temperature, with gentle shaking. Following additional three washes in TBST, immunoreactive bands were detected with the SuperSignal West Pico PLUS Chemiluminescent Substrate (Catalog no. 34577, Thermo-Fisher Scientific). Images were collected at set exposure parameters per antibody and amount of protein loaded. Differential antibody/housekeeping protein ratios were calculated using ImageJ (NIH) by an experimenter blind to the genotypes.

Primary antibodies used included rabbit anti-TfII-I (1:1000, catalog no. CST-4562S, Cell Signaling), rat anti-Mbp (1:500, catalog no. MAB386, Merck), rabbit anti-Mog (1:500, catalog no. AB32760, Abcam), rabbit anti-tubulin (1:1000, catalog no. AB108342, Abcam), and rabbit anti-tubulin β4 (1:1000, catalog no. AB179509, Abcam) antibodies. Secondary antibodies included goat anti-rabbit (1:10,000, catalog no. AP132P, Merck), and goat anti-rat (1:10,000, catalog no. AP136P, Merck) antibodies.

## Behavioral tests

Behavioral tests were conducted during the light cycle (07:00–19:00). Before each test, mice were acclimated to the test room for at least 1 h. The temperature was kept at 20–24 °C throughout all behavioral experiments. Lighting properties were verified before each test. Each mouse participated in a maximum of three behavioral tests, with at least 3 day intervals between different tests. The experimenter was blind to genotypes.

**Open field.** Mice were placed in a Plexiglas box (40 cm long × 40 cm wide × 30 cm high) and spontaneous locomotion was recorded for 1 h. Margins were set as the area between each wall and a 10 cm distance into the box. Light parameters were ~60 lux in the center of each arena and ~10 lux at the margins. The placement of the mouse within the box

was detected using Ethovision XT14 tracking software. Semi-manual verification of the identity of each mouse was performed.

**Rotarod.** Mice were placed on a rod (Catalog no. 47650, Ugo Basile, Gemonio, Italy) rotating at a constant speed of 4 revolutions per minute (rpm). Following verification of correct body orientation and placement on the rod, a constant acceleration was set up to 40 rpm, with a 5-min limit. To minimize grasping of the rod which leads to mice hanging, rather than walking on the rod, we used 3D printed adapters to increase the diameter of the rod to 30 cm instead of 15 cm. Mice that either gripped the rod for 2 or more full rotations, or fell from the rod before 20 s passed from the beginning of each trial were disqualified and their performance was not included in the analysis. Each mouse participated in 3 trials on the same day, with 1 h of an inter-trial interval between trials. Mice were trained once on the machine a day before beginning the actual test.

**Elevated zero maze.** Mice were placed in the closed arm of the elevated maze (height 60 cm) and their locomotor behavior was recorded for 15 min. Light parameters at the closed arms were -10 lux. The placement of the mouse within the maze was detected using Ethovision XT14 tracking software and the time spent in the open arms was calculated. Data are presented in time bins of 5 min and overall time spent in open arms.

**Three chambers social interaction test.** The social interaction test apparatus was made of a clear Plexiglas box (65 cm long × 44 cm wide × 30 cm high) with partitions dividing the box into left, center and right chambers. These three chambers were interconnected, with 5 cm openings between each chamber that could be opened or closed manually by the experimenter with a lever-operated door. The inverted wire cups containing the inanimate object or stranger mice were cylindrical, 10 cm high, with a bottom diameter of 10 cm and with metal bars spaced 0.8 cm apart. A weighted cup was placed on top of the inverted wire cup to prevent the test mice from climbing onto the wire cup. Each wire cup was thoroughly cleaned after each use, with water and 70% ethanol. The arena itself was cleaned after each trial, using 70% ethanol.

WT male C57Bl/6J mice with similar ages and body weights were used as stimulus mice. The habituation of stimulus mice was performed in proximity to the test for 3 consecutive days, 2 sessions a day, 30 min each session, in which mice were placed inside an inverted wire cup. These mice were used as stimulus mice for a maximum of two non-consecutive trials each day, and no more than 10 trials overall.

The test consisted of two distinct phases. Each phase lasted 15 min, and at the end of each phase the test mouse was gently placed in the center arena while preparation of other sections of the arena took place. For the habituation phase, the test mouse was gently introduced to the center section of the arena. Then, the experimenter opened both doors leading to the left and right sections of the arena simultaneously, and the test mouse was allowed to explore the entire arena. During the social interaction phase, an inanimate object and stranger stimulus mouse were placed on opposing sections of the arena, under an inverted wire cup. The location of the object and stranger mice changed and was randomized for each trial. The experimenter then opened both doors simultaneously and the test mouse was allowed to explore and interact with both the inanimate object and the stranger mouse.

Using EthoVision XT 14.0.1326 software (Noldus Information Technology BV, Wageningen, The Netherlands), behavior and interaction time of the test mouse was assessed in the sociability phase, and was then analyzed by the same experimenter, blinded to the genotype. The social index was determined in the sociability phase, by dividing the time spent in interaction with the stranger mouse by the time spent in interaction with the inanimate object.

## RNA extraction, cDNA preparation, and qPCR

Cortices were dissected as described above and placed in RNA-later solution (catalog no. AB-AM7020, Invitrogen) at 4 °C overnight. The next day, RNAlater was disposed of and samples were frozen at −80 °C for no longer than 1 month, until the RNA extraction step. RNA extraction, cDNA preparation, and quantitative PCR were performed as previously described[196]. Briefly, cortices were homogenized in TRIzol reagent (Thermo-Fisher Scientific) and homogenized in TissueLyser 2 (Qiagen). Following complete homogenization of the tissue, additional TRIzol was added, and samples were incubated for 5 min at room temperature. Chloroform was added to the samples, which were then manually shaken and incubated for an additional 3 min at room temperature. Samples were then centrifuged to create three different layers, the top one containing RNA, which was transferred to a new tube and diluted 1:1 with isopropanol. The tubes were then manually shaken and incubated at room temperature for 5 min followed by another round of centrifugation and 2 wash steps with 80% ethanol. Samples were left to dry thoroughly before resuspending them with DEPC-treated water and measuring total RNA concentration using NanoDrop One (Thermo-Fisher Scientific). RNA was then diluted to a working concentration of 100 or 20 ng/µl, and reverse transcription was performed using random primers and a High-Capacity cDNA Reverse Transcription Kit (Thermo-Fisher Scientific).

Quantitative PCR was performed using the Fast SYBR Green Master Mix (Thermo-Fisher Scientific) and the CFX Connect Real-Time PCR Detection System (Bio-Rad). mRNA levels were measured and calculated by the comparative cycle threshold method[197]. *Gapdh* levels were used to normalize each sample. The final results are calculated as fold change relative to the control group. Primers sequences used were *Gapdh:* F' – GCCTTCCGTGTTCCTACC, R' – CCTCAGTGTAGCCCAAG ATG, *Mbp:* F' - CCTCAGTGTAGCCCAAGATG, R' – TGTCTCTTCCTCCCC AGCTAAA, *Sox10:* F' - ACAGCAGCAGGAAGGCTTCT, R' – TGTCCTCA GTGCGTCCTTAG, *Mog:* F' – AGCTGCTTCCTCTCCCTTCTC, R' – ACTA AAGCCCGGATGGGATAC, and *Plp1:* F' - TTGTTTGGGAAAATGGCTAG GACA, R'- GGTCCAGGTATTGAAGTAAATGTAC.

## Immunohistochemistry

A basic immunohistochemical procedure was used as described previously[198]. Briefly, isoflurane was used to deeply anesthetize mice, which were then transcardially perfused with 15 mL PBS and 15 mL freshly made 4% paraformaldehyde (PFA) in PBS. The mice were decapitated and their brains dissected and kept in 4% PFA solution at 4 °C overnight. Brain slices of 100 µm thickness were cut using a vibratome (Leica). The slices were washed three times with PBS for 5 min each time, followed by permeabilization with 1.2% Triton X-100 in PBS for 15 min at room temperature, and an additional 3 washes (5 min each). Slices were then put in a blocking solution containing 5% normal goat serum, 2% bovine serum albumin, and 0.2% Triton X-100 in PBS for 1 h at room temperature. Primary antibodies were diluted in blocking solution added to clean wells in a 96-well plate containing brain slices for overnight incubation at 4 °C. The next day, following three washes in PBS for 15 min each, secondary antibodies conjugated to Alexa Fluor 488, 555, or 647 in blocking solution were added for 1 h at room temperature. The slices were then washed three times with PBS for 15 min each, and mounted on pre-coated glass slides (Bar-Naor) using VECTASHIELD Hardset Antifade Mounting Medium (Vector Laboratories). Images were captured using a light microscope (Olympus IX83).

Primary cell cultures (OPC and mOL cultures) were stained using a similar method with minor modifications. Here, permeabilization was performed as part of the blocking step, with 0.3% of Triton X-100 being added. Incubation with primary antibodies was for 1 h at room temperature.

For co-localization experiments, images of the motor cortex region were collected by an experimenter blind to the genotypes at ×10 magnification (bregma 0.5 mm). Both hemispheres were separately imaged, and the average number of cells was calculated for each mouse. For co-localization experiments at the midline of the CC, images were taken at ×20 magnification from the same brain bregma. Images of OPC and mOL cultures and supplementary images of Tfii-i localization in various cell types were taken at ×60 magnification. Quantification analysis of immunohistochemical experiments was performed by an experimenter blinded to the genotypes using ImageJ (NIH).

For NoR experiments, images of the midline CC were collected by an experimenter blind to the genotypes at x60 magnification (same bregma as the CC TEM experiment, Supplementary Fig. 3), with 10 z-stacks at 0.1 μm step (1 μ overall), using a BC43 CF confocal microscope (Oxford Instruments). For NoR analysis, images were restacked using the z project tool in ImageJ (NIH) at "max intensity", and NoR length was calculated as the distance between two Caspr stains, only when NaV 1.6 stain is perfectly visible between them, meaning only when the node was angled towards the image and the myelin segments could be identified as the flanking segments of the same node. Quantification analysis of immunohistochemical experiments was performed by an experimenter blinded to the genotypes.

CTCF analysis in mOL-enriched primary cell culture was performed by an experimenter blinded to the genotypes. The experimenter manually marked the region of interest (ROI) for analysis. For Sox10 and Tfii-i that is the nucleus, for Mbp that is either the soma or the whole cell. Following this selection of ROI, intensity of the ROI was measured using ImageJ (NIH). To calculate specific background intensity of each image, 3 random areas with identical dimensions (outside of the ROI) were selected and their intensity measured and averaged. Then, this formula was used:

$$CTCF = Total\ intensity\ of\ the\ ROI$$
$$- (Mean\ intensity\ of\ the\ background*ROI\ area)$$

For each slide, the average CTCF of control cells was calculated and used to normalize the calculated CTCF of all cells from this slide. Hence, control values are all 1 and iCre values represent the fold change.

Commercial primary antibodies used in this study: rabbit anti-TFII-I (1:1000, catalog no. CST-4562S, Cell signaling), mouse anti-APC (Ab-7, CC1) (1:500, catalog no. OP80, Calbiochem), rabbit anti-Olig2 (1:1000, catalog no. AB9610, Sigma-Aldrich), rat anti-Pdgfrα (CD140a) (1:700, catalog no. 14-1401-82, Invitrogen), mouse anti-NeuN (1:1000, catalog no. MAB-377, Sigma-Aldrich), mouse anti-Sox10 (1:500, catalog no. SC-365692, Santa Cruz), rat anti-Mbp (1:500, catalog no. MAB386, Sigma-Aldrich), chicken anti-mCherry (1:1000, catalog no. ab205402, Abcam), rabbit anti-NaV 1.6 (1:100, catalog no. ASC-009, Alomone labs), and mouse anti-Caspr (1:50, catalog no. ab252535, Abcam). Secondary antibodies used in this study: goat anti-rabbit (1:1000, catalog no. ab150077, Abcam), goat anti-mouse (1:1000, catalog no. A11001, Invitrogen), goat anti-rat (1:1000, catalog no. ab150165, Abcam) conjugated to Alexa Flour 488. Goat anti-rabbit (1:1000, catalog no. A32732, Invitrogen), goat anti-chicken (1:1000, catalog no. A32932, Invitrogen), goat anti-mouse (1:1000, catalog no. A21424, Invitrogen) conjugated to Alexa Flour 555. Goat anti-rabbit (1:1000, catalog no. A21245, Invitrogen), goat anti-rat (1:1000, catalog no. A21247, Invitrogen) conjugated to Alexa Flour 647.

### In-vivo fEPSP recordings in the CC
Recordings were performed as previously described[20]. Briefly, 1 month-old mice were anaesthetized with 5% isoflurane by volume for induction, head fixed to a stereotaxic apparatus (David Kopf instruments) and then maintained anesthetized by continuous isoflurane (1.5%) inhalation throughout the procedure. Body temperature was recorded and maintained by a heating pad (FHC, DC temperature controller) at 34 °C throughout the surgery. Small diameter holes were drilled in the skull at the position of the recording and stimulating electrodes (bipolar stainless steel; 0.1397 mm diameter with coating, 0.0762 mm bare diameter). The stimulating electrode was inserted into the CC (left hemisphere) at a set position (1.5 mm posterior to bregma, 1 mm lateral from midline, 1000–1300 μm beneath the cortical surface) and the recording electrode was placed on the same coordinates, on the right hemisphere. Ground electrode was screwed to the skull above the cerebellum. The position of both recording and stimulating electrodes was adjusted to acquire an optimal signal when electrical stimulation (100–150 μA, 0.2 ms pulse width, 1 Hz) was delivered. After reaching optimal signal and position, fEPSP (150 μA, 0.2 ms pulse width, 0.2 Hz) was acquired and quantified for 15–30 min to verify signal stability, and only then the fEPSP recordings used in this study were acquired. Following recording acquisition, a lesion was made at the tip of each electrode by a 10 μA constant current delivered for 10 s. The correct locations of the electrodes were then verified in 50 μm-thick brain slices. The experimenter was blinded to genotypes. Response latency was calculated as the time between stimulation onset and the onset of the evoked field response.

### Electron microscopy
Mice were trans-cardially perfused as described above with minor alterations. The fixation solution included 2% PFA, 2.5% glutaraldehyde, and 0.1 M sodium cacodylate buffer (pH 7.4). Brains were dissected and cut at the midline in sagittal fashion. Then, -1 mm cubes from the CC at the level of the fornix (Supplementary Fig. 3) were extracted by an experiment blinded to the genotypes and incubated overnight in the fixation solution. Following CC extraction, the SN was exposed and incubated overnight in the fixation solution as well. Samples were washed and transferred into 0.1 M sodium cacodylate solution and held at 4 °C for up to 3 weeks. Preparation of ultrathin sections was performed by the electron microscopy unit at Tel Aviv University and the Peles lab at the Weizmann Institute of Science. A transmission electron microscope (TEM; JEM-1400Plus) was used to obtain images at either ×5k, ×12k, or ×30k magnification from different regions of each section. g-ratio quantification, myelin thickness, and myelin deformation analysis (SN only) was performed by an experiment blinded to the genotype. g-ratio values were calculated by manually measuring the ratio of the axon diameter and the myelinated axon diameter by an experimenter blinded to genotype.

### Primary OPC and mOL cultures
Primary mixed glia cell cultures were prepared from the cortices of P0-P2 control ($Gtf2i^{f/f}$) or Gtf2i-KO mice, seeded on PDL-coated flasks, and maintained for 9–10 days in DMEM/F12 (Sigma-Aldrich, catalog no. D6421) supplemented with L-glutamine (Sigma-Aldrich, catalog no. G7513), 10% fetal bovine serum, and penicillin-streptomycin. On DIV7, 5 μg/mL of insulin (Sigma-Aldrich, catalog no. I0516) was added to the medium. On DIV 9–10 days, OPCs were isolated by manual shaking, seeded on PDL-coated 12 mm coverslips placed in 24-well plates at a density of 60–80k cells per well, and maintained in DMEM/F12 supplemented with L-glutamine, penicillin-streptomycin and SM1 supplement (STEMCELL technologies, catalog no. 05711). To increase OPC differentiation into mOLs, T3 (Sigma-Aldrich, catalog no. T5516) was added to the medium at a final concentration of 340 ng/mL. AAV expressing either mCherry or iCre-mCherry under the CBAP promoter was added on DIV3 (following OPC isolation) for 48 h, followed by a complete change of medium. Five days later, on DIV 9–10 (following OPC isolation) cells were fixed for 10 min at room temperature using 4% PFA followed by staining as detailed above. To assess OPC enrichment, cells were fixed on DIV2 following OPC isolation and were not supplemented with T3.

## ChIP-seq and ChIP-qPCR

For the ChIP-seq experiment, primary OPCs were subjected to differentiation as previously described. Differentiated mOLs from 4 to 5 mice were pooled by genotype to ensure enough starting material for a ChIP-seq procedure. At DIV8−9 cells were washed gently with PBS to remove cellular debris then harvested and pooled using Trypsin A (Sigma-Aldrich, catalog no. T4049). Trypsin activity was neutralized by adding DMEM/F12 supplemented with L-glutamine and 20% FBS. Cells were then centrifuged at $100 \times g$ for 5 min. For the ChIP-qPCR experiment, the cortices of P30 mice were dissected as described above. The cortices were then added to a tube and manually digested with a plastic probe in PBS. Next steps are identical for both the ChIP-seq and ChIP-qPCR procedures. The pellets (or tissue) were then crosslinked in 1% PFA for 10 min at room temperature. To avoid over-fixation, 100 μl of glycine (Cell Signaling, catalog no. 7005S) was added after 10 min. The lysates were then centrifuged at $2655 \times g$ for 5 min, and the supernatant was carefully removed. Three washing steps with PBS and centrifugation with the same parameters were then performed. Fresh sonication buffer (400 μl) containing 1% SDS, 50 mM Tris-HCl, pH 8.8, 20 mM EDTA, and a protease inhibitor in DEPC was then added to each sample. DNA shearing was performed by nine rounds of sonication (Covaris) of 10 pulses each to obtain 200−800 bp DNA fragments. Then, 100 μl of sheared chromatin was added to 400 μl ChIP buffer supplemented with protease inhibitor, and 5 μl of rabbit anti-TFII-I antibodies (catalog no. CST-4562S, Cell signaling) were added for overnight incubation at 4 °C, with gentle shaking. For the ChIP-qPCR experiment, 1 μl of rabbit anti-IgG antibodies (catalog no. 3900S, Cell Signaling) was also added to different tubes to create the IgG condition. Additionally, for both the ChIP-seq and ChIP-qPCR experiments, 5 μl of sheared chromatin was taken as an input control and kept at −20 °C until the elution step. On the second day, 20 μl of Magna ChIP Protein A + G Magnetic Beads (Millipore catalog no. 16−663) were added to each tube for overnight incubation at 4 °C, with gentle shaking. On the third day, the ChIP buffer was removed using a magnetic stand (catalog no. S1506S, NEB) and three washes were performed with three different solutions−low salt solution (150 mM NaCl, 20 mM Tris-HCl, pH 8, 1% SDS, 500 μl Triton X-100, 1 mM EDTA, in DEPC), high salt solution (500 mM NaCl, 20 mM Tris-HCl, pH 8, 1% SDS, 500 μl Triton X-100, 2 mM EDTA, in DEPC) and TE solution (10 mM Tris-HCl, pH 8, 1 mM EDTA, in DEPC). Next, 100 μl of elution buffer (9 mL NaHCO$_3$ (100 mM) with 1 mL 10% SDS) supplemented with 1 μl of RNAse A was added to each sample (including input control samples). Samples were incubated at 37 °C for 20 min at 300 rpm. Next, 4 μl of 5 M NaCl and 1 μl of Proteinase K were added to each sample, which was then incubated at 62 °C, 600 rpm for 4 h. The samples were next heated at 95 °C for 10 min. Then, the beads were removed using the magnetic stand, and each sample underwent DNA cleaning and extraction using a DNA purification kit (catalog no. 14209, Cell Signaling). DNA was kept at −20 °C until library preparation. Library preparation and paired-end sequencing using Illumina NovaSeq S1 (100 cycles) was performed at the Crown Genomics institute Weizmann Institute of Science. For the ChIP-qPCR experiment, DNA was run as previously described in the qPCR section and normalized to input and IgG binding levels. Primer sequences are available in Supplementary Data 3.

## ChIP-seq data analysis

Alignment of the raw fastq reads to the GRCm39 genome was performed using Bowtie2. Differential peak calling was performed using Macs2 (version 2.2.9.1) with the command line "callpeak -t IP.bam -c Input.bam -g mm -f BAMPE -q 0.1 --nolambda --keep-dup all −B", where immunoprecipitate (IP) over input scores were calculated. Additional manual filtration was performed to exclude peaks with; $q$-value > 0.01, fragment size larger than 600 bp and peaks that were mapped to mitochondrial DNA. The Bedtools package (using the intersect and cluster tools) was employed to identify Tfii-i consensus peaks, defined as those present in at least three out of five samples in the control group. Peaks were classified as "lost" if no overlap was observed between the KO group and the Tfii-i consensus peaks. Respective heatmaps were generated by custom R scripts and the Complexheatmap R package. HOMER package[199] was used for peaks annotation, motif enrichment, and colocalization analyses. Motif analysis was performed using HOMER with a 100 bp and 20 bp window. Genomic tracks were generated in IGV[200]. Evolutionary conservation ratios between murine and human genomes were assessed using the ECR browser tool[201]. Tfii-i binding data in mOLs was lifted over to the mm10 genome (from GRCm39) using UCSC LiftOver tool[202].

## Chromosomal conformation capture assay followed by qPCR (3C-qPCR)

3C-qPCR was performed as previously described[155]. Briefly, mOL-enriched samples were acquired from primary differentiated OPC cultures as detailed above. Cells were subjected to fixation in 1% PFA (Sigma) for 10 min, followed by quenching with glycine. Lysis was then performed using nuclear buffer consisting of 0.5% Triton X-100 (Sigma), 0.1 M sucrose (Sigma), 5 Mm MgCl (Sigma), 1 mM EDTA (Sigma), 10 mM Tris pH 8 (Sigma). Following 20 min on ice, samples were subjected to digestion using the DpnII enzyme (New England Bio Labs) for 24 h at 37 °C at 1400 rpm. Inactivation of the enzyme was then performed for 20 min at 65 °C. Samples were then ligated using T4 ligase (Invitrogen) and decrossed with proteinase K (Invitrogen) overnight. DNA was extracted from the samples with DNA-purification buffers and spin columns (Cell Signaling Technology). For whole cortex samples, digestion and ligation efficiencies, as well as sample purity and concentration, were assessed and validated as previously described[169]. Due to limited yield and concentration in the mOL-derived samples, four control samples were pooled to generate a single representative 3 C library. Following qPCR amplification, products were resolved on a 2% agarose gel, and interaction frequencies were quantified based on band intensity at the expected product size, as determined by primer design, using Bio-Rad Image Lab software. Each sample was run in duplicates. Primer sequences are available in Supplementary Data 3.

## Statistical analysis

All data were collected from *Gtf2i*-KO and littermate control mice by an experimenter blinded to the genotypes. Datasets were analyzed for normality by D'Agostino−Pearson or Shapiro−Wilk tests. Datasets with normal distribution were analyzed by either unpaired two-tailed student's *t* test or one/two-way ANOVA with proper post-hoc analyses (Prism, GraphPad Software). Datasets with non-normal distributions were analyzed by a Wilcoxon signed-rank test or Kolmogorov-Smirnov test. Sample sizes are similar to those reported in previous publications[13]. Graphs were created using Prism GraphPad 10.5.0. All results are presented as means ± S.E.M.

## Reporting summary

Further information on research design is available in the Nature Portfolio Reporting Summary linked to this article.

## Data availability

The mass spectrometry proteomics data have been deposited to the ProteomeXchange Consortium via the PRIDE[203] partner repository with the dataset identifier PXD054341. The ChIP-seq data have been deposited in the GEO accession database, under the accession identifier GSE285541. Source data are provided with this paper.

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

## Acknowledgements

The authors gratefully acknowledge past and current members of the Barak laboratory and the Peles laboratory for their support and insightful comments on the manuscript. The authors would also like to thank Gonçalo Castelo-Branco for his insights and consultation during the writing of this manuscript, and Ramona Jung for technical support. AAVs used in the study were a generous gift from Prof. Inna Slutsky's laboratory. Special thanks to Naama Harari Uzan for her assistance in preparing the figures. This work was supported by grants awarded to B.B. from the Fritz Thyssen Stiftung, the National Institute of Psychobiology in Israel, and the Israel Science Foundation (grant 2305/20). I.S. was supported by the European Research Council (grant 101097788). E.P. was supported by the Israel Science Foundation and the Sagol Center for Research on the Aging Brain. G.L. was supported by the Eshkol Fellowship from The Ministry of Science and Technology, Israel. B.B. was the recipient of The Alon Fellowship for outstanding young researchers awarded by the Israeli Council for Higher Education.

## Author contributions

G.L., E.P., N.S., and B.B. conceptualized the study, designed the experiments and wrote the manuscript. G.L. and M.R. collected, analyzed and interpreted the results of the behavioral, developmental and immunofluorescence experiments. O.J, S.B.S., and H.B.W. analyzed and interpreted the results of the myelin proteome analysis. M.R, M.G, I.I.E., and G.L collected, analyzed and interpreted the results of the qPCR experiments. G.L., M.R., S.S.T., and G.E.S. collected, analyzed and interpreted the results of the Western blot experiments, and performed myelin isolation. I.F. and Y.A. collected, analyzed and interpreted the results from the MRI-DTI experiment. O.K.F., Y.E.E, A.V., and G.L. collected, analyzed and interpreted the results of the electron microscopy experiment. S.S. and I.S. collected, analyzed and interpreted the results from electrophysiological studies from the corpus callosum. G.L., M.R., T.R., J.B., H.P., and A.M. collected, analyzed and interpreted the results of the ChIP-seq and the 3C-qPCR experiments. G.L., M.R., Y.E.E., and E.B. collected, analyzed and interpreted the results from the mOL-enriched primary cell culture experiments. All authors have read and agreed to the published version of the manuscript.

## Competing interests

The authors declare no competing interests.

## Additional information

[1]Sagol School of Neuroscience, Tel-Aviv University, Tel-Aviv, Israel. [2]Gray Faculty of Medical and Health Sciences, Tel-Aviv University, Tel-Aviv, Israel. [3]Faculty of Social Sciences, School of Psychological Sciences, Tel-Aviv University, Tel-Aviv, Israel. [4]Faculty of Life Sciences, School of Neurobiology, Biochemistry & Biophysics, Tel-Aviv University, Tel-Aviv, Israel. [5]The Robert H. Smith Faculty of Agriculture, Food and Environment, The Hebrew University of Jerusalem, Rehovot, Israel. [6]Departments of Molecular Cell Biology and Molecular Neuroscience, The Weizmann Institute of Science, Rehovot, Israel. [7]Neuroproteomics Group, Department of Molecular Neurobiology, Max Planck Institute for Multidisciplinary Sciences, Göttingen, Germany. [8]Translational Neuroproteomics Group, Department of Psychiatry and Psychotherapy, University Medical Center Göttingen, Georg-August-University, Göttingen, Germany. [9]Department of Neurogenetics, Max Planck Institute for Multidisciplinary Sciences, Göttingen, Germany. [10]Faculty of Biology and Psychology, University of Göttingen, Göttingen, Germany. [11]Sieratzki Institute for Advances in Neuroscience, Tel-Aviv University, Tel-Aviv, Israel. ✉e-mail: boazba@tauex.tau.ac.il

