## [Transparent Peer Review file · Nature Communications]

Gtf2i-encoded transcription factor Tfii-i regulates myelination via Sox10 and Mbp regulatory elements

Corresponding Author: Professor Boaz Barak

Version 0:

Reviewer comments:

Reviewer #1

(Remarks to the Author)

Levy and colleagues analyze in their study the role of the general transcription factor Gtf2i/Tfii-I in differentiating oligodendrocytes and myelination. They show alterations in the number of white matter fibers in the brain of adolescent mice and in the myelination pattern that correlates with increased conduction properties in the corpus callosum and altered behavior in tests for motor coordination and anxiety-like behavior. They furthermore present evidence in support of a model where Gtf2i exerts its effects in oligodendrocytes by binding to regulatory regions of Sox10 and Mbp and repressing its expression, presumably by Suz12/PRC2 recruitment and increased H3K27 trimethylation. The study follows a previous analysis on the consequences of Gtf2i deletion in neurons and derives part of its relevance from the fact Gtf2i is within a chromosomal region whose deletion or duplication leads to severe neurodevelopmental syndromes such as Williams syndrome.

While it is interesting in principle to unravel the cause of neurodevelopmental syndromes and the molecular mechanism, by which a component of the general transcription machinery can specifically interfere with processes in myelinating oligodendrocytes, my enthusiasm for the present study is not particularly high because of several limitations, shortcomings, exaggerations and its preliminary state. Specifically:

It is a general limitation that the authors chose to study Gtf2i loss in oligodendrocytes using the Cnp-Cre driver. Cnp-Cre deletes very late in oligodendrocyte progenitor cells (OPCs). As a consequence, the study will only be able to detect Gtf2i effects on late events of oligodendrocyte development. Any earlier effects remain unexplored. This, however, may be highly relevant if the idea of the study is to learn about neurodevelopmental disorders such as Williams syndrome.

The authors do not provide quantifications for the efficiency of Gtf2i deletion (Fig. 1). Instead they show representative pictures with one or two oligodendrocytes. Quantification is definitely needed and pictures should show at least 10-20 oligodendrocytes.

The authors argue that their Gtf2i mutants exhibit hypermyelination. At the same time their data show that “g-ratio values in the overall axonal population of the CC were not significantly different” (p.7, line 154). This rather argues against a simple hypermyelination. Instead, the authors rather observe higher axon diameters and a lesser impact of axon diameters on g-ratio. This observation may actually indicate that the alterations in the Gtf2i mutants is due to an altered interaction of the myelin sheath with the axon and not so much due to alterations in the myelination process per se.

As the authors are surely aware, the Cnp-Cre driver is also active early in some neural crest cells and in myelinating Schwann cells. Therefore, it cannot be excluded that part of the behavioral alterations are caused by peripheral effects instead of changed oligodendrocyte/myelin properties. This is particularly relevant for motor behavior.

The authors argue for increased anxiety-like behavior in their mouse mutants on the basis of open-field exploration tests. While the obtained results are consistent with this assumption, it would probably need additional tests for confirmation.

The evidence for the proposed molecular mechanism is very thin. There is no general approach to map the Gtf2i-induced changes by RNA-seq, ChIP-seq or similar methods. Instead, previously published ChIP-seq data are taken as justification to focus on Sox10 and Mbp. However, most of these ChIP-seq data are not from oligodendrocytes but from other cells. Considering that Gtf2i is present in most, if not all cells, but differentially recruited to promoters that are active in the

respective cell at the particular time, Gtf2i occupancy in neural precursor cells etc. is not very relevant for oligodendrocytes. As is, the focus on Sox10 and Mbp appears arbitrary.

The proposed mode of action of Gtf2i on Sox10 and Mbp expression is highly speculative, and probably not too likely. There is little experimental evidence provided for the proposed PRC2 recruitment. PRC2 recruitment and H3K27 trimethylation are predominantly used for switching genes off rather than reducing expression levels gradually, and might therefore not be fitting as a mechanism in the presented case. The little evidence provided in the paper furthermore comes from ChIP-PCR experiments (Fig. 7D,E) that are not too convincing because effects are fairly moderate and experiments were performed with chromatin from whole cortex (and thus from a mix of many different cells) and not from oligodendrocytes. By the way, what are the “targeted sites” (p.12, line 244) studied in ChIP-PCR experiments? Are these the respective gene promoters?

From the flow of the paper, Fig. 8 is meant to support the proposed molecular mode of Gtf2i interaction. However, all it does is to show that manipulations of Gtf2i levels in oligodendrocytes also change Sox10 and Mbp levels. This is furthermore done by measuring intensities of immunofluorescently stained cells, which is probably not a very robust assay. What the experiment does not show is whether effects are direct or indirect, and whether the proposed mechanism holds. No rescue has been performed.

Given the fairly mild effects of Gtf2i deletion on oligodendrocyte differentiation and myelination, it is unclear to me why one would state that “we identified Tfii-i as a novel master regulator of the myelin transcriptome” (p.17, line 285). In the same vein, I have a tough time to see the “important basis for a potential new therapeutic target for WM disorders and conditions in which CNS myelination is required” (p.19, line 348). To me it is not even obvious what I learn from the study regarding 7q11.23 microdeletion/duplication syndromes.

Reviewer #2

(Remarks to the Author)

The authors report the interesting observation that a general transcription factor serves as an endogenous “repressor” of myelination in that the loss of function mutation in oligodendrocytes causes a hypermyelination phenotype, but without changing glial cell numbers. This is associated with the activation of a known promyelination transcription factor (SOX10) and a critical protein for myelin assembly (MBP), which reportedly improves functional performance in myelination dependent tasks. The latter is quite surprising given that wildtype myelination (g-ratio) is “optimal” and that other hypermyelinating mice have not shown increased nerve conduction velocities (which is different from improved motor performance in mice with more oligodendrocytes, e.g. Gibson et al. 2014).

Overall, this is an interesting study. Some analyses appear still a little superficial, but that does not diminish the importance of these findings in the field. The following points should be revised/considered:

1) Line 80-92: The relationship between gene and protein names GTF21, Gtf2i, GTFII-I, Tfii-i is very confusing for the reader. It took me an internet search to realize who is who. Please use the same name throughout. Also in the introduction, gain and loss of function effects determined in previous publications should be better explained and contrasted, also within the present paper, if necessary with a supplementary small table.

2) Line 90: please add to the introduction (and reference) prior work that showed this TF to be widely expressed and to engage with the transcriptional core complex. That helps understanding Fig 1B. Does the TF have similar functions in other cells and how specific is the effect here to myelin genes?

3) Line 101 – and Fig 2: what means “increased number of WM fibers”, more axons or higher percentage of myelinated axons? DTI may not be the right tool to measure myelination in mice, in which any histological approach is superior to MRI. At least it must be complemented with histological analyses.

4) Fig1A (genotyping) can go into supplementary data, and Fig 1C and D are not really helpful. A quantitation of the observation in Fig1B would be helpful.

5) Fig 3: the statistics of the g ratio measurement is not clear. How many mice were used and what is the N number? I have the impression the authors used the number of myelinated axons rather than the number of mice.

6) Lines 285 and 342: I find the term “master regulator” too strong for a protein that is not specifically involved in myelination (compare e.g. to the Akt-mTOR pathway which also hardly a master regulator of myelination). Likewise, Fig 9 is not reflecting the biology of a widely expressed transcription factor.

7) Fig 5: I could not find any information as to the number of mice used in the behavioral tests. What age was used?

8) Is the myelination of the peripheral nervous system affected? CnpCre should target the gene also in Schwann cells.

Minor points:

Line 129: what means “proper development”?

Line 145: 1 month old mice?

Line 150: note that g ratios are not used to describe “structural integrity” of myelin, merely myelin sheath thickness.

Line 301: what is the relevance of “increased cell surface area”? A surrogate marker for myelination? That would be misleading.

Reviewer #3

(Remarks to the Author)

The present manuscript describes a cell-autonomous role for the transcription factor *Gtf2i* in oligodendrocytes. Deletion of *Gtf2i* from oligodendrocyte lineage cells, using *CNP*Cre, resulted in hypermyelination in the absence of changes in oligodendrocyte density, increased functional connectivity, and increased axon diameter in the corpus callosum. These physiological changes were accompanied by changes in motor coordination and anxiety-like behavior. Mechanistically, the authors found that *Gtf2i* negatively regulates expression levels of *Sox10* and *MBP*. Together, these data support the idea that *Gtf2i* cell-autonomously regulates the extent of myelination in oligodendrocytes.

Overall, this study is well-designed and in addition to establishing *Gtf2i* as a novel negative regulator of myelination, there are several exciting observations reported here. In particular, the hypermyelination without increases in oligodendrocyte cell density, the increase in axon density resulting from an oligodendrocyte-specific manipulation, and the increase in functional connectivity are noteworthy. One major question that remains is whether these phenotypes are the result of accelerated myelination, or whether the hypermyelination persists into adulthood. This question could be addressed with the addition of a later time point (>3mo) in a couple of key quantifications. Ideally, the authors would also quantify oligodendrocyte differentiation at an earlier timepoint and in vitro to confirm the theory that *Gtf2i* exclusively regulates myelination and not differentiation.

Minor comment - the authors should quantify whether the percentage of myelinated axons is altered in the CC of control and *Gtf2i*-KO mice.

Version 1:

Reviewer comments:

Reviewer #1

(Remarks to the Author)

In their manuscript, Levy and colleagues analyze the *Gtf2i*-encoded transcription factor *Tfii-i* in oligodendrocyte differentiation and identify a role in myelination, presumably via its impact on *Sox10* and *Mbp* expression. Despite the fact that substantial amounts of data were added during the revision, my enthusiasm for the study remains limited. To mention a few aspects:

While I do not doubt the presence of myelin abnormalities in the *Gtf2i* knockouts, the myelination phenotype in the corpus callosum is very subtle.

According to the authors, the g-ratio is unaltered but there are more axons with larger diameter and as a consequence with thicker myelin. From my experience, axons with different diameter are not equally distributed throughout the corpus callosum so that the phenotype described by the authors could be a sampling artifact. From the information provided in the Materials and Methods section, I cannot judge whether this can be excluded.

Additionally, control mice and conditional *Gtf2i* knockout mice differ by the presence/absence of the *Cnp*-Cre. Considering that Cre is knocked into the *Cnp* locus and disables *Cnp* expression from one allele, the *Gtf2i* knockouts express lower levels of the myelin protein *Cnp*. Have the authors convinced themselves that altered *Cnp* expression in the knockouts does not contribute to the observed alterations in myelin and axonal conductivity? There is evidence in the literature that altered *Cnp* levels have effects on *Mbp* (e.g. PMID: 9373033).

In the revised version, the authors now show that Schwann cells and PNS myelin are also affected in the *Gtf2i* knockouts. In the PNS, there are g-ratio alterations. The phenotypic abnormalities in the PNS seem even stronger than in the CNS and may actually be the main factor for the improved motor coordination. It is important to have this information in the revised manuscript. Nevertheless, the analysis of the PNS phenotype is very preliminary, only mentioned in passing and hardly discussed in its implications. Although, I understand that the PNS phenotype is not the main focus of the current study, its current presentation is a bit odd.

To confirm the higher level of anxiety-like behavior, the authors added an elevated zero maze test (EZM) to the open-field exploration test already presented in the original version. However, the results from the EZM only show “a trend towards higher anxiety levels in *Gtf2i*-KO mice”. As confirmation, this is not really optimal.

Figure 5 shows increased expression levels of nuclear *Sox10* and *Mbp* in cultured oligodendrocytes with *Gtf2i* deficiency as well as an increased cell surface. Only one single cell is shown in the representative images and the information provided in

the Materials and Methods section and figure legend does not allow to judge the robustness of the quantifications. How did the cultures look in general? Were transduced cells healthy? How many cells per "n" underwent quantification? How exactly were intensity measurements performed?

Although I appreciate that the authors have undergone the trouble of performing new ChIP-seq experiments for the revised version, it is very difficult to judge the quality of the provided data in the absence of a GEO submission and an accessible GSE entry. With 8500 consensus peaks, the number is substantially higher than the one in a previous study from E13.5 brain (ref. 111 in the paper). How high is the overlap between peaks in both studies?

One of the highlighted outcome from the ChIP-seq studies is the identification of TfiI-i peaks in the vicinity of the Mbp and Sox10 genes (Fig6E,F). Do any of the identified peaks overlap with the regulatory elements mapped in previous studies for the Mbp and Sox10 genes? There is plenty of literature around, for instance from the labs of Alan Peterson, Tony Antonellis, Andy McCallion and Michael Wegner. If not, it seems far from conclusive that the peaks are bona fide regulatory elements of the Mbp and Sox10 genes. The folding of the distant cRE with the identified TfiI-I peak onto the Sox10 promoter is, for instance, merely an in silico prediction and there are several genes between the cRE and the Sox10 promoter so that assignment of the cRE the Sox10 gene is not obvious.

In the current state, I find the evidence provided for the proposed mechanism very preliminary and not convincing. Much remains on the correlational and suggestive level – both for Gtf2i as a direct repressor of Mbp and Sox10, and for the role of Mbp and Sox10 as effectors of the Gtf2i deletion. To substantiate the authors' hypothesis that increased Sox10 and Mbp expression upon Gtf2i deletion causes the phenotypic alterations, backup from the literature would be helpful. Is it known what experimentally induced overexpression of Mbp or Sox10 (starting in late OPCs at the earliest) does to myelin?

Reviewer #3

(Remarks to the Author)

The authors have addressed my previous concerns and the manuscript is much improved.

Version 2:

Reviewer comments:

Reviewer #4

(Remarks to the Author)

The revised manuscript describes an impressive phenotype obtained after Gtf2i deletion, including increased myelination and increased expression of myelin genes. While most of the figures are relatively convincing in the revised version, there are significant deficiencies with the requested response to the genomic analysis that was performed in Figure 7.

The issues in this figure are first apparent in the panel 7A, which shows radically different peak patterns in the biological replicates of the control samples. The results of panels E and F show mostly background binding and a lack of convincing peaks. An enrichment of SOX10 and OLIG2 motifs is interesting, but the examples for Mbp and Sox10 genes do not show colocalization with any of the known SOX10 and OLIG2 binding sites nor with any of the known enhancers of these genes (see H3K27ac track). An enrichment of these binding sites would normally predict a more substantial association of GTF2I with H3K27ac. Overall, as detailed below, there is little evidence to conclude that (as stated): "These findings suggest that TfiI-i may function in a synergistic transcriptional role alongside Olig2 and Sox10 in OLs."

The genome browser sessions are not optimal since the principal transcription start site of Mbp (downstream of M3 and M5) is not clearly designated in panel E. The farthest upstream exon is for the golli isoform of Mbp, which is a relatively minor transcript. In addition, the Sox10 locus is showing putative regulatory elements downstream of SOX10, although a number of studies have principally localized enhancers upstream of the SOX10 gene, which is omitted in this diagram. The suggestion that the TFII-I peaks coincide with predicted enhancer/promoter elements is not supported by reference to oligodendrocyte data. Since there is a previous analysis of OLIG2 binding sites (in reference 160) and GTF2I binding (ref. 113), these should be shown in these browser sessions as comparisons. Some basic information is lacking such as whether the sequences at the GTF2I binding sites are conserved in other species (e.g. mouse/human) and whether GTF2I binding is similar to the patterns in ref. 113. Also, the colocalization of GTF2I with CTCF binding as reported in ref. 113 is not assessed here, which is important to assess consistency with previous findings. Finally, the addition of the 3CqPCR experiment in panels G and H should have multiple negative control primer sets in the same locus rather than a single negative control on a different chromosome. In general, analysis of 3C data requires multiple negative controls that are lacking here. Based on these considerations, I would recommend deleting Figure 7 in its entirety.

Since the genomic analysis is not too convincing, there should be a more explicit consideration of models in which the SOX10 overexpression itself may account for the entire phenotype (or also the depletion of GPR37 as a negative regulator of myelination). More explicit statements should be provided as to how this phenotype compares with the phenotypes of SOX10 overexpression or GPR37 loss-of-function to assess whether these changes could largely account for the phenotype shown here. Another notable omission is the well established role of GTF2I in DNA repair, and indeed sites of DNA repair can be enriched in H3K36me3, but no mention of that is provided in the manuscript.

Other comments:

p. 16: is the "myelin fraction" purified myelin, or does it have intact cells?

p. 17: please correct: "1.375+/-0.0

Figure 5: Figure legend should state how protein and mRNA data are normalized.

figure 6: there are several references to "one-sample t-test" here and elsewhere, should this be 1-sided t-test?

It is noted that the manuscript employed a different antibody from a previous publication (ref. 113) with much better results of GTF2i binding. Also, since the read mapping from mouse cells was done in the mouse genome, why was the LiftOver tool used in the ChIP-seq data analysis? Primer sets for 3C-qPCR are not provided.

Version 3:

Reviewer comments:

Reviewer #4

(Remarks to the Author)

The revised manuscript has addressed some of the critiques raised in the first round regarding the rigor of the genomic analysis of Gtf2i binding. In particular, the 3C analysis in Figure 7H has much more convincing controls in this round. However, I remain concerned regarding the binding analysis by ChIP-seq in the remainder of Figure 7. The revised manuscript has done some further comparisons as requested with previous data sets of SOX10 and OLIG2 binding sites, along with a previous analysis of Gtf2i from another publication. However, there remain some internal contradiction since there is an apparent colocalization of Gtf2i with OLIG2 and SOX10, but this is not observed in the two loci shown: Mbp and Sox10. If there are Gtf2i regulated genes where such colocalization is evident, it should be shown. Moreover, the colocalization of Gtf2i with H3K36 methylation may simply be due to the broad distribution of H3K36 methylation within gene bodies, and there are only discrete sites of Gtf2i binding within these broad regions that constitute the overlap. I would recommend that the statistical significance of this overlap should be calculated. The significance of the overlap with H3K27ac should also be calculated, since this modification is typically more restricts, and is associated with enhancer elements.

Finally, the response provided for the apparent wide disparity of Gtf2i binding patterns among biological replicates in the revised Figure 7A is not adequate. It is still apparent that many very strong peaks are not commonly shared across biological replicates, which is unexpected. At the very least, probably it should be stated how many peaks are called for each replicate, and how many are shared across all 5, since it is stated that the analysis requires consensus peaks across biological replicates. Typically, one would expect strong peaks to be shared across replicates, and the lack of replication of strong peaks is evident in the revised Figure 7A. To this point, I would recommend showing peak calling for all 5 replicates for a single locus (e.g. Sox10) (or perhaps in a single chromosome) in a supplemental figure to provide a better illustration of inter-replicate variability, which is a key issue in interpreting the data.

While I remain concerned regarding the quality and usefulness of the ChIP-seq data, the added analyses and modified figures do now allow readers to assess how these data conform (or do not conform) to previous studies of GTF2i, SOX10 and OLIG2 binding, along with the known regulatory elements of Sox10 and Mbp. The remainder of the manuscript in figures 1-6 provides a fairly convincing analysis, and appropriate changes were made to the Discussion.

There seems to be a formatting issue with Supplemental Figure 12.

REVIEWER COMMENTS

Reviewer #1 (Remarks to the Author):

Levy and colleagues analyze in their study the role of the general transcription factor Gtf2i/Tfii-I in differentiating oligodendrocytes and myelination. They show alterations in the number of white matter fibers in the brain of adolescent mice and in the myelination pattern that correlates with increased conduction properties in the corpus callosum and altered behavior in tests for motor coordination and anxiety-like behavior. They furthermore present evidence in support of a model where Gtf2i exerts its effects in oligodendrocytes by binding to regulatory regions of Sox10 and Mbp and repressing its expression, presumably by Suz12/PRC2 recruitment and increased H3K27 trimethylation. The study follows a previous analysis on the consequences of Gtf2i deletion in neurons and derives part of its relevance from the fact Gtf2i is within a chromosomal region whose deletion or duplication leads to severe neurodevelopmental syndromes such as Williams syndrome.

While it is interesting in principle to unravel the cause of neurodevelopmental syndromes and the molecular mechanism, by which a component of the general transcription machinery can specifically interfere with processes in myelinating oligodendrocytes, my enthusiasm for the present study is not particularly high because of several limitations, shortcomings, exaggerations and its preliminary state. Specifically:

We thank the reviewer for the valuable and thorough input. We have taken the reviewer's remarks with great consideration and attempted to answer all comments. We believe that the revised manuscript has greatly improved, thanks to the reviewer's informative notes.

1. It is a general limitation that the authors chose to study *Gtf2i* loss in oligodendrocytes using the *Cnp-Cre* driver. *Cnp-Cre* deletes very late in oligodendrocyte progenitor cells (OPCs). As a consequence, the study will only be able to detect *Gtf2i* effects on late events of oligodendrocyte development. Any earlier effects remain unexplored. This, however, may be highly relevant if the idea of the study is to learn about neurodevelopmental disorders such as Williams syndrome.

Response:

We thank the reviewer for the keen comment. The reviewer is correct in pointing that the *Cnp-Cre* line is relevant only to assess late developmental aspects in OLs, from between the very late OPC stage and newly formed OL stage, specifically. While assessing the role of *Tfii-i* across the oligodendroglial lineage is extremely interesting, as the reviewer mentioned, our goal in this manuscript was to specifically elucidate *Tfii-i*'s role in mature OLs, and how its deletion affects myelination. Because neuronal deletion of *Tfii-i* resulted in myelination deficits^[1], with specific deficits in OLs health and numbers, we sought to assess *Tfii-i*'s role in mature OLs, that are in charge of producing myelin in the nervous system. We believe that this work is of high interest to researchers in the field of transcriptional regulation of myelin as well as to the *Gtf2i/Tfii-i* community.

As such, the focus of the manuscript and its relevance towards 7q11.23-related neurodevelopmental disorders (WS amongst them), is limited. Following the reviewer's comment, we changed text along the manuscript, to better define the focus and meaning of our manuscript:

- **Abstract – added this sentence at the end of the abstract section:** 'These findings add to our understanding of myelination regulation and specifically elucidate a cell-autonomous mechanism for *Tfii-i* in myelinating glia transcriptional network.'
- **Discussion – replaced the opening text of this section to better convey our message:** 'In the present study, we identified *Tfii-i* as a novel regulator of the myelin transcriptome, acting as a transcription mediator of *Mbp* and *Sox10*, specifically in mOLs. *Gtf2i*'s deletion from myelinating cells resulted in PNS hypermyelination and altered interplay of myelin thickness and axonal diameters in the CNS, with functional physiological and behavioral alterations. These mutant mice exhibited WM

alterations, enhanced axonal conductivity and improved motor coordination. Furthermore, cellular and molecular data in our study identify Tfii-i as a novel regulator in the transcriptional program of mOLs and suggest its mode of action is cell-autonomous.'

2. The authors do not provide quantifications for the efficiency of Gtf2i deletion (Fig. 1). Instead they show representative pictures with one or two oligodendrocytes. Quantification is definitely needed and pictures should show at least 10-20 oligodendrocytes.

Response:

We thank the reviewer for raising this concern. We've conducted a new analysis and replaced the representative pictures, now showing about 10 OLs (Figure 1A). Moreover, as requested by the reviewer, a quantification of the deletion of Tfii-i in mOLs was taken (Figure 1B).

Attached are the relevant new figures, for convenience:

In addition, the efficiency of *Gtf2i* deletion from mature OLs is further quantified by additional method, demonstrating decreased Tfii-i expression level in the crude myelin fraction derived from the cortex *Gtf2i*-KO mice, as compared to controls (supplementary figure B-C, also presented below).

We also made minimal changes in the text to describe these results. The revised text is attached below: ‘To validate the genetic deletion of *Gtf2i* selectively from mOLs (Figure 1A) and the absence of the protein product, Tfii-i, in mOLs, we performed immunohistochemical staining, showing decrease in Tfii-i expression levels in CC1⁺ cells in *Gtf2i*-KO mice and intact expression in littermate controls (*Gtf2i*^{fl/fl}, *Cnp-Cre*^{-/-}, hereby termed control mice) (Figure 1B, Supplementary Figure 1). Tfii-i was previously shown to be expressed and have different roles in the nucleus and in the cytoplasm^[2, 3]. However, its expression level and roles were never directly assessed in the myelin fraction. Thus, we characterized its expression in the myelin fraction purified from whole cortex to study whether Tfii-i is specifically localized to myelin. While we found no evidence for Tfii-i expression in the purified myelin fraction (Supplementary figure 1B), Tfii-i levels in the crude myelin fraction, which may be partly contaminated with cellular and nuclear debris from both OLs and axons^[4], were decreased in *Gtf2i*-KO mice, as compared to controls (Supplementary figure 1C). As such, we showed that Tfii-i is expressed in OLs and validated its deletion from mOLs in *Gtf2i*-KO mice.’

3. The authors argue that their *Gtf2i* mutants exhibit hypermyelination. At the same time their data show that “g-ratio values in the overall axonal population of the CC were not significantly different” (p.7, line 154). This rather argues against a simple hypermyelination. Instead, the authors rather observe higher axon diameters and a lesser impact of axon diameters on g-ratio. This observation may actually indicate that the alterations in the *Gtf2i* mutants is due to an altered interaction of the myelin sheath with the axon and not so much due to alterations in the myelination process per se.

Response:

We thank the reviewer for this insightful comment. We agree with the reviewer that the data support an altered interplay between axonal diameter and myelin thickness. As such, we revised the text and omitted the term 'hypermyelination' to more accurately describe the moderately altered interplay between axonal diameter and myelin thickness in the CC of *Gtf2i*-KO mice (Figure 2A-D, also presented below).

Figure 2

To strengthen our data, we added an additional timepoint at postnatal day 90 (P90). These new results indicate that while the increased axonal diameter does not persist into adulthood in *Gtf2i*-KO mice (Supplementary Figure 3D, presented below), the moderate effect of *Gtf2i* deletion from mOLs on g-ratio (Supplementary Figure 3C, presented below) and myelin thickness does persist through adulthood (Supplementary Figure 3E, presented below).

Supplementary Figure 3

Please note that in the original manuscript, we mistakenly presented the overall axonal diameter (inner diameter + myelin sheath) in the EM figures of the CC. We have now corrected this in the revised manuscript to show the inner diameter values. This correction does not drastically change the results or conclusions. We still observe significantly increased axonal diameter in our mutant mice (Figure 2C, above), along with a significantly increased myelin thickness (Figure 2D, above). The corrected figure and text are attached below:

‘While g-ratio values in the overall axonal population of the CC were not significantly different, the interplay between axonal diameter and g-ratio values differed significantly between *Gtf2i*-KO

mice and controls (Figure 2B). Specifically, the slope representing this interplay is more gradual in *Gtf2i*-KO mice axons, indicating that the *g*-ratio values of *Gtf2i*-KO mice axons are less affected by axonal diameter, compared to controls. In addition, the diameters of *Gtf2i*-KO mice axons were significantly increased, as compared to controls (Figure 2C, dashed vertical line indicates the median axon diameter of the control group, 0.53 μ m). Axons with larger diameters, as seen in the CC of *Gtf2i*-KO mice, are typically wrapped with a thicker myelin sheath^[5]. Accordingly, *Gtf2i*-KO mice axons demonstrated thicker myelin sheath, compared to controls (Figure 2D).'

4. As the authors are surely aware, the *Cnp*-Cre driver is also active early in some neural crest cells and in myelinating Schwann cells. Therefore, it cannot be excluded that part of the behavioral alterations are caused by peripheral effects instead of changed oligodendrocyte/myelin properties. This is particularly relevant for motor behavior.

Response:

The reviewer raised an important point, prompting us to thoroughly investigate this issue. Indeed, *Cnp* promoter is active in the PNS, specifically in myelinating Schwann cells (SCs). To determine whether the absence of *Gtf2i* in SCs affects myelin properties in the PNS, we assessed the myelin properties in the tibial branch of the sciatic nerve (SN) using EM (Figure 2J, presented below). Our results show that the deletion of *Gtf2i* in SCs leads to hypermyelination in the PNS, as indicated by decreased *g*-ratio values in the axons of *Gtf2i*-KO mice compared to controls (Figure 2K-L, presented below).

Figure 2

The following text was added to the revised manuscript to communicate these new finding: ‘Since the *Cnp* promoter is also active in SCs of the PNS, we examined the myelin ultrastructure in the tibial branch of the SN of P30 mice by TEM (Figure 2J). The *g*-ratio values of *Gtf2i*-KO mice axons were significantly lower compared to controls (Figure 2K-L), indicating a hypermyelination phenotype in the PNS of *Gtf2i*-KO mice, while axonal diameters remained unchanged (Figure 2M).’

We also added an additional timepoint for this assessment, showing similar results in older mice (P90), suggesting that this hypermyelination is long-lasting (Supplementary Figure 3G-H, presented below).

While it's difficult to determine whether the altered CNS or PNS myelin properties are the main drivers of the behavioral phenotype, we have added text to the results section of the manuscript to underline both possibilities. This includes a specific focus on the altered motor behavior, as the reviewer mentioned in their comment, as follows: 'Motor coordination was shown to be dependent on myelin condition^[6-8]. Key regions involved in motor coordination include the cerebellum in the CNS, a brain region responsible for the control, timing and coordination of motor functions^[9], and the SN in the PNS^[10, 11]. In addition, deficits in CC myelination have been linked to abnormal anxiety behavior^[12, 13]. Given the observed alterations in CNS and PNS myelin ultrastructure in *Gtf2i*-KO mice, we next assessed whether these alterations had behavioral consequences'.

5. The authors argue for increased anxiety-like behavior in their mouse mutants on the basis of open-field exploration tests. While the obtained results are consistent with this assumption, it would probably need additional tests for confirmation.

Response:

We thank the reviewer for raising this valid concern and agree that additional data will strengthen the claim of increased anxiety in *Gtf2i*-KO mice. To address this, we conducted the elevated zero maze (EZM) experiment with a new cohort of test mice. We observed moderately increased anxiety in the mutant mice compared to controls (Figure 3D-E, presented below). These findings are consistent with the open field results, reinforcing our observation that *Gtf2i*-KO mice exhibit moderately increased anxiety.

We have revised the text accordingly, and the text to describe the EZM results in the manuscript is as follows: ‘To further examine anxiety-like behavior, control and *Gtf2i*-KO mice were subjected to the elevated zero maze (EZM) test. The results showed a trend towards higher anxiety levels in *Gtf2i*-KO mice, indicated by their spending less time in the open arms compared to controls (Figure 3D-E, mean control 249.1s±19.62s, mean *Gtf2i*-KO 200.3s±16.59s). Together, these data suggest that *Gtf2i*-KO mice possess improved motor coordination and moderately increased anxiety-like behavior, as compared to controls.’

6. The evidence for the proposed molecular mechanism is very thin. There is no general approach to map the *Gtf2i*-induced changes by RNA-seq, ChIP-seq or similar methods. Instead, previously published ChIP-seq data are taken as justification to focus on Sox10 and Mbp. However, most of these ChIP-seq data are not from oligodendrocytes but from other cells. Considering that *Gtf2i* is present in most, if not all cells, but differentially recruited to promoters that are active in the respective cell at the particular time, *Gtf2i* occupancy in neural precursor cells etc. is not very relevant for oligodendrocytes. As is, the focus on Sox10 and Mbp appears arbitrary.

Response:

The reviewer raises a valid point regarding the thin evidence for the proposed molecular mechanism and the reliance on previously published ChIP-seq data that are not specific to OLs. To address this, we have conducted a comprehensive proteome analysis of the myelin sheath (biochemically-enriched myelin fraction) from the hemispheres of *Gtf2i*-KO and control mice (Figure 4A, presented below). This analysis revealed increased expression levels of Mbp and Mog, as well as a decrease in the expression level of Gpr37. Gpr37 is known to be a negative regulator of myelination, and its deletion results in precocious myelination^[14].

Overall, these findings provide a broader overview of *Gtf2i*-induced changes in the myelin sheath and support the involvement of these specific proteins in the observed phenotype. We acknowledge the limitations of focusing solely on Sox10 and Mbp based on non-oligodendrocyte ChIP-seq data and have thus expanded our approach to include proteomic analysis for a more comprehensive understanding.

The data from our proteome analysis are visualized below, providing stronger evidence for the molecular changes induced by *Gtf2i* deletion.

The text to describe these data is as follows: ‘To assess how *Tfii-i* absence from OLS alters myelin properties on a molecular level, we examined the biochemically-enriched myelin fraction^[15] purified from a single hemisphere of P30 mice. The protein composition of the myelin fraction was assessed by label-free quantitative mass spectrometry^[15]. Proteome analysis revealed altered protein abundance of several key myelin proteins (Figure 4A and Supplementary Figure 5). Specifically, the relative abundance of Mog and Mbp was significantly elevated in myelin purified from *Gtf2i*-KO mice (Figure 4A), while the abundance of Gpr37 was reduced. Mbp is an abundant constituent of compact myelin, its expression being a rate-limiting step in myelination^[16]. Mog is a marker for the abaxonal myelin membrane^[17]. Gpr37 has been demonstrated as a negative regulator of myelination^[14]. Overall, the altered myelin proteome observed in *Gtf2i*-KO mice suggests a molecular hypermyelination phenotype in the absence of *Tfii-i*.’

The increased expression levels of Mbp and Mog were also observed in the cortical myelin fraction of P90 *Gtf2i*-KO mice (Figure 4G-H, presented below). These new results, not presented in the original manuscript, further emphasize the relevance of Mbp and Sox10 regulation in our study, as Sox10 has been shown to positively regulate Mbp expression^[18]. Testing at this later stage of development underscores the persistent impact of *Gtf2i* deletion on myelin regulation and supports the importance of these proteins in our proposed mechanism. This comprehensive approach not only addresses the reviewer’s concern but also solidifies our understanding of the molecular underpinnings driving the observed myelination changes in *Gtf2i*-KO mice.

The text to describe these new data in the manuscript is as follows: ‘*Gtf2i* and *Tfii-i*’s expression levels change across different developmental stages^[19]. To assess whether these alterations persist into adulthood, we examined the pure myelin fraction derived from the cortices of P90 mice. Our analysis showed increased protein expression levels of *Mbp* (Figure 4G, mean control 1 ± 0.027 , mean *Gtf2i*-KO 1.164 ± 0.066) and *Mog* (Figure 4H, mean control 1 ± 0.035 , mean *Gtf2i*-KO 1.217 ± 0.067). Additionally, we examined mRNA expression levels of *Sox10* (Figure 4I, mean control 1 ± 0.102 , mean *Gtf2i*-KO 1.279 ± 0.063), *Mbp* (Figure 4J, mean control 1 ± 0.09 , mean *Gtf2i*-KO 1.34 ± 0.071), and *Mog* (Figure 4K, mean control 1 ± 0.047 , mean *Gtf2i*-KO 1.328 ± 0.051) in the cortex of P90 *Gtf2i*-KO mice, revealing increased transcript expression levels of these genes compared to controls.

Overall, these molecular alterations detected in *Gtf2i*-KO mice suggest that *Tfii-i* plays a crucial role in regulating the myelination process in OLs across different developmental stages. Its absence leads to elevated mRNA and protein expression levels of key myelin components.’

We agree with the reviewer that a generalized approach to mapping *Tfii-i* occupancy at regulatory elements, specifically in mature oligodendrocytes (mOLs), would strengthen our claim and shed light on *Tfii-i*’s mode of action in mOLs. These data will be of utmost interest to both the *Gtf2i*/*Tfii-i* community and the transcriptional regulation of myelin community. To assess *Tfii-i* occupancy at regulatory elements specifically in OLs, we performed a ChIP-Seq experiment on an OL-enriched cell culture (Figure 6, presented also below for convenience).

We first confirmed the validity of our ChIP-Seq data by showing a significant decrease in Tfii-i binding in *Gtf2i*-KO samples compared to controls (Supplementary Figure 10). Consistent with OL-specific data derived from brain tissue of WS patients^[20], the ChIP-Seq experiment revealed that Tfii-i's mode of action in OLs is predominantly focused on enhancers rather than promoters (Figure 6A). Furthermore, we observed that Tfii-i binding sites in mOLs are in close proximity to those of Sox10 (Figure 6B), suggesting that these two transcription factors may synergize in regulating myelination in mOLs. While a small number of peaks were observed in these genomic loci, GO analysis of Tfii-i peaks at promoters, exons, and transcription termination sites revealed an enrichment of genes associated with pathways related to DNA modification, negative regulation of gene expression, and brain development (Figure 6C).

Next, we aligned Tfii-i peaks with predicted active enhancers specifically in mOLs^[21]. We found 1,352 overlapping peaks at predicted enhancers, which contribute to the regulation of genes associated with nervous system development, cell projection morphogenesis, and negative regulation of myelination (Figure 6D). Finally, we observed several Tfii-i binding sites on different regulatory regions of *Mbp* (Figure 6E) and at a genomic locus predicted to act as an enhancer that folds into the promoter of *Sox10* (Figure 6F). These data add an important layer of knowledge to Tfii-i's mode of action and contribution to the transcriptional regulation scheme of mOLs, and how its absence affects development and myelination. Below is the text of the ChIP-Seq results and the figure: 'In searching for a cellular mechanism to explain the altered myelination properties in *Gtf2i*-KO mice, we characterized the general OL population and the specific mOL pool in the motor cortex and CC using immunohistochemistry (Supplementary Figures 7, 8). Our analysis revealed that the total number of oligodendroglial lineage cells (Olig2⁺ cells) and mOLs (Olig2⁺ + CC1⁺ cells) in the motor cortex and CC from both strains did not differ significantly (Supplementary Figure 8). To ensure that differentiation properties remain unchanged throughout development, we performed similar experiments at postnatal ages of 14 and 90 days (Supplementary Figure 8), as well as *in-vitro* using a primary OL-enriched culture (Supplementary Figure 9). Overall, no significant differences in differentiation were found between *Gtf2i*-KO and control mice.

As such, changes in the number of mOLs cannot account for the observed myelination and axonal alterations in *Gtf2i*-KO mice, suggesting that these alterations are due to a cell-

autonomous mechanism triggered by the deletion of *Gtf2i* in mOLs. However, the precise mechanisms by which Tfi-i modulates the expression of myelination-related genes remain inadequately understood. To unravel the nature of its regulation, it is crucial to identify the specific DNA loci to which Tfi-i directly binds in mOLs. By pinpointing these binding sites, we can gain valuable insights into Tfi-i's mode of action in mOLs. Consequently, in the next phase of our research, we aimed to comprehensively map the genome-wide Tfi-i binding sites and enrichment levels, specifically in mOLs.

To achieve this, mOLs derived from primary, fully differentiated, OL-enriched cell cultures were subjected to chromatin immunoprecipitation followed by sequencing (ChIP-seq), targeting Tfi-i. As expected, Tfi-i peaks were highly abundant in control compared to *Gtf2i*-KO samples (Supplementary Figure 10). Overall, 8429 unique genomic loci were identified as high-confidence Tfi-i binding sites in control samples (i.e. consensus peaks), with minimal overlap in *Gtf2i*-KO samples (Figure 6A and Supplementary Figure 10). Functional annotation analysis revealed that Tfi-i binding sites in control samples were predominantly enriched in introns and intergenic regions; intergenic (4568 peaks), introns (3314 peaks), exons (306 peaks), promoters (140 peaks), and transcription termination sites (TTS) (101 peaks) (Figure 6A). These data correlate well with our previous findings, where we identified cell-type-specific aberrant methylation profiles on enhancers in post-mortem brain tissue of individuals with WS^[20]. Combined, these data suggest that, in mOLs, Tfi-i predominantly binds to enhancers rather than promoters to regulate cell-type-specific gene programs. Interestingly, analysis of genomic sites in the vicinity of Tfi-i peaks (within a 200 bp window) revealed an enrichment of binding motifs for specific transcriptional regulators, most notably is Sox10 (Figure 6B). Hence, it is plausible that Tfi-i could exhibit synergistic transcriptional activities with these regulators, Sox10 among them, pivotal in modulating myelination processes.

Gene ontology (GO) analysis of Tfi-i consensus peaks annotated to exons, promoters and TTSs revealed enrichment in pathways associated with DNA modification, negative regulation of gene expression, and brain development (Figure 6C). To determine whether the intergenic and intronic regions harboring Tfi-i peaks function as transcriptional REs in OL, we overlapped our ChIP-seq datasets with a previously published OL-specific predictive atlas of *cis* REs (cREs) in mice

(CATlas)^[21]. Overall, Tfi-i consensus peaks aligned with 1352 unique OL-specific cREs and were then mapped to putative target genes using CATlas^[21]. GO analysis of these target genes showed enrichment in pathways associated with nervous system development, cell projection morphogenesis, locomotory behavior, negative regulation of myelination, and negative regulation of gene expression (Figure 6D). Among the high-confidence Tfi-i peaks, we observed relevant Tfi-i binding peaks at the *Mbp* and *Sox10* loci (Figure 6E and F). For *Mbp*, Tfi-i binding was observed at three different genomic loci; Exon 4 (out of 4), intron 2 (out of 6), and TTS (Figure 6E). Additionally, Tfi-i binding was detected upstream to *Mbp* at a distant intergenic locus (~25kb upstream to TSS, Figure 6E). Although we did not observe direct binding at the *Sox10* genomic loci, we found a high-confidence Tfi-i peak at a distant RE predicted to fold into the promoter region of *Sox10* (Figure 6F).

Our findings suggest that in mOLs, Tfi-i transcriptional regulation is primarily mediated through enhancers. Furthermore, in mOLs, Tfi-i is involved in the transcriptional program of the myelination process and binds to REs of genes associated with governing cellular structural properties. Importantly, these findings shed light on Tfi-i's transcriptional mode of action in mOLs, specifically its role in regulating key myelination genes such as *Sox10* and *Mbp*. These data may underline the mechanism of the observed altered interplay of myelin thickness and axonal diameters in the CC of *Gtf2i*-KO mice.'

As both the proteome analysis and ChIP-Seq experiments provide an extensive overview on *Gtf2i*-induced changes in OLs and myelination, the data derived from these experiments further validates Tfii-i's role in regulating *Mbp* and *Sox10* in OLs and immensely strengthens the manuscript.

7. The proposed mode of action of Gtf2i on Sox10 and Mbp expression is highly speculative, and probably not too likely. There is little experimental evidence provided for the proposed PRC2 recruitment. PRC2 recruitment and H3K27 trimethylation are predominantly used for switching genes off rather than reducing expression levels gradually, and might therefore not be fitting as a mechanism in the presented case. The little evidence provided in the paper furthermore comes from ChIP-PCR experiments (Fig. 7D,E) that are not too convincing because effects are fairly moderate and experiments were performed with chromatin from whole cortex (and thus from a mix of many different cells) and not from oligodendrocytes. By the way, what are the “targeted sites” (p.12, line 244) studied in ChIP-PCR experiments? Are these the respective gene promoters?

Response:

We thank the reviewer for this insightful comment. We agree that the mechanism involving Suz12/PRC2 recruitment is mostly speculative, and therefore, we have removed the text regarding Suz12/PRC2 recruitment in the revised results section.

To strengthen our claim that Tfii-i binds to the regulatory elements of Sox10 and Mbp, and to better represent the mode of transcription regulation Tfii-i is executing in mOLs, we have conducted a ChIP-Seq experiment using an OL-enriched culture, as detailed in our previous response. This new set of data provides a clearer picture of Tfii-i's binding sites and its role in transcriptional regulation in mOLs. As a result, we have removed the results of the ChIP-qPCR experiment from the revised manuscript.

Additionally, the ChIP-Seq data from the OL-enriched culture indicate that Tfii-i binding predominantly occurs at enhancers rather than promoters, which aligns with the transcriptional regulation mechanisms in oligodendrocytes. This new evidence provides a more convincing and specific demonstration of Tfii-i's role in regulating Sox10 and Mbp expression.

The "targeted sites" mentioned in the original manuscript referred to the respective gene promoters. However, with the new ChIP-Seq data, we can more accurately define these regulatory regions and provide a comprehensive understanding of Tfii-i's involvement in the transcriptional regulation of key myelination genes.

8. From the flow of the paper, Fig. 8 is meant to support the proposed molecular mode of Gtf2i interaction. However, all it does is to show that manipulations of Gtf2i levels in oligodendrocytes also change Sox10 and Mbp levels. This is furthermore done by measuring intensities of immunofluorescently stained cells, which is probably not a very robust assay. What the experiment does not show is whether effects are direct or indirect, and whether the proposed mechanism holds. No rescue has been performed.

Response:

We thank the reviewer for their insightful comment and agree that the original Figure 8 did not fully support the flow of the paper. The goal of the exogenous deletion of *Gtf2i* in an OL-enriched primary cell culture was to assess whether the molecular effects of *Gtf2i* deletion depend on other cell types. By showing that Sox10 and Mbp levels are increased following exogenous deletion of *Gtf2i* in an OL-enriched primary cell culture, we provide evidence that the mode of action is probably cell-autonomous and specific to OLs. We have clarified this point in the text and reordered the figure in the manuscript to better support this claim. The added text is as follows:

‘These findings suggest that, in the absence of Tfii-i, the observed increased expression levels of Mbp and Sox10 are a result of a cell-autonomous mechanism in mOLs, unrelated to other cell types.’

While we agree with the reviewer that CTCF is not the most robust assay for assessing expression levels, experiments involving viral transfection (such as AAV) on OL-enriched cultures are particularly straining on these cells, making downstream applications challenging. Due to the low number of viable cells following manipulation, we sought an assay that could capture molecular alterations at single-cell resolution. Thus, CTCF was used to assess the expression levels of Sox10 and Mbp. For the same reason, we have not succeeded in performing a rescue experiment.

9. Given the fairly mild effects of *Gtf2i* deletion on oligodendrocyte differentiation and myelination, it is unclear to me why one would state that “we identified *Tfii-i* as a novel master regulator of the myelin transcriptome” (p.17, line 285). In the same vein, I have a tough time to see the “important basis for a potential new therapeutic target for WM disorders and conditions in which CNS myelination is required” (p.19, line 348). To me it is not even obvious what I learn from the study regarding 7q11.23 microdeletion/duplication syndromes.

Response:

We thank the reviewer for this important clarification. Following this comment, we have removed the term "master regulator" to describe *Tfii-i*'s role in myelination from the revised manuscript, as we agree with the reviewer's perspective. Throughout the manuscript, we have toned down our claims to better reflect the extent of our data interpretation.

Examples of alterations in original text meant to tone down our claims:

1. Deleted line 31 (previously last line in abstract): ‘...possibly underlying a new therapeutic approach toward white matter pathologies.’
2. Previous claim (p. 17, line 285): ‘In the present study, we identified *Tfii-i* as a novel master regulator of the myelin transcriptome, acting as a transcription inhibitor of *Mbp* and *Sox10*. Moreover, we characterized a new mouse model of CNS hypermyelination with improved physiological and behavioral phenotypes.’

Revised claim (p. 14, line 387, first line in discussion): ‘In the present study, we identified *Tfii-i* as a novel regulator of the myelin transcriptome, acting as a transcription mediator of *Mbp* and *Sox10*, specifically in mOLs.’

3. Previous claim (p. 19, line 348): ‘our findings nonetheless provide an important basis for a potential new therapeutic target for WM disorders and conditions in which CNS re-myelination is required.’

Revised claim (p. 16, line 453, last two lines in discussion): ‘*Gtf2i* absence from mOLs promoted cellular, molecular, and structural changes, resulting in alterations of myelination in the CNS and PNS. Finally, we provide evidence that *Tfii-i*'s mode of action in mOLs is cell-specific, and shed light on its regulatory targets in mOLs.’

We would like to emphasize that the focus of this paper is on Tfii-I's transcriptional role in mature oligodendrocytes (OLs) and how its deletion contributes to the altered interplay between axons and the myelin sheath in the CNS, as well as hypermyelination in the PNS. We acknowledge that this focus was not clearly conveyed in the original manuscript. Therefore, we have revised the text across the manuscript to better communicate this point.

Examples in revised text to better convey the aim of the study:

1. P.1, line 27: 'In this study, we have identified *Gtf2i*-encoded general transcription factor II-I (Tfii-i), as a regulator of key myelination-related genes'.
2. P.1, line 37: 'These findings add to our understanding of myelination regulation and specifically elucidate a cell-autonomous mechanism for Tfii-i in myelinating glia transcriptional network'.
3. P.4, line 111: 'While these findings indicate that neuronal *Gtf2i* expression affects OL properties^[22], the role of *Gtf2i* role in OLs is unknown and was, therefore, the focus of the current study.'
4. P.4, line 120: 'Therefore, to study Tfii-i-mediated transcriptional regulation specifically in myelin-producing cells (i.e., mOLs and SCs), we homozygously deleted *Gtf2i* in myelinating glia using the *Cre-loxP* system'.
5. P.4, line 135: 'Together, these data suggest that Tfii-i serves an important cell-autonomous role in mOLs, acting as a regulator of myelination, with its absence resulting in functional alterations of CNS myelin and hypermyelination in the PNS.'
6. P.14, line 387: 'In the present study, we identified Tfii-i as a novel regulator of the myelin transcriptome, acting as a transcription mediator of *Mbp* and *Sox10*, specifically in mOLs'.
7. P.16, line 450: 'In summary, we identified *Gtf2i*-encoded Tfii-i as a novel regulator of myelination in mOLs, demonstrating its direct binding, specifically in mOLs, to *Mbp* regulatory regions and *Sox10* distal enhancer'.

Reviewer #2 (Remarks to the Author):

The authors report the interesting observation that a general transcription factor serves as an endogenous “repressor” of myelination in that the loss of function mutation in oligodendrocytes causes a hypermyelination phenotype, but without changing glial cell numbers. This is associated with the activation of a known promyelination transcription factor (SOX10) and a critical protein for myelin assembly (MBP), which reportedly improves functional performance in myelination dependent tasks. The latter is quite surprising given that wildtype myelination (g-ratio) is “optimal” and that other hypermyelinating mice have not shown increased nerve conduction velocities (which is different from improved motor performance in mice with more oligodendrocytes, e.g. Gibson et al. 2014).

Response:

We thank the reviewer for the time and effort invested in improving our manuscript. We have carefully considered all comments and have addressed them below. Thanks to the reviewer’s considerable contribution, we feel the manuscript has greatly improved.

We agree with the reviewer that hypermyelination does not necessarily result in increased nerve conduction velocities. However, our data suggest that axonal diameters were also increased in the CC of *Gtf2i*-KO mice compared to controls, which may also mediate the enhanced signal conduction properties observed in the CC of our mutant mice.

Overall, this is an interesting study. Some analyses appear still a little superficial, but that does not diminish the importance of these findings in the field. The following points should be revised/considered:

1. Line 80-92: The relationship between gene and protein names *GTF21*, *Gtf2i*, *GTFII-I*, *Tfii-i* is very confusing for the reader. It took me an internet search to realize who is who. Please use the same name throughout. Also in the introduction, gain and loss of function effects determined in previous publications should be better explained and contrasted, also within the present paper, if necessary with a supplementary small table.

Response:

We thank the reviewer for their remarks. To address the confusion, we have clarified the relationship between the gene and protein names in the introduction. *GTF2I* is the human gene name, and *GTFII-I* is its protein product. *Gtf2i* is the murine gene name, and *Tfii-i* is its protein product. To ensure consistency and clarity throughout the manuscript, we have standardized the terminology used.

Below are some of the text alterations we made to clarify the relationship between gene and protein names:

- **Line 84-86:** 'Encoded by *GTF2I* in humans, *GTFII-I* is a widely expressed TF, shown to interact with promoter initiator elements and DNA binding motifs such as the E-box element^[23].'
- **Line 109:** '...*Gtf2i*, the murine homolog of *GTF2I*,...'
- **Line 115:** '...*Tfii-i* (the murine protein product of *Gtf2i*)...'

Furthermore, we have added more information and references specifically regarding *GTF2I*'s contribution to behavioral and cognitive manifestations in 7q11.23 copy number variation syndromes to the introduction of the revised manuscript. For convenience, we have attached the text we added to the revised manuscript here: 'Copy number variation studies of this chromosomal region specifically associate *GTF2I* levels with the cognitive and behavioral manifestation^[24-30]. For example, *GTF2I* duplications were shown to result in increased anxiety^[25], and ASD-like phenotype^[31]. Furthermore, atypical deletions of *GTF2I* results in social disinhibition^[24, 27, 29, 32-34] and intellectual disability^[24, 26, 30, 33]. Additionally, Single nucleotide polymorphisms (SNPs) of *GTF2I* were associated with increased anxiety^[35, 36] and ASD^[31]. Finally, SNPs of *GTF2I* are also associated with autoimmune disorders^[36, 37] and thymic epithelial cancer^[38, 39].'

2. Line 90: please add to the introduction (and reference) prior work that showed this TF to be widely expressed and to engage with the transcriptional core complex. That helps understanding Fig 1B. Does the TF have similar functions in other cells and how specific is the effect here to myelin genes?

Response:

The reviewer raises valid concerns. We have incorporated additional information and references regarding *GTF2I*'s expression levels and functions in different cell types into the revised manuscript (also copied below for convenience).

It is important to note that while GTFII-I has been shown to interact with various components of transcriptional complexes, it does not interact directly with the transcriptional core complex itself^[40]. Given the significance of this note for the TF community, we have added the following text and references to the manuscript: 'GTFII-I interacts with various components of transcriptional complexes to mediate transcription^[41], such as Suz12 of the polycomb repressive complex 2 (PRC2)^[42], but not with the transcriptional core complex itself^[40]. GTFII-I is also active outside of the nucleus, and acts as an inhibitor of agonist-induced calcium entry in the cytoplasm^[2]. Additionally, GTFII-I is essential in embryonic development^[43, 44], critical for cell-type specific immune functions in B^[45, 46] and T cells^[47], and is involved in the regulation of the endoplasmic reticulum stress response pathway^[48]'.

As for the reviewer's comment regarding *Tfii-i*'s functions in different cell types and its effect on myelin gene, we fully agree with the reviewer that it is crucial to assess the specificity of the effect of *Gtf2i* deletion from mOLs on myelinating genes. These data will be of utmost interest to both the *Gtf2i*/*Tfii-i* community and those studying the transcriptional regulation of myelin. To address this, we undertook a comprehensive set of experimental efforts, as described below and in the revised manuscript.

To directly investigate this, we performed a ChIP-Seq experiment specifically in mOLs (Figure 6, presented also below for convenience). This approach allowed us to precisely map the binding sites of *Tfii-i* in oligodendrocytes and determine its direct targets in these cells.

First, we validated our ChIP-Seq data by showing a significant decrease in Tfii-i binding in *Gtf2i*-KO samples compared to controls (Supplementary Figure 10). This validation step was crucial to ensure the reliability of our data. In line with OL-specific data derived from brain tissue of WS patients^[20], our ChIP-Seq results revealed that Tfii-i's mode of action in OLs is predominantly focused on enhancers rather than promoters (Figure 6A).

Furthermore, we observed that Tfii-i binding sites in mOLs are in close proximity to those of Sox10 (Figure 6B). This proximity suggests that these two TFs may work together to regulate myelination in mOLs. This potential synergy between Tfii-i and Sox10 is a critical aspect of our study, highlighting a novel interaction that could be pivotal in understanding myelination processes.

While only a small number of peaks were observed in these genomic loci, GO analysis of Tfii-i peaks at promoters, exons, and transcription termination sites revealed an enrichment of genes associated with pathways related to DNA modification, negative regulation of gene expression, and brain development (Figure 6C). This analysis provided a broader context for Tfii-i's role in OLs, linking it to essential regulatory pathways.

Next, we aligned Tfii-i peaks with predicted active enhancers specifically in mOLs^[21]. We identified 1,352 overlapping peaks at predicted enhancers, which contribute to the regulation of genes associated with nervous system development, cell projection morphogenesis, and negative regulation of myelination (Figure 6D). This substantial overlap underscores the extensive regulatory network influenced by Tfii-i in OLs.

Finally, we observed several Tfii-i binding sites on different regulatory regions of *Mbp* (Figure 6E), and at a genomic locus predicted to act as an enhancer that folds into the promoter of *Sox10* (Figure 6F). These findings are particularly compelling as they directly link Tfii-i to critical myelination genes, providing a clear mechanistic pathway for its role in myelination.

We hope this comprehensive experimental effort addresses the reviewer's concerns and provides a clearer understanding of Tfii-i's specific effects in OLs.

Below is the text of the ChIP-Seq results and the figure: In searching for a cellular mechanism to explain the altered myelination properties in *Gtf2i*-KO mice, we characterized the general OL population and the specific mOL pool in the motor cortex and CC using immunohistochemistry (Supplementary Figures 7, 8). Our analysis revealed that the total number

of oligodendroglial lineage cells (Olig2⁺ cells) and mOLs (Olig2⁺ + CC1⁺ cells) in the motor cortex and CC from both strains did not differ significantly (Supplementary Figure 8). To ensure that differentiation properties remain unchanged throughout development, we performed similar experiments at postnatal ages of 14 and 90 days (Supplementary Figure 8), as well as *in-vitro* using a primary OL-enriched culture (Supplementary Figure 9). Overall, no significant differences in differentiation were found between *Gtf2i*-KO and control mice.

As such, changes in the number of mOLs cannot account for the observed myelination and axonal alterations in *Gtf2i*-KO mice, suggesting that these alterations are due to a cell-autonomous mechanism triggered by the deletion of *Gtf2i* in mOLs. However, the precise mechanisms by which Tfi-i modulates the expression of myelination-related genes remain inadequately understood. To unravel the nature of its regulation, it is crucial to identify the specific DNA loci to which Tfi-i directly binds in mOLs. By pinpointing these binding sites, we can gain valuable insights into Tfi-i's mode of action in mOLs. Consequently, in the next phase of our research, we aimed to comprehensively map the genome-wide Tfi-i binding sites and enrichment levels, specifically in mOLs.

To achieve this, mOLs derived from primary, fully differentiated, OL-enriched cell cultures were subjected to chromatin immunoprecipitation followed by sequencing (ChIP-seq), targeting Tfi-i. As expected, Tfi-i peaks were highly abundant in control compared to *Gtf2i*-KO samples (Supplementary Figure 10). Overall, 8429 unique genomic loci were identified as high-confidence Tfi-i binding sites in control samples (i.e. consensus peaks), with minimal overlap in *Gtf2i*-KO samples (Figure 6A and Supplementary Figure 10). Functional annotation analysis revealed that Tfi-i binding sites in control samples were predominantly enriched in introns and intergenic regions; intergenic (4568 peaks), introns (3314 peaks), exons (306 peaks), promoters (140 peaks), and transcription termination sites (TTS) (101 peaks) (Figure 6A). These data correlate well with our previous findings, where we identified cell-type-specific aberrant methylation profiles on enhancers in post-mortem brain tissue of individuals with WS^[20]. Combined, these data suggest that, in mOLs, Tfi-i predominantly binds to enhancers rather than promoters to regulate cell-type-specific gene programs. Interestingly, analysis of genomic sites in the vicinity of Tfi-i peaks (within a 200 bp window) revealed an enrichment of binding motifs for specific transcriptional

regulators, most notably is Sox10 (Figure 6B). Hence, it is plausible that Tfii-i could exhibit synergistic transcriptional activities with these regulators, Sox10 among them, pivotal in modulating myelination processes.

Gene ontology (GO) analysis of Tfii-i consensus peaks annotated to exons, promoters and TTSs revealed enrichment in pathways associated with DNA modification, negative regulation of gene expression, and brain development (Figure 6C). To determine whether the intergenic and intronic regions harboring Tfii-i peaks function as transcriptional REs in OL, we overlapped our ChIP-seq datasets with a previously published OL-specific predictive atlas of *cis* REs (cREs) in mice (CATlas)^[21]. Overall, Tfii-i consensus peaks aligned with 1352 unique OL-specific cREs and were then mapped to putative target genes using CATlas^[21]. GO analysis of these target genes showed enrichment in pathways associated with nervous system development, cell projection morphogenesis, locomotory behavior, negative regulation of myelination, and negative regulation of gene expression (Figure 6D). Among the high-confidence Tfii-i peaks, we observed relevant Tfii-i binding peaks at the *Mbp* and *Sox10* loci (Figure 6E and F). For *Mbp*, Tfii-i binding was observed at three different genomic loci; Exon 4 (out of 4), intron 2 (out of 6), and TTS (Figure 6E). Additionally, Tfii-i binding was detected upstream to *Mbp* at a distant intergenic locus (~25kb upstream to TSS, Figure 6E). Although we did not observe direct binding at the *Sox10* genomic loci, we found a high-confidence Tfii-i peak at a distant RE predicted to fold into the promoter region of *Sox10* (Figure 6F).

Our findings suggest that in mOLs, Tfii-i transcriptional regulation is primarily mediated through enhancers. Furthermore, in mOLs, Tfii-i is involved in the transcriptional program of the myelination process and binds to REs of genes associated with governing cellular structural properties. Importantly, these findings shed light on Tfii-i's transcriptional mode of action in mOLs, specifically its role in regulating key myelination genes such as *Sox10* and *Mbp*. These data may underline the mechanism of the observed altered interplay of myelin thickness and axonal diameters in the CC of *Gtf2i*-KO mice.'

These new data suggest that in mOLs, Tfii-i's mode of action is predominantly through interaction with active enhancers. Previous studies have shown that in neurons, Tfii-i regulates transcription mainly by binding to promoter sequences. This cell-type-

specific mode of action highlights the versatility and adaptability of Tfi-i in different cellular contexts.

To address the reviewer's question about Tfi-i functions in other cell types, we have inserted the following text into the discussion section to fill this gap in the original manuscript:

‘For example, while Tfi-i was shown to predominantly bind to promoters^[41], here we show that in mOLs Tfi-i primarily binds to active enhancers rather than promoters.’

And the revised Figure 6 is presented below, for convenience:

3. Line 101 – and Fig 2: what means “increased number of WM fibers”, more axons or higher percentage of myelinated axons? DTI may not be the right tool to measure myelination in mice, in which any histological approach is superior to MRI. At least it must be complemented with histological analyses.

Response:

We thank the reviewer for the valuable input. We agree that DTI may not be the most accurate tool to assess the exact number of axons or the percentage of myelinated axons. DTI measures reconstructed streamlines from fiber tracking, which we have termed ‘tracts’ in the revised manuscript. While DTI measures connectivity rather than direct myelination or WM properties, there is a correlation between DTI measurements and myelin-related properties.

Furthermore, it has been shown that the general pattern of connectivity from the DTI connectome and the Allen mouse connectivity atlas (performed by IHC tract-tracing) are similar^[49]. Additionally, DTI analysis is superior in rodents compared to humans due to significantly fewer crossing fibers, resulting in better segregation of tracts and higher accuracy.

To address this aspect, we clarified the terminology and the interpretation of DTI data in the revised manuscript, by adding the following clarification:

‘While DTI underlines connectivity patterns rather than directly measuring myelination or WM properties, there is a correlation between the number of reconstructed streamlines (tracts) and myelin-related properties^[50, 51].’

To avoid confusion, we removed the term "number of WM fibers" and now define the number of constructed streamlines by MRI-DTI as the "number of tracts." In the revised manuscript, we describe the MRI-DTI results as showing an "enhanced connectivity pattern," ensuring clarity and preventing misinterpretation regarding WM or myelin properties.

Additionally, we strengthened these results with EM of the CC (Figure 2A-E, presented also below), which shows a moderate increase in myelin thickness in *Gtf2i*-KO mice.

Figure 2

We also provide additional molecular results, such as an increase in *Mbp* transcript and protein expression levels in the myelin sheath itself, along the development (Figure 4, presented below). This was done by qPCR, WB and whole proteome proteomics advanced analysis on biochemically-enriched myelin fraction.

Last, the myelination ultrastructural properties of the PNS was also assessed and reported in our revised manuscript (Figure 2J-M, presented below), to better represent the myelination properties in the mutant mouse and describe a more comprehensive overview of the myelination impact as a result of *Gtf2i* deletion in myelinating cells.

Figure 2

4. Fig1A (genotyping) can go into supplementary data, and Fig 1C and D are not really helpful. A quantitation of the observation in Fig1B would be helpful.

Response:

We agree with the reviewer's viewpoint and thank for the technical comment. We adjusted the order of the figures according to the reviewer's comments. Specifically, we moved Figure 1A (genotyping) to the supplementary data. Additionally, we removed Figure 1C and 1D, as they were indeed not contributing significantly to the narrative.

To address the need for quantitative data, we added a new set of representative images showing a higher number of mature OLs and included quantification of the immunofluorescence images to verify *Tfii-i* deletion from mOLs (Figure 1A-B, also presented below).

Additionally, the efficiency of *Gtf2i* deletion from mature OLs was further quantified by showing decreased Tfii-i expression levels in the crude myelin fraction derived from the cortex of *Gtf2i*-KO mice, as compared to controls (Supplementary Figure B-C, presented also below).

We have also made minimal, technical alterations to the text to describe these results. The revised text is attached below: ‘To validate the genetic deletion of *Gtf2i* selectively from mOLs (Figure 1A) and the absence of the protein product, Tfii-i, in mOLs, we performed immunohistochemical staining, showing decrease in Tfii-i expression levels in CC1⁺ cells in *Gtf2i*-KO mice and intact expression in littermate controls (*Gtf2i*^{fl/fl}, *Cnp-Cre*^{-/-}, hereby termed control mice) (Figure 1B, Supplementary Figure 1). Tfii-i was previously shown to be expressed and have different roles in the nucleus and in the cytoplasm^[2, 3]. However, its expression level and roles were never directly assessed in the myelin fraction. Thus, we characterized its expression in the myelin fraction purified from whole cortex to study whether Tfii-i is specifically localized to myelin. While we found no evidence for Tfii-i expression in the purified myelin fraction (Supplementary figure 1B), Tfii-i levels in the crude myelin fraction, which may be partly contaminated with cellular and nuclear debris from both OLs and axons^[4], were decreased in *Gtf2i*-KO mice, as compared to controls (Supplementary figure 1C). As such, we showed that Tfii-i is expressed in OLs and validated its deletion from mOLs in *Gtf2i*-KO mice.’

5. Fig 3: the statistics of the g ratio measurement is not clear. How many mice were used and what is the N number? I have the impression the authors used the number of myelinated axons rather than the number of mice.

Response:

We thank the reviewer for pointing out the unclarity of this analysis. The N for the P30 EM experiments is 3 mice per genotype. Regarding the EM of the CC, the analysis was performed on the population of axons that were examined (control – 231 axons, *Gtf2i*-KO 226 axons, Kolmogorov-Smirnov test). While this may be an unorthodox method of analysis for this type of data, it enables the assessment of moderate alterations in axonal and myelin parameters. As we agree with the reviewer’s excellent comment, we have altered the text in the revised manuscript, and describe the altered interplay between axonal diameter and myelin thickness in the CC of *Gtf2i*-KO mice, compared to controls, rather than use the term ‘hypermyelination’.

To strengthen these data, we’ve also added an additional timepoint, postnatal day 90 (P90). These results indicate that while the increased axonal diameter does not persist through adulthood in *Gtf2i*-KO mice (supplementary figure 3D), the moderate effect of *Gtf2i* deletion from mOLs on myelin thickness persists through adulthood (supplementary figure 3E).

In addition, to elucidate the full scope of the effects of *Gtf2i* deletion on myelination, we assessed myelin properties in the tibial branch of the sciatic nerve (SN) of the PNS by EM (figure 2J-M). Our results indicate that the deletion of *Gtf2i* from SCs results in hypermyelination of the PNS, as the *g*-ratio values of *Gtf2i*-KO mice axons of the SN are decreased, compared to controls (figure 2K-L).

We've also incorporated this text in the manuscript to describe the observed hypermyelination of the SN: 'Since the *Cnp* promoter is also active in SCs of the PNS, we examined the myelin ultrastructure in the tibial branch of the SN of P30 mice by TEM (Figure 2J). The *g*-ratio values of *Gtf2i*-KO mice axons were significantly lower compared to controls (Figure 2K-L), indicating a hypermyelination phenotype in the PNS of *Gtf2i*-KO mice, while axonal diameters remained unchanged (Figure 2M).'

Additionally, we've also added an additional timepoint for this assessment, showing similar results in older mice (P90), suggesting this hypermyelination is long-lasting (supplementary figure 3G-H).

Overall, these data suggest that *Gtf2i* deletion led to altered interplay and moderate increase in myelin thickness in the CNS and hypermyelination of the PNS, with these effects lasting throughout adulthood.

Please also note that in the original manuscript we mistakenly presented the overall axonal diameter (inner diameter + the myelin sheath) in the figures related EM of the CC. We apologize for this unfortunate mistake and have now corrected it in the revised manuscript and present the inner diameter values. This, however, does not drastically change the results and conclusions drawn from these data. Hence, we still observe increase axonal diameter in our mutant mice, with moderate increase in myelin thickness. The corrected version of the figure and text is hereby attached below:

‘While *g*-ratio values in the overall axonal population of the CC were not significantly different, the interplay between axonal diameter and *g*-ratio values differed significantly between *Gtf2i*-KO mice and controls (Figure 2B). Specifically, the slope representing this interplay is more gradual in *Gtf2i*-KO mice axons, indicating that the *g*-ratio values of *Gtf2i*-KO mice axons are less affected by axonal diameter, compared to controls. In addition, the diameters of *Gtf2i*-KO mice axons were significantly increased, as compared to controls (Figure 2C, dashed vertical line indicates the median axon diameter of the control group, 0.53 μ m). Axons with larger diameters, as seen in the CC of *Gtf2i*-KO mice, are typically wrapped with a thicker myelin sheath^[5]. Accordingly, *Gtf2i*-KO mice axons demonstrated thicker myelin sheath, compared to controls (Figure 2D).’

6. Lines 285 and 342: I find the term “master regulator” too strong for a protein that is not specifically involved in myelination (compare e.g. to the Akt-mTOR pathway which also hardly a master regulator of myelination). Likewise, Fig 9 is not reflecting the biology of a widely expressed transcription factor.

Response:

We thank the reviewer for the insightful comment. Our initial use of the term "master regulator" was intended to highlight that Tfii-i regulates multiple myelin-related genes. However, we agree that this term may be exaggerated, and thus we have revised the phrasing of Tfii-i as a ‘master regulator’ throughout the text and revised our text to tone down our claims.

Examples of alterations in original text meant to tone down our claims:

1. Deleted line 31 (previously last line in abstract): ‘...possibly underlying a new therapeutic approach toward white matter pathologies.’
2. Previous claim (p. 17, line 285): ‘In the present study, we identified Tfii-i as a novel master regulator of the myelin transcriptome, acting as a transcription inhibitor of *Mbp* and *Sox10*. Moreover, we characterized a new mouse model of CNS hypermyelination with improved physiological and behavioral phenotypes.’

Revised claim (p. 14, line 387, first line in discussion): ‘In the present study, we identified Tfii-i as a novel regulator of the myelin transcriptome, acting as a transcription mediator of *Mbp* and *Sox10*, specifically in mOLs.’

3. Previous claim (p. 19, line 348): ‘our findings nonetheless provide an important basis for a potential new therapeutic target for WM disorders and conditions in which CNS re-myelination is required.’

Revised claim (p. 16, line 453, last two lines in discussion): ‘*Gtf2i* absence from mOLs promoted cellular, molecular, and structural changes, resulting in alterations of myelination in the CNS and PNS. Finally, we provide evidence that Tfii-i’s mode of action in mOLs is cell-specific, and shed light on its regulatory targets in mOLs.’

In response to the reviewer's request, we have also modified the graphical abstract (previously Figure 9, now Figure 7). Given that the mode of action of *Tfii-i* is still not fully understood (as discussed in our response to comment #2), we believe the modified graphical abstract better emphasizes the key points of *Tfii-i*'s regulatory role in OLs.

The corrected figure is attached below, for the convenience of the reviewer:

7. Fig 5: I could not find any information as to the number of mice used in the behavioral tests. What age was used?

Response:

We thank the reviewer for pointing out the missing information. We have now ensured that all statistical parameters are readily available throughout the manuscript. Specifically, the number of mice (N) used in the behavioral tests are as follows:

- Rotarod test: 19 control and 15 *Gtf2i*-KO mice.
- Open field test: 15 control and 14 *Gtf2i*-KO mice.
- Elevated zero maze: 13 control and 19 *Gtf2i*-KO mice.

All behavioral tests were performed on P30 (one-month-old) mice.

Additionally, to strengthen the claim of increased anxiety in *Gtf2i*-KO mice, we performed an elevated zero maze (EZM) experiment. We observed moderately increased anxiety in the mutant mice compared to controls (Figure 3D-E, also presented below). These data align with the open field results, further supporting our claim that *Gtf2i*-KO mice exhibit moderately increased anxiety.

The graphs related to the EZM data are attached below:

The text to describe the EZM results in the manuscript is as follows: 'To further examine anxiety-like behavior, control and *Gtf2i*-KO mice were subjected to the elevated zero maze (EZM) test. The results showed a trend towards higher anxiety levels in *Gtf2i*-KO mice, indicated by their spending less time in the open arms compared to controls (Figure 3D-E, mean control 249.1s±19.62s, mean *Gtf2i*-KO 200.3s±16.59s).

Together, these data suggest that *Gtf2i*-KO mice possess improved motor coordination and moderately increased anxiety-like behavior, as compared to controls.'

8. Is the myelination of the peripheral nervous system affected? *CnpCre* should target the gene also in Schwann cells.

Response:

The reviewer raised an important point, prompting us to thoroughly investigate this issue. Indeed, *Cnp* promoter is active in the PNS, specifically in myelinating Schwann cells (SCs). To determine whether the absence of *Gtf2i* in SCs affects myelin properties in the PNS, we assessed the myelin properties in the tibial branch of the sciatic nerve (SN) using EM (Figure 2J, presented below). Our results show that the deletion of *Gtf2i* in SCs leads to hypermyelination in the PNS, as indicated by decreased *g*-ratio values in the axons of *Gtf2i*-KO mice compared to controls (Figure 2K-L, presented below).

Figure 2

The following text was added to the revised manuscript to communicate these new finding: ‘Since the *Cnp* promoter is also active in SCs of the PNS, we examined the myelin ultrastructure in the tibial branch of the SN of P30 mice by TEM (Figure 2J). The *g*-ratio values of *Gtf2i*-KO mice axons were significantly lower compared to controls (Figure 2K-L), indicating a hypermyelination phenotype in the PNS of *Gtf2i*-KO mice, while axonal diameters remained unchanged (Figure 2M).’

We also added an additional timepoint for this assessment, showing similar results in older mice (P90), suggesting that this hypermyelination is long-lasting (Supplementary Figure 3G-H, presented below).

Minor points:

1. Line 129: what means “proper development”?

Response:

We thank the reviewer for pointing out this unclear description. We’ve corrected the text, changing the word ‘proper’ to ‘neurotypical’.

2. Line 145: 1 month old mice?

Response:

We thank the reviewer for their attention to this confusing description. We’ve redefined the ‘1-month old’ age group to ‘postnatal day 30’ (or P30) along the text.

3. Line 150: note that g ratios are not used to describe “structural integrity” of myelin, merely myelin sheath thickness.

Response:

We thank the reviewer’s keen comment. We’ve corrected the text according to this note.

4. Line 301: what is the relevance of “increased cell surface area”? A surrogate marker for myelination? That would be misleading.

Response:

The reviewer raises an important point that requires clarification. While increased OL cell surface area is not a direct marker of increased myelination, it suggests that Tfii-i negatively regulates critical cellular processes necessary for myelination, such as process extension and outgrowth. We agree with the reviewer that the original description in the text was confusing. Therefore, we have revised the text to clarify this interpretation, supported by a relevant reference. This addition to the text is presented here as well:

‘While not a direct marker of myelination, the increased cell surface area may suggest that, in OLs, *Gtf2i* regulates process extension and outgrowth, essential for myelination^[52].’

Reviewer #3 (Remarks to the Author):

The present manuscript describes a cell-autonomous role for the transcription factor *Gtf2i* in oligodendrocytes. Deletion of *Gtf2i* from oligodendrocyte lineage cells, using CNPCre, resulted in hypermyelination in the absence of changes in oligodendrocyte density, increased functional connectivity, and increased axon diameter in the corpus callosum. These physiological changes were accompanied by changes in motor coordination and anxiety-like behavior. Mechanistically, the authors found that *Gtf2i* negatively regulates expression levels of *Sox10* and *MBP*. Together, these data support the idea that *Gtf2i* cell-autonomously regulates the extent of myelination in oligodendrocytes.

Overall, this study is well-designed and in addition to establishing *Gtf2i* as a novel negative regulator of myelination, there are several exciting observations reported here. In particular, the hypermyelination without increases in oligodendrocyte cell density, the increase in axon density resulting from an oligodendrocyte-specific manipulation, and the increase in functional connectivity are noteworthy.

Response:

We thank the reviewer for the time and effort spent assessing our manuscript. We have carefully considered the reviewer's comments and have added additional data to support our claims. As a result, we believe the manuscript has improved considerably.

1. One major question that remains is whether these phenotypes are the result of accelerated myelination, or whether the hypermyelination persists into adulthood. This question could be addressed with the addition of a later time point (>3mo) in a couple of key quantifications.

Response:

We thank the reviewer for the valuable input. Given that *Gtf2i* and *Tfii-i* expression levels change during development and throughout life^[19], we fully agree that the reviewer raises valid and important concerns. To address these concerns, we added a later time point at 3 months old (P90) to our revised manuscript. To ensure a comprehensive characterization of these P90 mice, we performed cellular, ultrastructural, and molecular experiments, as follows:

- **Myelin ultrastructural properties:** We assessed myelin properties in the CC of P90 mice (Supplementary Figure 3A-E, presented also below). Our observations indicated increased myelin thickness in the CC of P90 *Gtf2i*-KO mice compared to controls.

- **OL cellular properties:** We evaluated OLs differentiation in the cortex and CC of P90 mice using immunofluorescence studies, which showed unchanged OL numbers in *Gtf2i*-KO mice compared to controls (Supplementary Figure 8, presented also below).

- Key myelin transcript and protein expression: We assessed key myelin transcript and protein expression levels in P90 mice (Figure 4G-K, presented also below), revealing similar results to the P30 data. Specifically, Mbp and Mog protein expression levels were elevated in the cortical pure myelin fraction of P90 mutant mice compared to controls. Furthermore, the mRNA expression levels of *Sox10*, *Mbp* and *Mog* were elevated in the cortex of P90 *Gtf2i*-KO mice, compared to controls.

These additional data support the notion that the hypermyelination phenotype persists into adulthood, rather than being solely the result of accelerated myelination during development. We believe this extended analysis provides a more complete understanding of the temporal dynamics of *Gtf2i* deletion effects on myelination.

We incorporated the following new text in the revised manuscript to describe these results: ‘*Gtf2i* and *Tfii*-i’s expression levels change across different developmental stages^[19]. To assess whether these alterations persist into adulthood, we examined the pure myelin fraction derived from the cortices of P90 mice. Our analysis showed increased protein expression levels of *Mbp* (Figure 4G, mean control 1±0.027, mean *Gtf2i*-KO 1.164±0.066) and *Mog* (Figure 4H, mean control 1±0.035, mean *Gtf2i*-KO 1.217±0.067). Additionally, we examined mRNA expression levels of *Sox10* (Figure 4I, mean control 1±0.102, mean *Gtf2i*-KO 1.279±0.063), *Mbp* (Figure 4J, mean control 1±0.09, mean *Gtf2i*-KO 1.34±0.071), and *Mog* (Figure 4K, mean control 1±0.047, mean *Gtf2i*-KO 1.328±0.051) in the cortex of P90 *Gtf2i*-KO mice, revealing increased transcript expression levels of these genes compared to controls.’

- **PNS Myelination Properties:** We added EM data of the tibial branch of the sciatic nerve to assess PNS myelination properties in P30 *Gtf2i*-KO mice, as requested by the reviewers (Figure 2J-M, also see below). To gain a comprehensive understanding of the myelination properties in P90 mice, we also assessed the same tissue at the P90 timepoint (Supplementary Figure 3F-I). The results for PNS myelin properties are consistent at both time points, indicating a persistent hypermyelination phenotype in our mutant mice.

EM of the SN – P90

Overall, we observed similar myelin-related cellular, ultrastructural, and molecular properties in P90 mutant mice compared to P30 mice. These data suggest that the hypermyelination phenotype persists into adulthood.

Please note that in the original manuscript, we mistakenly presented the overall axonal diameter (inner diameter + myelin sheath) in the figures related to EM of the CC. The revised manuscript now presents the corrected inner diameter values. This correction does not drastically change the results and conclusions drawn from these data. We still observe an increased axonal diameter in our mutant mice, along with a moderate increase in myelin thickness. The corrected version of the figure and text is attached below:

‘While g -ratio values in the overall axonal population of the CC were not significantly different, the interplay between axonal diameter and g -ratio values differed significantly between *Gtf2i*-KO mice and controls (Figure 2B). Specifically, the slope representing this interplay is more gradual in *Gtf2i*-KO mice axons, indicating that the g -ratio values of *Gtf2i*-KO mice axons are less affected by axonal diameter, compared to controls. In addition, the diameters of *Gtf2i*-KO mice axons were significantly increased, as compared to controls (Figure 2C, dashed vertical line indicates the median axon diameter of the control group, 0.53 μm). Axons with larger diameters, as seen in the CC of *Gtf2i*-KO mice, are typically wrapped with a thicker myelin sheath^[5]. Accordingly, *Gtf2i*-KO mice axons demonstrated thicker myelin sheath, compared to controls (Figure 2D).’

2. Ideally, the authors would also quantify oligodendrocyte differentiation at an earlier timepoint and in vitro to confirm the theory that *Gtf2i* exclusively regulates myelination and not differentiation.

Response:

We agree with the reviewer that the suggested experiments will strengthen our claim of a cell-autonomous mechanism of *Tfii-i* regulation in OLs. Therefore, we performed differentiation experiments in P14 mice (Supplementary Figure 8A-B, also presented below) and in OL-enriched primary cell culture (Supplementary Figure 9E-F, also presented below). Both experiments showed unchanged OL differentiation in mutant cells, further supporting the theory that the observed altered myelination properties in *Gtf2i*-KO mice (as were detailed in previous responses and are also detailed later in this response) are likely due to a cell-autonomous mechanism unrelated to OL differentiation.

Supplementary Figure 8A-B

E

Supplementary Figure 9E-F

F

To strengthen the molecular proof of myelin-related alterations in *Gtf2i*-KO mice, we have conducted a comprehensive proteome analysis of the myelin sheath (biochemically-enriched myelin fraction) from the hemispheres of *Gtf2i*-KO and control mice (Figure 4A, presented below). This analysis revealed increased expression levels of *Mbp* and *Mog*, as well as a decrease in the expression level of *Gpr37*. *Gpr37* is known to be a negative regulator of myelination, and its deletion results in precocious myelination^[14].

Overall, these findings provide a broader overview of *Gtf2i*-induced changes in the myelin sheath and support the involvement of these specific proteins in the observed phenotype. We acknowledge the limitations of focusing solely on *Sox10* and *Mbp* based on non-oligodendrocyte ChIP-seq data and have thus expanded our approach to include proteomic analysis for a more comprehensive understanding.

The data from our proteome analysis are visualized below, providing stronger evidence for the molecular changes induced by *Gtf2i* deletion.

The text to describe these data is as follows: ‘To assess how *Tfii-i* absence from OLS alters myelin properties on a molecular level, we examined the biochemically-enriched myelin fraction^[15] purified from the brain hemisphere of P30 mice. The protein composition of the myelin fraction was assessed by label-free quantitative mass spectrometry^[15]. Proteome

analysis revealed altered protein abundance of several key myelin proteins (Figure 4A and Supplementary Figure 5). Specifically, the relative abundance of Mog and Mbp was significantly elevated in myelin purified from *Gtf2i*-KO mice (Figure 4A), while the abundance of Gpr37 was reduced. Mbp is an abundant constituent of compact myelin, its expression being a rate-limiting step in myelination^[16]. Mog is a marker for the abaxonal myelin membrane^[17]. Gpr37 has been demonstrated as a negative regulator of myelination^[14]. Overall, the altered myelin proteome observed in *Gtf2i*-KO mice suggests a molecular hypermyelination phenotype in the absence of Tfi-i.'

We also provide additional molecular results, such as an increase in *Mbp* transcript and protein expression levels in the myelin sheath itself, along the development (Figure 4, presented below). This was done by qPCR, WB and whole proteome proteomics advanced analysis on biochemically-enriched myelin fraction.

3. Minor comment - the authors should quantify whether the percentage of myelinated axons is altered in the CC of control and *Gtf2i*-KO mice.

Response:

We agree with the reviewer that quantifying the percentage of myelinated axons in the CC of control and *Gtf2i*-KO mice is important for providing a comprehensive picture of axon myelination following *Gtf2i* deletion. We assessed the number and ratio of myelinated and unmyelinated axons in the CC of P30 *Gtf2i*-KO and control mice. Our analysis revealed an unchanged ratio (Figure 2E, also presented below) and axon numbers (data not shown). This further supports the claim that *Tfii-i*'s mode of action is cell-autonomous, rather than related to cell numbers.

Relevant text incorporated in the revised manuscript is:

‘The ratio of myelinated to unmyelinated axons was similar between *Gtf2i*-KO mice and controls (Figure 2E, mean control 0.266 ± 0.02 , mean *Gtf2i*-KO 0.26 ± 0.029)’.

References:

1. Barak, B., et al., *Neuronal deletion of Gtf2i, associated with Williams syndrome, causes behavioral and myelin alterations rescuable by a remyelinating drug*. Nature Neuroscience, 2019. **22**(5): p. 700-708.
2. Caraveo, G., et al., *Action of TFII-I Outside the Nucleus as an Inhibitor of Agonist-Induced Calcium Entry*. Science, 2006. **314**(5796): p. 122-125.
3. Roy, A.L., *Transcription Factor TFII-I Conducts a Cytoplasmic Orchestra*. ACS Chemical Biology, 2006. **1**(10): p. 619-622.
4. Larocca, J.N. and W.T. Norton, *Isolation of Myelin*. Current Protocols in Cell Biology, 2006. **33**(1): p. 3.25.1-3.25.19.
5. Pease-Raissi, S.E. and J.R. Chan, *Building a (w)rapport between neurons and oligodendroglia: Reciprocal interactions underlying adaptive myelination*. Neuron, 2021. **109**(8): p. 1258-1273.
6. Giese, K.P., et al., *Mouse *P₀* gene disruption leads to hypomyelination, abnormal expression of recognition molecules, and degeneration of myelin and axons*. Cell, 1992. **71**(4): p. 565-576.
7. McKenzie, I.A., et al., *Motor skill learning requires active central myelination*. Science, 2014. **346**(6207): p. 318-322.
8. Xiao, L., et al., *Rapid production of new oligodendrocytes is required in the earliest stages of motor-skill learning*. Nature Neuroscience, 2016. **19**(9): p. 1210-1217.
9. De Zeeuw, C.I. and M.M. Ten Brinke, *Motor Learning and the Cerebellum*. Cold Spring Harbor Perspectives in Biology, 2015. **7**(9).
10. Lopes, S., et al., *Absence of Tau triggers age-dependent sciatic nerve morphofunctional deficits and motor impairment*. Aging Cell, 2016. **15**(2): p. 208-216.
11. Chen, P., et al., *Collagen VI regulates peripheral nerve myelination and function*. The FASEB Journal, 2014. **28**(3): p. 1145-1156.
12. Xiu, Y., et al., *Ultrastructural abnormalities and loss of myelinated fibers in the corpus callosum of demyelinated mice induced by cuprizone*. Journal of Neuroscience Research, 2017. **95**(8): p. 1677-1689.
13. Franco-Pons, N., et al., *Behavioral deficits in the cuprizone-induced murine model of demyelination/remyelination*. Toxicology Letters, 2007. **169**(3): p. 205-213.
14. Yang, H.-J., et al., *G protein-coupled receptor 37 is a negative regulator of oligodendrocyte differentiation and myelination*. Nature Communications, 2016. **7**(1): p. 10884.
15. Erwig, M.S., et al., *Myelin: Methods for Purification and Proteome Analysis*, in *Oligodendrocytes: Methods and Protocols*, D.A. Lyons and L. Kegel, Editors. 2019, Springer New York: New York, NY. p. 37-63.
16. Readhead, C., et al., *Expression of a myelin basic protein gene in transgenic shiverer mice: Correction of the dysmyelinating phenotype*. Cell, 1987. **48**(4): p. 703-712.
17. Brunner, C., et al., *Differential Ultrastructural Localization of Myelin Basic Protein, Myelin/Oligodendroglial Glycoprotein, and 2',3'-Cyclic Nucleotide 3'-Phosphodiesterase in the CNS of Adult Rats*. Journal of Neurochemistry, 1989. **52**(1): p. 296-304.
18. Stolt, C.C., et al., *Terminal differentiation of myelin-forming oligodendrocytes depends on the transcription factor Sox10*. Genes & Development, 2002. **16**(2): p. 165-170.
19. Nir Sade, A., et al., *Neuronal Gtf2i deletion alters mitochondrial and autophagic properties*. Communications Biology, 2023. **6**(1): p. 1269.
20. Trangle, S.S., et al., *In individuals with Williams syndrome, dysregulation of methylation in non-coding regions of neuronal and oligodendrocyte DNA is associated with pathology and cortical development*. Molecular Psychiatry, 2023. **28**(3): p. 1112-1127.

21. Li, Y.E., et al., *An atlas of gene regulatory elements in adult mouse cerebrum*. Nature, 2021. **598**(7879): p. 129-136.
22. Osso, L.A. and J.R. Chan, *A surprising role for myelin in Williams syndrome*. Nat Neurosci, 2019. **22**(5): p. 681-683.
23. Roy, A.L., et al., *Cooperative interaction of an initiator-binding transcription initiation factor and the helix–loop–helix activator USF*. Nature, 1991. **354**(6350): p. 245-248.
24. Crespi, B.J. and P.L. Hurd, *Cognitive-behavioral phenotypes of Williams syndrome are associated with genetic variation in the GTF2I gene, in a healthy population*. BMC Neuroscience, 2014. **15**(1): p. 127.
25. Mervis, C.B., et al., *Duplication of GTF2I results in separation anxiety in mice and humans*. Am J Hum Genet, 2012. **90**(6): p. 1064-70.
26. Morris, C.A., et al., *GTF2I hemizygosity implicated in mental retardation in Williams syndrome: Genotype–phenotype analysis of five families with deletions in the Williams syndrome region*. American Journal of Medical Genetics Part A, 2003. **123A**(1): p. 45-59.
27. Sakurai, T., et al., *Haploinsufficiency of Gtf2i, a gene deleted in Williams Syndrome, leads to increases in social interactions*. Autism Research, 2011. **4**(1): p. 28-39.
28. López-Tobón, A., et al., *GTF2I dosage regulates neuronal differentiation and social behavior in 7q11.23 neurodevelopmental disorders*. Science Advances, 2023. **9**(48): p. eadh2726.
29. Dai, L., et al., *Is it Williams syndrome? GTF2IRD1 implicated in visual-spatial construction and GTF2I in sociability revealed by high resolution arrays*. Am J Med Genet A, 2009. **149a**(3): p. 302-14.
30. Antonell, A., et al., *Partial 7q11.23 deletions further implicate GTF2I and GTF2IRD1 as the main genes responsible for the Williams-Beuren syndrome neurocognitive profile*. J Med Genet, 2010. **47**(5): p. 312-20.
31. Malenfant, P., et al., *Association of GTF2I in the Williams-Beuren Syndrome Critical Region with Autism Spectrum Disorders*. Journal of Autism and Developmental Disorders, 2012. **42**(7): p. 1459-1469.
32. Tassabehji, M., et al., *GTF2IRD1 in Craniofacial Development of Humans and Mice*. Science, 2005. **310**(5751): p. 1184-1187.
33. Gagliardi, C., et al., *Unusual cognitive and behavioural profile in a Williams syndrome patient with atypical 7q11.23 deletion*. Journal of Medical Genetics, 2003. **40**(7): p. 526-530.
34. Karmiloff-Smith, A., et al., *Using case study comparisons to explore genotype-phenotype correlations in Williams-Beuren syndrome*. Journal of Medical Genetics, 2003. **40**(2): p. 136-140.
35. Swartz, J.R., et al., *A Common Polymorphism in a Williams Syndrome Gene Predicts Amygdala Reactivity and Extraversion in Healthy Adults*. Biological Psychiatry, 2017. **81**(3): p. 203-210.
36. Kim, K., et al., *Association-heterogeneity mapping identifies an Asian-specific association of the GTF2I locus with rheumatoid arthritis*. Scientific Reports, 2016. **6**(1): p. 27563.
37. Li, Y., et al., *A genome-wide association study in Han Chinese identifies a susceptibility locus for primary Sjögren's syndrome at 7q11.23*. Nature Genetics, 2013. **45**(11): p. 1361-1365.
38. Kim, I.-K., et al., *Mutant GTF2I induces cell transformation and metabolic alterations in thymic epithelial cells*. Cell Death & Differentiation, 2020. **27**(7): p. 2263-2279.
39. Petrini, I., et al., *A specific missense mutation in GTF2I occurs at high frequency in thymic epithelial tumors*. Nature Genetics, 2014. **46**(8): p. 844-849.
40. Roy, A.L., et al., *An alternative pathway for transcription initiation involving TFII-I*. Nature, 1993. **365**(6444): p. 355-359.
41. Roy, A.L., *Biochemistry and biology of the inducible multifunctional transcription factor TFII-I: 10years later*. Gene, 2012. **492**(1): p. 32-41.

42. Crusselle-Davis, V.J., et al., *Recruitment of coregulator complexes to the β -globin gene locus by TFII-I and upstream stimulatory factor*. The FEBS Journal, 2007. **274**(23): p. 6065-6073.
43. Enkhmandakh, B., et al., *Essential functions of the Williams-Beuren syndrome-associated TFII-I genes in embryonic development*. Proceedings of the National Academy of Sciences, 2009. **106**(1): p. 181-186.
44. Blomen, V.A., et al., *Gene essentiality and synthetic lethality in haploid human cells*. Science, 2015. **350**(6264): p. 1092-1096.
45. Singh, A., et al., *Transcription factor TFII-I fine tunes innate properties of B lymphocytes*. Frontiers in Immunology, 2023. **14**.
46. Ren, X., et al., *Direct Interactions of OCA-B and TFII-I Regulate Immunoglobulin Heavy-Chain Gene Transcription by Facilitating Enhancer-Promoter Communication*. Molecular Cell, 2011. **42**(3): p. 342-355.
47. Sacristán, C., et al., *Characterization of a novel interaction between transcription factor TFII-I and the inducible tyrosine kinase in T cells*. European Journal of Immunology, 2009. **39**(9): p. 2584-2595.
48. Hong, M., et al., *Transcriptional Regulation of the Grp78 Promoter by Endoplasmic Reticulum Stress: ROLE OF TFII-I AND ITS TYROSINE PHOSPHORYLATION**. Journal of Biological Chemistry, 2005. **280**(17): p. 16821-16828.
49. Assaf, Y., et al., *Conservation of brain connectivity and wiring across the mammalian class*. Nature Neuroscience, 2020. **23**(7): p. 805-808.
50. Blumenfeld-Katzir, T., et al., *Diffusion MRI of Structural Brain Plasticity Induced by a Learning and Memory Task*. PLOS ONE, 2011. **6**(6): p. e20678.
51. Sampaio-Baptista, C., et al., *Motor Skill Learning Induces Changes in White Matter Microstructure and Myelination*. The Journal of Neuroscience, 2013. **33**(50): p. 19499-19503.
52. Bercury, Kathryn K. and Wendy B. Macklin, *Dynamics and Mechanisms of CNS Myelination*. Developmental Cell, 2015. **32**(4): p. 447-458.

Rebuttal letter – Reviewer 1

In their manuscript, Levy and colleagues analyze the *Gtf2i*-encoded transcription factor Tfii-i in oligodendrocyte differentiation and identify a role in myelination, presumably via its impact on Sox10 and Mbp expression. Despite the fact that substantial amounts of data were added during the revision, my enthusiasm for the study remains limited.

Response:

We thank the reviewer for their time and attention spent reviewing our manuscript. We appreciate the reviewer's comments and believe that the revised manuscript is greatly improved thanks to their insights. Please find below a point-by-point response letter to all the concerns that were raised.

1. While I do not doubt the presence of myelin abnormalities in the *Gtf2i* knockouts, the myelination phenotype in the corpus callosum is very subtle. According to the authors, the g-ratio is unaltered but there are more axons with larger diameter and as a consequence with thicker myelin. From my experience, axons with different diameter are not equally distributed throughout the corpus callosum so that the phenotype described by the authors could be a sampling artifact. From the information provided in the Materials and Methods section, I cannot judge whether this can be excluded.

Response:

We thank the reviewer for their mindful comment. The reviewer is correct that axonal diameters are not homogenously distributed across the CC. To strengthen our findings and address concerns regarding a potential sampling artifact, we have conducted additional sets of experiments:

a. **We obtained new data utilizing MRI-DTI scans, which reveal a reduction in the number of WM tracts across the entire CC in *Gtf2i*-KO mice compared to controls.** These new findings reinforce our conclusions in several ways:

i. The WM fiber tracking in our MRI-DTI analysis covers the entire CC, thereby addressing the concern that our EM data from the midline CC may have been affected by sampling bias.

ii. When combined with our data showing that CC thickness remains unchanged between control and *Gtf2i*-KO mice (Figure 1F in the manuscript), the observed reduction in WM tracts across the CC in *Gtf2i*-KO mice supports the notion that axonal diameters may be increased in the CC of *Gtf2i*-KO mice, as demonstrated by our EM data. As such, these new findings complement our EM results from the midline CC, further substantiating the presence of myelination alterations in the CC of *Gtf2i*-KO mice.

b. **To further substantiate our findings of myelination abnormalities in the CC of *Gtf2i*-KO mice, we conducted an immunofluorescence (IF) assay targeting nodes of Ranvier (NoRs).** To do so, we stained 50µm sections with antibodies against Capsr and Nav1.6, focusing our analysis on the

midline CC at the same bregma level as the EM experiment, thereby directly complementing our previous data. **Our analysis revealed a significant decrease in the number of NoRs in the midline CC of *Gtf2i*-KO mice compared to controls.** These newly obtained data further reinforce our claim of altered myelination in the CC of *Gtf2i*-KO mice. Furthermore, previous studies have also shown that a reduced number of NoRs can correlate with enhanced myelination properties in the CNS^[1], while other studies have shown that an increased number of paranodes is associated with reduced conduction velocity^[2]. These findings collectively support the notion that myelination abnormalities in *Gtf2i*-KO mice may have functional consequences on axonal conduction properties.

These newly obtained data have been incorporated into the revised manuscript within the Results and Discussion sections and added as new panels in the figures. For the reviewer's convenience, the updated text and corresponding figure are provided below (with the new data in the figure labeled within a blue rectangle).

Results section:

'To better capture the heterogeneity of axonal and WM properties across the CC and investigate structural and myelination abnormalities in *Gtf2i*-KO mice, we performed MRI-DTI and quantified WM tracts across the entire CC using tractography analysis (Figure 2F). Our results show significantly lower number of WM tracts in the CC of *Gtf2i*-KO mice, as compared to controls (Figure 2G, mean control 1159.75 ± 83.637 , mean *Gtf2i*-KO 942.214 ± 50.016). Taken together with our finding of unchanged CC thickness in *Gtf2i*-KO mice (Figure 1F), these results are in accordance with our finding of increased axonal diameter in the CC of *Gtf2i*-KO mice (Figure 2C). Given that WM tract integrity and axonal properties are closely linked to myelination and conductivity, we further examined NoRs in the midline CC (Figure 2H, Supplementary figure 3). Alterations in NoR number and length have been associated with myelination dynamics and conductivity efficiency^[1, 3]. While NoR length in the midline CC was similar between *Gtf2i*-KO and control mice (Figure 2I, mean control $1.609 \pm 0.056 \mu\text{m}$, mean *Gtf2i*-KO $1.536 \pm 0.064 \mu\text{m}$), the number of NoRs per field of view (FOV) was significantly reduced in *Gtf2i*-KO mice compared to control mice (Figure 2J, mean control 20.812 ± 1.213 , mean *Gtf2i*-KO 15.25 ± 0.7).

Discussion section:

'Variations in NoRs number and length are associated with myelination dynamics and conductivity properties^[1, 3]. In a recent study, monocular deprivation was shown to result in reduced internode length and increased number of paranodes in the optic tract, resulting in reduction of conduction velocity^[2]. As such, the reduced number of NoRs may also contribute to the enhanced conduction properties observed in the CC of *Gtf2i*-KO mice. Homozygous deletion of *Cnp* was previously shown to reduce the number of NoRs in the optic nerve of adult mice, but not in 1-month-old mice, which exhibited a similar number of NoRs as controls^[4]. Thus, it is unlikely that the innate *Cnp* haploinsufficiency in our study accounts for the reduced NoR numbers observed in the CC of 1-month-old *Gtf2i*-KO mice compared to controls'.

Figure 2. Altered axonal and myelin properties with enhanced conduction characteristics in the CC of *Gtf2i*-KO mice. (A) Representative TEM images of axons from the CC of control and *Gtf2i*-KO mice. (B) Scatter plot of g -ratio values and their respective axon diameters. The dashed line indicates the median diameter of control mice axons ($0.53\mu\text{m}$), while the numbers on each side of the line represent the percentage of axons below (left) and above (right) this value for each genotype (simple linear regression, slopes $P=0.0045$). (C) Myelinated axon diameter distribution demonstrates significantly larger axonal diameters in *Gtf2i*-KO mice (Kolmogorov-Smirnov test, $P=0.0004$). (D) Myelin thickness of *Gtf2i*-KO axons is significantly increased, as compared to controls (Kolmogorov-Smirnov test, $P=0.0231$). (E) Myelinated to unmyelinated axons ratio is unchanged in *Gtf2i*-KO mice, compared to controls ($n=3$, Student's t -test, $P=0.86$). (F) Representative images of tractography analysis from the entire CC of control and *Gtf2i*-KO mice. (G) The CC of *Gtf2i*-KO mice presents with decreased number of WM tracts, as compared to controls ($n=12$ control, $n=14$ *Gtf2i*-KO, Student's t -test, $P=0.03$). (H) Representative image of NoR staining at the midline CC. NoRs are marked on the zoomed-in (right) image (white lines). Cells nuclei are stained with DAPI. (I) The midline CC of *Gtf2i*-KO mice presents with unchanged NoR length (Student's t -test, $P=0.428$) and (J) decreased number of NoRs per FOV, as compared to controls (Student's t -test, $P=0.0074$). (K) Illustration of *in-vivo* EPSP recordings experiment, electrodes placement in the CC and representative traces from recordings of both control and *Gtf2i*-KO mice. (L-N) Electrophysiological recordings from the CC (control $n=7$, *Gtf2i*-KO $n=7$), *in-vivo*. (L) Significantly shorter fEPSP latencies across the CC in *Gtf2i*-KO mice (Student's t -test, $P=0.0078$), with (M) significantly higher slope (mV/ms) values (Student's t -test, $P=0.0497$) compared to controls. (N) Average slope (mV/ms) values are significantly steeper in *Gtf2i*-KO mice across different stimulus intensities (Mixed-effects analysis, $P=0.042$). (B-D) control $n=3$, 231 axons, *Gtf2i*-KO $n=3$, 226 axons. (I-J) control $n=4$, 315 NoRs, *Gtf2i*-KO $n=4$, 301 NoRs. ns – non-significant, * $p < 0.05$, ** $p < 0.01$, *** $p < 0.001$.

Furthermore, we have expanded the Methods section to provide a more detailed description of the acquisition of the CC EM data. This includes the addition of an illustration specifying the exact location from which the data were obtained, along with a more comprehensive explanation of the methodology. The supplementary figure referenced above is attached below for the reviewer’s convenience:

Supplementary figure 3. Anatomic location of TEM and NOR experiments in the midline of the CC. **(A)** Midsagittal section with the CC and fornix highlighted (light purple). TEM samples were taken from the CC at the level of the fornix, marked as a black square in the midsagittal section. **(B)** Zoom-in image of the anatomic location from which TEM samples were taken. The numbers 1-3 delineates our TEM image acquisition strategy where three different locations along the rostro-caudal and dorso-ventral axis of the CC were quantified, per sample. **(C)** Coronal section taken from the same bregma and depicts the location in which images for NOR properties quantification were taken.

2. Additionally, control mice and conditional *Gtf2i* knockout mice differ by the presence/absence of the *Cnp-Cre*. Considering that *Cre* is knocked into the *Cnp* locus and disables *Cnp* expression from one allele, the *Gtf2i* knockouts express lower levels of the myelin protein *Cnp*. Have the authors convinced themselves that altered *Cnp* expression in the knockouts does not contribute to the observed alterations in myelin and axonal conductivity? There is evidence in the literature that altered *Cnp* levels have effects on *Mbp* (e.g. PMID: 9373033).

Response:

We thank the reviewer for the valuable comment. We acknowledge that *Cnp* expression is genetically reduced in *Gtf2i*-KO mice, as confirmed by our CNS myelin proteome data, which show significantly decreased *Cnp* protein levels in the myelin fraction of *Gtf2i*-KO mice, as expected. To address the reviewer's concerns, we highlight the following key points:

a. To specifically address the reviewer's concern and exclude the potential contribution of *Cnp* heterozygosity to the altered *Mbp* and *Sox10* levels, **we intentionally included an experiment in our original manuscript that is independent of the *Cnp-Cre* line.** Specifically, our *in vitro* experiments presented in Figure 6, provide further evidence that the observed changes in *Mbp* and *Sox10* expression are not influenced by *Cnp-Cre* expression. We demonstrated that *Mbp* and *Sox10* levels are increased in cultured OLs infected with a *Cre*-recombinase-expressing AAV, where *Gtf2i* was deleted exogenously, **without the involvement of the *Cnp-Cre* line. These findings strongly support that the phenotypes observed in our study result specifically from *Gtf2i* deletion in OLs and are not attributable to altered *Cnp* expression.**

b. The *Cnp-Cre* mouse line is widely used across many laboratories and is highly cited in the original publication describing its generation (with more than 1,000 citations) ^[5]. **This article reports no significant changes in lipid abundance, WM properties (based on *Mbp* immunostaining), or myelin sheath structure and spacing in homozygous *Cnp*-deficient mice at 2.5 months-of-age, compared to WT mice.** Furthermore, the morphology and microtubular network of OLs were unchanged. Since our study utilizes mice with heterozygous deletion of *Cnp* at f 1 or 3 months-of-age, it is highly unlikely that reduced *Cnp* expression contributes to the observed enhanced myelination phenotypes.

c. The reviewer references a study in which ***Cnp* was overexpressed at six times its normal levels** (PMID:9373033). **This study reported alterations in myelin ultrastructure, but the general *Mbp* pattern remained unchanged, and *Mbp* accumulation in compacted myelin was not affected.** While this does not entirely rule out an interaction between *Cnp* and *Mbp* expression, **it does not provide direct evidence that the reduced *Cnp* levels in our model impact *Mbp* upregulation** in a functionally meaningful way.

d. Several other studies using the *Cnp-Cre* line have reported **no significant changes in *Mbp* expression or other key myelin-related genes, or in the percentage of axons myelinated or degenerated between WT and *Cnp-Cre* (+/-) mice**, among them is Madsen et al (PMID:27147664)^[6]. Additionally, *Cnp-Cre* (+/-) mice showed normal performance on the rotarod test, while our model demonstrated improved motor behavior on this test^[6].

In contrast, one study did report a significant decrease in *Mbp* isoform expression relative to β -tubulin levels in 8-week-old *Cnp-cre* (+/-) mice in the cortex and the CC as compared to WT controls^[7]. This reduction in *Mbp* levels contrasts with the significantly increased *Mbp* expression observed in our mouse model, further supporting our evidence of enhanced myelination properties. Therefore, when considering the body of literature, these overall findings suggest that the heterozygous *Cnp* deletion inherent to the *Cnp-Cre* mouse line is unlikely to drive the phenotypes observed following *Gtf2i* deletion in OLs in our study.

3. In the revised version, the authors now show that Schwann cells and PNS myelin are also affected in the *Gtf2i* knockouts. In the PNS, there are g-ratio alterations. The phenotypic abnormalities in the PNS seem even stronger than in the CNS and may actually be the main factor for the improved motor coordination. It is important to have this information in the revised manuscript. Nevertheless, the analysis of the PNS phenotype is very preliminary, only mentioned in passing and hardly discussed in its implications. Although, I understand that the PNS phenotype is not the main focus of the current study, its current presentation is a bit odd.

Response:

We thank the reviewer for their insightful observation, which has significantly enhanced both the manuscript and the impact of our scientific findings. We fully acknowledge that the PNS phenotype was not sufficiently explored or discussed in the previous version of the manuscript. To address this, we have expanded our analysis of the PNS, incorporating additional assessments of myelin thickness, myelin abnormalities, Remak's bundles, and axonal properties.

These new data reveal novel PNS-related myelination abnormalities in our mouse model compared to controls, as detailed below in the revised text presented in the Results section.

In addition, to address the reviewer's suggestion regarding the potential behavioral implications of these PNS alterations, we have included a discussion in the revised Discussion section of our manuscript considering how PNS myelination changes may contribute to motor coordination, while CNS myelination deficits may be more closely linked to anxiety and social behavior.

These newly obtained data have been incorporated into the revised manuscript within the Results and Discussion sections and added as new panels in the figures. For the reviewer's convenience, the updated text and corresponding figure are provided below (with the new data in the figure labeled within a blue rectangle).

Results section:

Hypermyelination of the SN in *Gtf2i*-KO mice

‘Since the *Cnp* promoter is also active in SCs of the PNS, we examined the myelin ultrastructure in the tibial branch of the SN of P30 mice by TEM (Figure 3A). The *g*-ratio values of *Gtf2i*-KO mice axons were significantly lower compared to controls (Figure 3B-C), indicating a hypermyelination phenotype in the PNS of *Gtf2i*-KO mice. We specifically observed that small and medium caliber axons present with significantly smaller *g*-ratio values in *Gtf2i*-KO mice, as compared to controls (Figure 3D-E, mean control 0.618±0.01, 0.635±0.01, 0.664±0.007, mean *Gtf2i*-KO 0.568±0.004, 0.6±0.006, 0.658±0.011, for axon calibers of 0-2, 2-4, and 4-6 μm, respectively). *Gtf2i*-KO mice axons in the tibial branch of the SN also exhibited significantly increased myelin thickness compared to control mice (Figure 3F). Specifically, in accordance with *g*-ratio values, myelin thickness was significantly increased in small and medium caliber axons of *Gtf2i*-KO mice, as compared to controls (Figure 3G, mean control 0.508±0.022μm, 0.837±0.033μm, 1.226±0.029μm, mean *Gtf2i*-KO 0.61±0.014μm, 0.988±0.036μm, 1.239±0.065μm, for axon calibers of 0-2, 2-4, and 4-6 μm, respectively). Axonal diameters remained unchanged between *Gtf2i*-KO and control mice (Figure 3H). Decreased *g*-ratio values have previously been shown to be associated with abnormalities in myelin ultrastructure^[8], often resulting from deficits in myelin compaction. To assess whether myelin ultrastructural properties in the SN are affected by *Gtf2i* deletion we quantified the number of myelin deformations in myelinated axons of the SN, revealing unchanged prevalence of deformations between *Gtf2i*-KO and control mice (Figure 3I). These findings indicate that *Gtf2i* deletion in SCs results in hypermyelination of small- and medium-caliber axons in the tibial branch of the SN without affecting axonal diameter or myelin ultrastructure.

While this highlights the role of *Gtf2i* in regulating myelination in myelinating SCs, its influence on non-myelinating SCs remains unknown. The PNS comprises various types of non-myelinating SCs, including Remak SCs, which ensheath multiple small-caliber axons to form ‘Remak’s bundles’^[9] (Figure 3J). To assess whether *Gtf2i* deletion affects these unmyelinated axons, we quantified the number of the unmyelinated axons in the SN of *Gtf2i*-KO and control mice, and the ratio between myelinated and unmyelinated axons, revealing unaltered properties (Figure 3K).

These results indicate that *Gtf2i* deletion from SCs results in hypermyelination of the SN, without affecting axonal properties, or altering myelin integrity and compaction properties.’

Discussion section:

‘The observed hypermyelination in the SN of *Gtf2i*-KO mice raises the possibility that PNS myelination changes contribute to motor coordination improvements, while CNS myelination abnormalities may play a role in altered anxiety and sociability behaviors. However, future studies are required to dissect the precise contribution of these alterations to behavioral phenotypes’.

Figure 3. Hypermyelination of the SN of *Gtf2i*-KO mice. (A) Representative TEM images of axons from the SN of control and *Gtf2i*-KO mice. (B) Scatter plot of *g*-ratio values and their respective axon diameters. (C) *Gtf2i*-KO mice axons present with significantly lower *g*-ratio values, compared to controls (Kolmogorov-Smirnov test, $P < 0.0001$). (D) Specifically small and medium caliber axons present with lower *g*-ratio values in *Gtf2i*-KO mice, as compared to controls (Student's *t*-test, $P = 0.01$, 0.046 , 0.7 , for axons with diameters of $0\text{-}2\mu\text{m}$, $2\text{-}4\mu\text{m}$, and $4\text{-}6\mu\text{m}$, respectively). (E) *Gtf2i*-KO small caliber axons present increased myelin thickness, representative images. (F) Myelin thickness of *Gtf2i*-KO axons is significantly increased, as compared to controls (Kolmogorov-Smirnov test, $P < 0.0001$). (G) Small and medium caliber specifically present with increased myelin thickness in *Gtf2i*-KO mice, as compared to controls (Student's *t*-test, $P = 0.019$, 0.037 , 0.863 , for axons with diameters of $0\text{-}2\mu\text{m}$, $2\text{-}4\mu\text{m}$, and $4\text{-}6\mu\text{m}$, respectively). (H) Axonal diameter is unchanged in *Gtf2i*-KO mice, compared to controls (Kolmogorov-Smirnov test, $P = 0.153$). (I) Similar number of myelin abnormalities in the SN of control and *Gtf2i*-KO mice (Student's *t*-test, $P = 0.675$). (J) Representative TEM images of Remak's bundles from the SN of control and *Gtf2i*-KO mice. (K) Unchanged ratio of myelinated, unmyelinated, and overall number of axons in the SN of control and *Gtf2i*-KO mice ($n = 3$, Student's *t*-test $P = 0.573$, $P = 0.819$, $P = 0.71$ for myelinated, unmyelinated, and overall number of axons accordingly). (B-D, F-H) Control $n = 3$, 597 axons. *Gtf2i*-KO $n = 3$, 599 axons. ns – non-significant, * $p < 0.05$, **** $p < 0.0001$.

4. To confirm the higher level of anxiety-like behavior, the authors added an elevated zero maze test (EZM) to the open-field exploration test already presented in the original version. However, the results from the EZM only show “a trend towards higher anxiety levels in *Gtf2i*-KO mice”. As confirmation, this is not really optimal.

Response:

We thank the reviewer for their comment. We agree with the reviewer that the anxiety-like behavior phenotype is not robust and have therefore **adopted a more cautious tone**, such as using “moderately increased anxiety” in the subtitle referring to behavioral characterization in the Results section and in other locations in the revised text.

Additionally, to enhance our overall behavioral analysis and expand into additional behavioral domains, **we have incorporated new data in the revised manuscript focusing on another behavioral aspect, social behavior**. This addition was inspired by the reviewer’s insightful and thoughtful comment in the first round of revisions, highlighting that “This, however, may be highly relevant if the idea of the study is to learn about neurodevelopmental disorders such as Williams syndrome (WS)”. We greatly appreciate the reviewer’s keen perspective, which has helped refine and strengthen the focus of our study.

Indeed, *Gtf2i* haploinsufficiency has been directly associated with increased sociability and, in some cases, elevated anxiety-related behaviors in the frame of WS. To further characterize the behavioral phenotype of *Gtf2i*-KO mice, we performed the three-chambers social test to complement the previously reported rotarod, open-field exploration and elevated zero maze (EZM) results. **Our findings show that *Gtf2i*-KO mice exhibit significantly increased sociability compared to controls, as reflected in a higher social index**. In a previous study, we demonstrated that neuronal deletion of *Gtf2i* in the forebrain of mice results in increased sociability and myelin deficits^[10]. The findings in our revised manuscript extend this concept, showing that OL-specific deletion of *Gtf2i* similarly results in heightened sociability alongside distinct myelin alterations, including elevated *Mbp* expression, increased axonal diameter, and thicker myelin sheaths. These results suggest that *Gtf2i* insufficiency throughout brain development, regardless of the affected cell type, and alterations in myelin properties, regardless of their nature, both contribute to increased sociability. **This highlights a broader link between *Gtf2i* function, myelin regulation, and social behavior, making these findings particularly relevant to both *Gtf2i* and WS researchers, as well as the myelin research community**.

These newly obtained data have been incorporated into the revised manuscript within the Results and Discussion sections and added as new panels in the figure summarizing the behavioral tests. For the reviewer’s convenience, the updated text and corresponding figure are provided below (with the new data in the figure labeled within a blue rectangle).

Results section:

‘Among the genes deleted in WS, *Gtf2i* haploinsufficiency has been directly linked to the distinctive hypersocial phenotype^[11-15]. To investigate whether the deletion of *Gtf2i* from myelinating cells affects social behavior, we employed the three-chambers paradigm to evaluate the sociability of *Gtf2i*-KO mice. Both control and *Gtf2i*-KO mice exhibited a preference for interacting with the stranger mouse over the inanimate object (Figure 4D). Although the total interaction time with the stranger mouse and inanimate object did not differ significantly between the two strains (Figure 4D, mean control 302.428±18.798s, mean *Gtf2i*-KO 337.282±14.99s), the social index, calculated as the ratio of time spent interacting with the stranger mouse to the time spent interacting with the inanimate object, was significantly higher in *Gtf2i*-KO mice compared to control mice (Figure 4E, mean control 1.49±0.103, mean *Gtf2i*-KO 1.817±0.1). This increase in the social index suggests that the deletion of *Gtf2i* from myelinating cells enhances sociability in mice’.

Discussion section:

‘While *Gtf2i* deletion from myelinating cells results in opposing effects on myelination compared to neuronal *Gtf2i* deletion^[10], mice lacking *Gtf2i* expression in myelinating cells demonstrate increased sociability, similar to mice with neuronal deletion of *Gtf2i*^[10]. Increased sociability is one of the most prominent behavioral phenotypes of individuals with WS^[16, 17], and *Gtf2i* haploinsufficiency was previously directly associated with said phenotype^[11-13, 18]. As such, our results suggest that reduction of *Gtf2i* levels, regardless of cell-type and myelin properties, results in increased sociability’.

Figure 4. The behavioral phenotype of *Gtf2i*-KO mice reveals improved motor coordination, increased anxiety-like behavior, and enhanced sociability. (A) Rotarod test. *Gtf2i*-KO mice show improved motor coordination as their latency to the first fall is significantly higher compared to controls along three trials of the rotarod test (control $n=19$, *Gtf2i*-KO $n=15$, Two-way ANOVA, $P=0.047$). (B) Open field test. *Gtf2i*-KO mice show increased anxiety-like behavior as they spend significantly more time in the margins of the arena compared to controls (controls $n=15$, *Gtf2i*-KO $n=14$, Two-way ANOVA, $P=0.0325$). (C) Elevated zero maze test (control $n=13$, *Gtf2i*-KO $n=17$). *Gtf2i*-KO mice show a trend towards increased anxiety as they spend less time in the open arms of the EZM compared to controls (Student's t-test, $P=0.066$). (D-E) Three-chambers sociability test (control $n=14$, *Gtf2i*-KO $n=21$). While both control and *Gtf2i*-KO mice show preference towards interacting with the social stimulus rather than with the inanimate object (control - Student's t-test $P=0.0001$, *Gtf2i*-KO - Student's t-test $P<0.0001$), *Gtf2i*-KO mice demonstrate increased sociability as their social index is higher, as compared to controls (Student's t-test, $P=0.035$). * $p < 0.05$, *** $p < 0.001$, **** $p < 0.0001$.

5. Figure 5 shows increased expression levels of nuclear Sox10 and Mbp in cultured oligodendrocytes with Gtf2i deficiency as well as an increased cell surface. Only one single cell is shown in the representative images and the information provided in the Materials and Methods section and figure legend does not allow to judge the robustness of the quantifications. How did the cultures look in general? Were transduced cells healthy? How many cells per “n” underwent quantification? How exactly were intensity measurements performed?

Response:

We thank the reviewer for the insightful comment. We agree that the information provided regarding the findings presented in the original Figure 5 (now Figure 6 in the revised manuscript) in the Methods section should be more detailed, and we have revised it accordingly. These changes have greatly enhanced the overall clarity and validity of our technical approach and findings, and we sincerely thank the reviewer for their valuable input.

Below, we outline the technical aspect of the analysis and provide additional details to address the reviewer’s concerns:

- I. n = 4 mice per group.
- II. Cells from each mouse were seeded on 3-4 coverslips at a density of 60-80K cells per well (based on the number of OPCs isolated from each cortex, plated in 24 well-plates).
- III. Each slide contained coverslips from both conditions (Control and iCre) to ensure consistency in staining and imaging.
- IV. For each marker analyzed (Sox10, Mbp, Tfii-i), each sample was represented by 4-10 cells (median = 6, average = 6.13).
- V. To quantify corrected total cell fluorescence (CTCF) using ImageJ, an experimenter blinded to conditions manually defined the region of interest (ROI) for analysis: Sox10 and Tfii-i: ROI=nucleus. Mbp: ROI=entire cell. For background correction, three randomly selected areas of identical dimensions (outside the ROI) were measured and averaged. The following formula was applied:

$$CTCF = Total\ intensity\ of\ the\ ROI - (Mean\ intensity\ of\ the\ background * ROI\ area).$$

For each slide, the average CTCF of control cells was calculated and used to normalize all other values (similar to normalization in Western blot analysis). As a result, control values were set to 1, and iCre values represent the fold change.

The expanded technical information has been incorporated into the revised manuscript within the Methods section. For the reviewer’s convenience, the updated text is provided below:

Methods section:

‘CTCF analysis in OL-enriched primary cell culture was performed by an experimenter blinded to the genotypes. The experimenter manually marked the region of interest (ROI) for analysis. For Sox10 and Tfii-i that is the nucleus, for Mbp that is either the soma or the whole cell. Following this selection of ROI, intensity of the ROI was measured using ImageJ. To calculate

specific background intensity of each image, 3 random areas with identical dimensions (outside of the ROI) were selected and their intensity measured and averaged. Then, this formula was used:

$$CTCF = Total\ intensity\ of\ the\ ROI - (Mean\ intensity\ of\ the\ background * ROI\ area)$$

For each slide, the average CTCF of control cells was calculated and used to normalize the calculated CTCF of all cells from this slide. Hence, control values are all 1 and iCre values represent the fold change’.

In regards to the reviewer’s concerns regarding the health and overall properties of the culture; we acknowledge that infection or transfection of primary OPC cultures differentiated into OLs is inherently challenging. **To address the reviewer’s concerns regarding culture health and robustness, we have included additional images in the revised manuscript that present representative frames containing multiple cells, providing a clearer reflection of the culture’s overall health.** Additionally, we validated our original results and incorporated new data sets.

Our results were reproducible in the additional experiment, showing increased Mbp intensity levels following exogenous deletion of *Gtf2i*. **Notably, reproducing this experiment enabled us to measure the reduction in Tfii-i intensity levels in the nucleus and the increase of Mbp intensity levels in the soma within the same cell, further strengthening our findings.** These results, along with additional representative images following exogenous deletion of *Gtf2i* in an OL-enriched cell culture, are presented in Supplementary figure 8 (consists entirely of new data), also presented below for convenience:

Supplementary figure 8. Exogenous deletion of *Gtf2i* *in-vitro* in an OL-enriched primary cell culture. (A) Representative images of an AAV-infected OL-enriched primary cell cultures. CBAP-mCherry or CBAP-iCre-mCherry AAVs were added to differentiated OL-enriched cell culture. While not all cells were transduced, the integrity of the culture and cells were unchanged following infection with either AAV or in the absence of transduction. **(B-D)** Intensity measurements from an OL-enriched cell culture following the introduction of control or iCre-expressing AAV. Control $n=3$ (11 cells overall), iCre $n=3$ (19 cells overall). Measurements of Tfii-i and Mbp intensity were taken from the same cell. **(B)** Tfii-i intensity levels are significantly reduced following the introduction of an iCre-expressing AAV (Student's t-test, $P<0.0001$) while **(C)** Mbp intensity levels are significantly increased (Student's t-test, $P=0.0156$). **(D)** The ratio between Mbp intensity levels (soma) and Tfii-i intensity levels (nuclear) is significantly higher following the introduction of an iCre-expressing AAV (Student's t-test, $P<0.0001$). Control levels normalized to 1. * $p < 0.05$, **** $p < 0.0001$.

6. Although I appreciate that the authors have undergone the trouble of performing new ChIP-seq experiments for the revised version, it is very difficult to judge the quality of the provided data in the absence of a GEO submission and an accessible GSE entry. With 8500 consensus peaks, the number is substantially higher than the one in a previous study from E13.5 brain (ref. 111 in the paper). How high is the overlap between peaks in both studies?

Response:

We thank the reviewer for their keen comment. To address this concern, **we have now deposited our ChIP-seq data in the GEO database and provided a reviewer link for access.** This is included in the revised manuscript Methods section, under the Data availability subtitle, where we added the following text: ‘The ChIP-seq data have been deposited in the GEO accession database, under the accession identifier GSE285541’.

As for the reviewer’s comment on the number of consensus peaks: While Kopp et. al^[19] performed ChIP-seq targeting Tfii-i, **a direct comparison between their data and ours is challenging due to several key differences:**

1. **Differences in biological context:** The Kopp et al. dataset originates from whole-brain tissue of embryonic day 13.5 mice, whereas our dataset is derived specifically from cultured OLs, differentiated from cortical OPCs on postnatal day 1. These substantial differences in cellular composition, tissue type, and developmental stage limit the feasibility of a direct comparison and the ability to draw meaningful conclusions regarding peak overlap. Because no previous studies have reported Tfii-i consensus peaks in OLs, we had no relevant existing data source to assess peak overlap, highlighting the novelty of our study on Tfii-i regulation in OLs. Nevertheless, as described in our response to the reviewer’s next comment, we extensively investigated Tfii-i regulation by analyzing the overlap of consensus peaks with data from Gonçalo Castelo-Branco’s group, which describe histone modifications specifically in OLs.

2. **Technical comparison of peak numbers:** While Kopp et. al^[19] reported 1,755 differential binding sites between WT and KO samples, their supplementary data indicate that the number of consensus peaks unique to WT samples averages approximately 30,000 per sample – a figure that is comparable to our dataset describing the consensus peaks number in controls only. Therefore, while the number of consensus peaks in our dataset may initially appear higher than their reported differential sites, it aligns with the total peaks observed in their WT samples, further supporting the robustness of our findings.

3. **Consensus peak selection approach:** In our analysis, we define Tfii-i consensus peaks in OLs by selecting peaks that are present in at least three out of five samples within the control group, ensuring that the consensus peaks are based solely on control samples. This methodology is consistent with established approaches used in other ChIP-seq studies^[20].

7. One of the highlighted outcome from the ChIP-seq studies is the identification of Tfii-i peaks in the vicinity of the *Mbp* and *Sox10* genes (Fig 6E,F). Do any of the identified peaks overlap with the regulatory elements mapped in previous studies for the *Mbp* and *Sox10* genes? There is plenty of literature around, for instance from the labs of Alan Peterson, Tony Antonellis, Andy McCallion and Michael Wegner. If not, it seems far from conclusive that the peaks are bona fide regulatory elements of the *Mbp* and *Sox10* genes. The folding of the distant cRE with the identified Tfii-I peak onto the *Sox10* promoter is, for instance, merely an *in silico* prediction and there are several genes between the cRE and the *Sox10* promoter so that assignment of the cRE the *Sox10* gene is not obvious.

Response:

We thank the reviewer for their detailed and insightful comment, which has significantly contributed to improving a major part of the manuscript and enhancing the scientific impact of our study. **To address the reviewer's comments and further substantiate our ChIP-seq findings, we reanalyzed the raw fastq files, applying more refined bioinformatics parameters, as described in the revised Methods section.**

To assess whether the identified Tfii-i peaks align with previously mapped regulatory elements (REs) of the *Mbp* and *Sox10* genes, we integrated our ChIP-seq data with existing datasets. While Tfii-i peaks did not overlap with previously described REs, we found a notable alignment with data from Gonçalo Castelo-Branco's group, which maps histone modifications specific to OLs^[21]. **Notably, we observed that a Tfii-i peak aligns with the H3K4me3 signal, a marker of active promoters, at the *Mbp* locus, suggesting a role in transcriptional regulation.**

Regarding *Sox10* regulation, our findings indicate that Tfii-i does not bind directly to the *Sox10* promoter region in OLs. However, **we identified Tfii-i peaks at two predicted enhancer-promoter elements near the *Sox10* promoter, suggesting a potential regulatory role.** To validate these *in silico* predictions, we performed a chromatin conformation capture (3C-qPCR) assay using OLs derived from primary OL-enriched cultures. **Our experimental results confirm that these cis-regulatory elements (cREs) physically interact with the *Sox10* promoter, providing additional evidence for a potential regulatory role of Tfii-i in *Sox10* expression.**

These newly obtained data have been incorporated into the revised manuscript within the Results and Discussion sections and added as new panels in Figure 7 (previously Figure 6). The figure has now been expanded with more detailed analyses compared to the original version. For the reviewer's convenience, the updated text (in bold font is newly added text) and corresponding figure are provided below (Figure 7 consists entirely of new data).

Results section:

Tfii-i binds *Mbp* and *Sox10* regulatory regions in OLs

‘As such, changes in the number of mOLs cannot account for the observed myelination and axonal alterations in *Gtf2i*-KO mice, suggesting that these alterations are due to a cell-autonomous mechanism triggered by the deletion of *Gtf2i* in mOLs. However, the precise mechanisms by which Tfii-i modulates the expression of myelination-related genes remain inadequately understood. To unravel the nature of its regulation, it is crucial to identify the specific DNA loci to which Tfii-i directly binds in OLs. By pinpointing these binding sites, we can gain valuable insights into Tfii-i’s mode of action in OLs. Consequently, in the next phase of our research, we aimed to comprehensively map the genome-wide Tfii-i binding sites and enrichment levels, specifically in OLs.

To achieve this, fully differentiated, mature OLs, derived from primary OL-enriched cell cultures, were subjected to chromatin immunoprecipitation followed by sequencing (ChIP-seq), targeting Tfii-i. As expected, Tfii-i peaks were highly abundant in control compared to *Gtf2i*-KO samples (Figure 7A). Overall, 10,467 unique genomic loci were identified as high-confidence Tfii-i binding sites in control samples (i.e., consensus peaks), with minimal overlap in *Gtf2i*-KO samples (Figure 7A and Supplementary Table 2). These findings confirm the robustness of the ChIP-seq protocol and validate the specificity of Tfii-i binding. **Given the near absence of binding in *Gtf2i*-KO samples, subsequent analyses focus exclusively on Tfii-i binding properties in control samples. This approach allows us to comprehensively characterize its genomic distribution and transcriptional regulatory potential without interference from the knockout condition, ensuring a clearer interpretation of its functional role.** Functional annotation analysis revealed that Tfii-i binding sites in control samples were predominantly enriched in intergenic regions and introns; intergenic (6237 peaks), introns (3776 peaks), exons (252 peaks), transcription termination sites (TTS) (114 peaks), and promoters (88 peaks) (Figure 7B). These data correlate well with our previous findings, where we identified cell-type-specific aberrant methylation profiles on enhancers in post-mortem brain tissue of individuals with WS^[22]. **Motif analysis of Tfii-i consensus peaks in OLs within introns, exons, TTS, and promoters (i.e. gene body) identified *Olig2* and *Sox10* enriched motifs, with *Olig2* motif being particularly prominent (Figure 7C). Both *Sox10* and *Olig2* are major TFs in OLs, known to interact with various regulators to drive oligodendroglial cell-fate and myelination^[23-27]. These findings suggest that Tfii-i may function in a synergistic transcriptional role alongside *Olig2* and *Sox10* in OLs.**

To dissect the specific contribution of Tfii-i to the transcriptional program of OLs, we integrated our ChIP-seq data with a cell type-specific database of post-translational modifications of histones (PTMH)^[21]. This integrative approach enabled us to contextualize our findings within the broader epigenetic landscape, providing deeper insights into how Tfii-i regulates gene expression in OLs. Tfii-i peaks in OLs were predominantly located at genomic regions enriched with the post-translational histone modification H3K36me3 (Histone H3 lysine 36 tri-methylations), a marker of actively transcribed euchromatin and transcriptional elongation (Figure 7D). Among the high-confidence Tfii-i peaks, notable binding enrichment was observed at the loci of *Mbp*

(Figure 7E) and *Sox10* (Figure 7F), both of which are critical for OL function. At the *Mbp* locus, the Tfii-i binding site co-localized with the H3K4me3 mark^[21], indicative of active promoters, as well as the H3K36me3 marker (Figure 7E). These findings suggest that in OLs, Tfii-i directly interacts with RE of *Mbp*, potentially contributing to its transcriptional regulation.

Although we did not observe direct binding at the *Sox10* genomic loci, we identified high-confidence Tfii-i peaks at two cis-REs (cREs) predicted to physically interact with the *Sox10* promoter region (Figure 7F, peaks 44 and 69). These predictions were previously assessed *in-silico* by applying the Cicero^[28] tool on OL-derived single-cell CUT&Tag data^[21]. To experimentally assess promoter–cREs interactions in OLs, we performed a chromatin conformation capture assay followed by qPCR (3C-qPCR) on cells derived from an OL-enriched primary culture. The interaction frequency between putative cREs and their target promoters was measured and normalized to untreated genomic DNA (input). First, as a positive control, we assessed interaction frequencies between *Mbp* promoter and known enhancers of *Mbp* (termed M3 and M5), which do not harbor Tfii-i binding sites^[29-32] (Supplementary figure 12). M3 and M5 interaction frequencies with the *Mbp* promoter were significantly higher, as compared to background (Figure 7G, M3 – 4.132 ± 0.509 , M5 – 5.038 ± 0.038 , background - 1 ± 0.192), affirming the technical success of the 3C-qPCR protocol. We then proceeded to assess cREs-promoter interaction frequencies at the *Sox10* locus. As a negative control, we quantified interaction frequencies on the same chromosome as *Sox10*, at an unrelated, distant *Mal2* gene promoter and a random intergenic region located at a similar genomic distance to the putative *Sox10* cRE. Our results revealed a significant increase in interaction frequency between both cREs and the *Sox10* promoter (peak 44 – 24.906 ± 3.564 , peak 69 – 17.283 ± 2.275), compared to baseline interaction levels detected at the *Mal2* locus (background 1 ± 0.453) (Figure 7H, Supplementary figure 12). These findings demonstrate that the *Sox10* promoter interacts with the specific predicted cREs. Enhancer-promoter interactions were previously shown to play a crucial role in OL transcriptional program^[33, 34]. As such, this result supports the notion that Tfii-i contributes to *Sox10* transcriptional regulation.

Collectively, our findings suggest that in OLs, Tfii-i is involved in the regulation of *Sox10* and *Mbp* transcription. This regulatory role may contribute to the observed altered interplay of myelin thickness and axonal diameter observed in the CC of *Gtf2i*-KO mice, further supporting a cell-autonomous effect of Tfii-i in OLs.’

Discussion section:

‘Importantly, we found that Tfii-i peaks in OLs were predominantly localized to genomic regions enriched with the H3K36me3 PTMH modification. Collectively, our findings suggest that in OLs, Tfii-i functions as an auxiliary regulator of *Mbp* and *Sox10* transcription. Our contribution is thus key to understanding the regulatory roles of Tfii-i specifically in OLs, and for describing the nature of this regulation on the key myelin genes *Sox10* and *Mbp*, which we suggest to be repressive’.

Figure 7. In OLS, Tfii-i binds to RE of *Mbp* and *Sox10*. (A) Heatmap showing relative Tfii-i binding in OLS of control and *Gtf2i*-KO mice. (B) Genomic distribution of Tfii-i consensus peaks in control OLS. (C) Motif analysis of Tfii-i peaks in OLS within gene body identified Olig2 and Sox10 binding motifs. (D) Tfii-i peaks overlap with OL-specific PTMH, predominantly in genomic regions enriched with H3K36me3 binding, marking actively transcribed regions. (E) Tfii-i binding at *Mbp* genomic locus. Upper row – Tfii-i binding in OLS with consensus peaks marked in gray (peak numbers – 41, 75, 62). **The track presented represents an aggregate of all Tfii-i peaks from the different OL samples. While some regions showed high enrichment in individual samples, they lacked consistent presence across all samples and therefore did not meet the criteria for consensus peaks.** Second, third, and fourth rows – H3K4me3, H3K36me3, and H3K27ac OL-specific binding data, accordingly. Fifth row – genome track. Lower row – mOL-specific predicted cRE data, M3 and M5 binding to the *Mbp* promoter are highlighted in blue. (F) Tfii-i binding at *Sox10* genomic locus. Upper row – Tfii-i binding in OLS with consensus peaks marked in gray (peaks 44 and 69). Second, third, and fourth rows – H3K4me3, H3K36me3, and H3K27ac OL-specific binding data, accordingly. Fifth row – genome track. Lower row – mOL-specific predicted cRE data, Tfii-i peaks (44 and 69) within the introns of *Micall1* gene align with these cREs that fold into the *Sox10* promoter (highlighted in blue). (G) 3C-qPCR ($n=3/4$, control only). Significantly higher interaction frequencies between *Mbp* promoter region and known *Mbp* enhancers (M3 and M5), compared to background region (Student's t-test, $P=0.0012$ for M3, $P<0.0001$ for M5). Background interaction frequency normalized to 1 (dashed line). (H) 3C-qPCR ($n=4$, control only). Significantly higher interaction frequencies between both cREs (peaks 44 and 69) and *Sox10* promoter as compared to baseline interaction frequency as detected at the background (*Mal2*) locus (Student's t-test, $P=0.0006$ for peak 44, $P=0.0004$ for peak 69). Background interaction frequency normalized to 1 (dashed line). ** $p < 0.01$, *** $p < 0.001$, **** $p < 0.0001$.

8. In the current state, I find the evidence provided for the proposed mechanism very preliminary and not convincing. Much remains on the correlational and suggestive level – both for Gtf2i as a direct repressor of Mbp and Sox10, and for the role of Mbp and Sox10 as effectors of the Gtf2i deletion. To substantiate the authors' hypothesis that increased Sox10 and Mbp expression upon Gtf2i deletion causes the phenotypic alterations, backup from the literature would be helpful. Is it known what experimentally induced overexpression of Mbp or Sox10 (starting in late OPCs at the earliest) does to myelin?

Response:

We hope that with the new data on ChIP-seq, 3C, and PTMH overlap, we have provided further evidence to support our proposed mechanism, addressing the reviewer's concerns and strengthening the overall validity of our findings.

We thank the reviewer for highlighting this important gap in our discussion. To better contextualize the role of *Sox10* and *Mbp* over- and re-expression in myelination, we have expanded the Discussion section to incorporate relevant findings from the literature. Below is the revised text, for convenience:

Discussion section:

‘Myelin, OLs, and axon health are tightly regulated by one another^[35-40]. *Sox10* expression levels were shown to decrease following exposure to cuprizone^[41], a copper chelator known to induce demyelination^[42, 43]. Overexpression of *Sox10* in the hippocampus following demyelination by cuprizone, improved behavioral deficits, myelin ultrastructural properties, and normalized myelin sheath-related protein expression levels^[44]. In another study of a cuprizone demyelination model, upon the introduction of *Sox10* through exosomes, OPC differentiation to OLs was promoted and an increase in *Mbp* protein levels was observed^[41]. As such, these data suggest that increased *Sox10* expression levels may contribute to remyelination and maintenance of the myelin sheath.

In accordance with the increased axonal diameter^[35, 45-47], we showed that *Gtf2i* deletion from mOLs resulted in increased myelin thickness wrapping these axons in the CC. Notably, the *Mbp* level is a rate-limiting consideration in CNS myelination^[48]. Furthermore, *Mbp* mRNA levels were shown to affect myelin sheath properties^[31]. Recently, *Mbp* expression was shown to support axonal regeneration in neural progenitor cells^[49], and to improve neurites properties^[50]. As such, excess *Mbp* transcripts and protein in *Gtf2i*-KO mice may underline the mechanism by which the myelin thickness of CC axons is increased in these mutant mice. Furthermore, in accordance with the increased axonal diameter and thicker myelin, we show enhanced signal conduction in *Gtf2i*-KO mice, as reflected in electrophysiological recordings of the CC’.

We sincerely appreciate the reviewer's time and insightful comments throughout the revision process and are truly grateful.

Best regards,

Prof. Boaz Barak, Ph. D., MBA

References relevant for the rebuttal letter

1. Yu, M., et al., *Visual abnormalities associated with enhanced optic nerve myelination*. Brain Research, 2011. **1374**: p. 36-42.
2. Etxeberria, A., et al., *Dynamic Modulation of Myelination in Response to Visual Stimuli Alters Optic Nerve Conduction Velocity*. The Journal of Neuroscience, 2016. **36**(26): p. 6937-6948.
3. Arancibia-Cárcamo, I.L., et al., *Node of Ranvier length as a potential regulator of myelinated axon conduction speed*. eLife, 2017. **6**: p. e23329.
4. Rasband, M.N., et al., *CNP is required for maintenance of axon–glia interactions at nodes of Ranvier in the CNS*. Glia, 2005. **50**(1): p. 86-90.
5. Lappe-Siefke, C., et al., *Disruption of Cnp1 uncouples oligodendroglial functions in axonal support and myelination*. Nat Genet, 2003. **33**(3): p. 366-74.
6. Madsen, P.M., et al., *Oligodendroglial TNFR2 Mediates Membrane TNF-Dependent Repair in Experimental Autoimmune Encephalomyelitis by Promoting Oligodendrocyte Differentiation and Remyelination*. The Journal of Neuroscience, 2016. **36**(18): p. 5128-5143.
7. Millet, V., M. Marder, and L.A. Pasquini, *Adult CNP::EGFP transgenic mouse shows pronounced hypomyelination and an increased vulnerability to cuprizone-induced demyelination*. Experimental Neurology, 2012. **233**(1): p. 490-504.
8. Fischer, I., et al., *Shank3 mutation impairs glutamate signaling and myelination in ASD mouse model and human iPSC-derived OPCs*. Science Advances, 2024. **10**(41): p. ead14573.
9. Harty, B.L. and K.R. Monk, *Unwrapping the unappreciated: recent progress in Remak Schwann cell biology*. Curr Opin Neurobiol, 2017. **47**: p. 131-137.
10. Barak, B., et al., *Neuronal deletion of Gtf2i, associated with Williams syndrome, causes behavioral and myelin alterations rescuable by a remyelinating drug*. Nature Neuroscience, 2019. **22**(5): p. 700-708.
11. Crespi, B.J. and P.L. Hurd, *Cognitive-behavioral phenotypes of Williams syndrome are associated with genetic variation in the GTF2I gene, in a healthy population*. BMC Neuroscience, 2014. **15**(1): p. 127.
12. Sakurai, T., et al., *Haploinsufficiency of Gtf2i, a gene deleted in Williams Syndrome, leads to increases in social interactions*. Autism Research, 2011. **4**(1): p. 28-39.
13. Dai, L., et al., *Is it Williams syndrome? GTF2IRD1 implicated in visual-spatial construction and GTF2I in sociability revealed by high resolution arrays*. Am J Med Genet A, 2009. **149a**(3): p. 302-14.
14. Gagliardi, C., et al., *Unusual cognitive and behavioural profile in a Williams syndrome patient with atypical 7q11.23 deletion*. Journal of Medical Genetics, 2003. **40**(7): p. 526-530.
15. Karmiloff-Smith, A., et al., *Using case study comparisons to explore genotype-phenotype correlations in Williams-Beuren syndrome*. Journal of Medical Genetics, 2003. **40**(2): p. 136-140.
16. Kozel, B.A., et al., *Williams syndrome*. Nature Reviews Disease Primers, 2021. **7**(1): p. 42.
17. Pober, B.R., *Williams–Beuren Syndrome*. New England Journal of Medicine, 2010. **362**(3): p. 239-252.
18. Tassabehji, M., et al., *GTF2IRD1 in Craniofacial Development of Humans and Mice*. Science, 2005. **310**(5751): p. 1184-1187.

19. Kopp, N.D., et al., *Functions of Gtf2i and Gtf2ird1 in the developing brain: transcription, DNA binding and long-term behavioral consequences*. Human Molecular Genetics, 2020. **29**(9): p. 1498-1519.
20. Elbaz, B., et al., *The bone transcription factor Osterix controls extracellular matrix- and node of Ranvier-related gene expression in oligodendrocytes*. Neuron, 2024. **112**(2): p. 247-263.e6.
21. Bartosovic, M., M. Kabbe, and G. Castelo-Branco, *Single-cell CUT&Tag profiles histone modifications and transcription factors in complex tissues*. Nature Biotechnology, 2021. **39**(7): p. 825-835.
22. Trangle, S.S., et al., *In individuals with Williams syndrome, dysregulation of methylation in non-coding regions of neuronal and oligodendrocyte DNA is associated with pathology and cortical development*. Molecular Psychiatry, 2023. **28**(3): p. 1112-1127.
23. Sock, E. and M. Wegner, *Using the lineage determinants Olig2 and Sox10 to explore transcriptional regulation of oligodendrocyte development*. Developmental Neurobiology, 2021. **81**(7): p. 892-901.
24. Hornig, J., et al., *The Transcription Factors Sox10 and Myrf Define an Essential Regulatory Network Module in Differentiating Oligodendrocytes*. PLOS Genetics, 2013. **9**(10): p. e1003907.
25. Li, H., et al., *Olig1 and Sox10 Interact Synergistically to Drive Myelin Basic Protein Transcription in Oligodendrocytes*. The Journal of Neuroscience, 2007. **27**(52): p. 14375-14382.
26. Lopez-Anido, C., et al., *Differential Sox10 genomic occupancy in myelinating glia*. Glia, 2015. **63**(11): p. 1897-1914.
27. Liu, Z., et al., *Induction of oligodendrocyte differentiation by Olig2 and Sox10: Evidence for reciprocal interactions and dosage-dependent mechanisms*. Developmental Biology, 2007. **302**(2): p. 683-693.
28. Pliner, H.A., et al., *Cicero Predicts cis-Regulatory DNA Interactions from Single-Cell Chromatin Accessibility Data*. Mol Cell, 2018. **71**(5): p. 858-871.e8.
29. Dionne, N., et al., *Functional organization of an Mbp enhancer exposes striking transcriptional regulatory diversity within myelinating glia*. Glia, 2016. **64**(1): p. 175-194.
30. Dib, S., et al., *Regulatory modules function in a non-autonomous manner to control transcription of the mbp gene*. Nucleic Acids Research, 2010. **39**(7): p. 2548-2558.
31. Bagheri, H., et al., *Myelin basic protein mRNA levels affect myelin sheath dimensions, architecture, plasticity, and density of resident glial cells*. Glia, 2024. **72**(10): p. 1893-1914.
32. Bagheri, H., et al., *Transcriptional regulators of the Golli/myelin basic protein locus integrate additive and stealth activities*. PLOS Genetics, 2020. **16**(8): p. e1008752.
33. Kim, D., et al., *A principled strategy for mapping enhancers to genes*. Scientific Reports, 2019. **9**(1): p. 11043.
34. Cheng, N., et al., *STAG2 promotes the myelination transcriptional program in oligodendrocytes*. eLife, 2022. **11**: p. e77848.
35. Nave, K.-A. and J.L. Salzer, *Axonal regulation of myelination by neuregulin 1*. Current Opinion in Neurobiology, 2006. **16**(5): p. 492-500.
36. Fruttiger, M., et al., *Crucial Role for the Myelin-associated Glycoprotein in the Maintenance of Axon-Myelin Integrity*. European Journal of Neuroscience, 1995. **7**(3): p. 511-515.
37. Griffiths, I., et al., *Axonal Swellings and Degeneration in Mice Lacking the Major Proteolipid of Myelin*. Science, 1998. **280**(5369): p. 1610-1613.
38. Nave, K.A., *Myelination and support of axonal integrity by glia*. Nature, 2010. **468**(7321): p. 244-52.

39. Nave, K.A., *Myelination and the trophic support of long axons*. Nat Rev Neurosci, 2010. **11**(4): p. 275-83.
40. Traka, M., et al., *Oligodendrocyte death results in immune-mediated CNS demyelination*. Nat Neurosci, 2016. **19**(1): p. 65-74.
41. He, J., et al., *Exosome-specific loading Sox10 for the treatment of Cuprizone-induced demyelinating model*. Biomedicine & Pharmacotherapy, 2024. **171**: p. 116128.
42. Franco-Pons, N., et al., *Behavioral deficits in the cuprizone-induced murine model of demyelination/remyelination*. Toxicology Letters, 2007. **169**(3): p. 205-213.
43. Matsushima, G.K. and P. Morell, *The Neurotoxicant, Cuprizone, as a Model to Study Demyelination and Remyelination in the Central Nervous System*. Brain Pathology, 2001. **11**(1): p. 107-116.
44. Shao, Y., et al., *Effect of Sox10 on remyelination of the hippocampus in cuprizone-induced demyelinated mice*. Brain and Behavior, 2020. **10**(6): p. e01623.
45. Friede, R.L., *Control of myelin formation by axon caliber. (With a model of the control mechanism)*. Journal of Comparative Neurology, 1972. **144**(2): p. 233-252.
46. Yin, X., et al., *Myelin-Associated Glycoprotein Is a Myelin Signal that Modulates the Caliber of Myelinated Axons*. The Journal of Neuroscience, 1998. **18**(6): p. 1953-1962.
47. Mayoral, S.R., et al., *Initiation of CNS Myelination in the Optic Nerve Is Dependent on Axon Caliber*. Cell Reports, 2018. **25**(3): p. 544-550.e3.
48. Readhead, C., et al., *Expression of a myelin basic protein gene in transgenic shiverer mice: Correction of the dysmyelinating phenotype*. Cell, 1987. **48**(4): p. 703-712.
49. Yan, Z., et al., *Myelin basic protein enhances axonal regeneration from neural progenitor cells*. Cell & Bioscience, 2021. **11**(1): p. 80.
50. Smith, G.S.T., et al., *Nucleus-localized 21.5-kDa myelin basic protein promotes oligodendrocyte proliferation and enhances neurite outgrowth in coculture, unlike the plasma membrane-associated 18.5-kDa isoform*. Journal of Neuroscience Research, 2013. **91**(3): p. 349-362.

POINT-BY-POINT RESPONSE

The revised manuscript describes an impressive phenotype obtained after *Gtf2i* deletion, including increased myelination and increased expression of myelin genes. While most of the figures are relatively convincing in the revised version, there are significant deficiencies with the requested response to the genomic analysis that was performed in Figure 7.

We thank the reviewer for their thoughtful and constructive feedback, and for the time and effort they invested in evaluating our revised manuscript. We are encouraged that the reviewer found the phenotype resulting from *Gtf2i* deletion in myelinating glial cells to be impressive, and that they were generally convinced by the data presented in the revised figures.

Regarding the genomic analysis, we appreciate the reviewer's comments on this important aspect of our work. We fully acknowledge that ChIP-seq from low-input material, such as the mOL population studied here, presents substantial technical challenges and may yield more limited information compared to high-input protocols. While we understand their concerns, we believe that the revisions incorporated into the updated manuscript have addressed the primary technical issues raised in the previous round of review. Specifically, **we applied a stringent and conservative bioinformatics pipeline, using a FDR threshold of <0.01 and requiring consensus peaks across biological replicates. To further strengthen our analysis, we cross-referenced our results with previously published mOL datasets and, critically, performed independent ChIP-qPCR validation using a separate cohort of samples.** These multiple layers of technical and biological validation enhance our confidence in the robustness of the dataset and support the biological conclusions presented in the revised manuscript. **Importantly, we emphasize that the conclusions drawn from this analysis significantly strengthen the overall narrative of the manuscript and enhance our understanding of *Gtf2i*'s role in regulating myelination specifically in mOLs.**

Below, we provide **a point-by-point response detailing the new datasets we generated, the improvements made to Figure 7, and our responses to any remaining concerns.**

The issues in this figure are first apparent in the panel 7A, which shows radically different peak patterns in the biological replicates of the control samples.

We thank the reviewer for highlighting this important issue. We acknowledge that the original presentation of the heatmap in Figure 7A may have been confusing. As noted in the Methods section, the heatmap was generated in R using default settings, which apply hierarchical clustering to the rows (peaks). **When clustering is enabled, peaks are ordered within each sample based on similarity in signal intensity, resulting in the most prominent peaks appearing at the top of each replicate.** While this approach is useful for visualizing internal structure within individual samples, it can obscure cross-sample comparisons.

To address the reviewer's concern and enhance visual comparability between biological replicates, we have re-generated the heatmap with clustering disabled. In this updated version, peak order is fixed based on chromosomal location, allowing for clearer visualization of signal consistency across samples. This adjustment enhances interpretability without altering the underlying data or analytical approach.

Importantly, as noted both above and in the revised manuscript, **we acknowledge the presence of some variability between replicates and therefore applied a stringent and conservative bioinformatics pipeline in our downstream analysis. Specifically, peaks were retained only if they were present in at least 3 out of 5 control replicates, ensuring robustness and reproducibility (i.e. consensus peaks).** We hope that this revised figure effectively addresses the reviewer's concern and more accurately conveys the reliability of our results.

For the reviewer's convenience, the revised version of Figure 7A is presented below:

Figure 7. In mOLs, Tfii-i binds RE of *Mbp* and *Sox10*. (A) Heatmap showing genome-wide Tfii-i binding intensity from ChIP-seq data in mOLs derived from control and *Gtf2i*-KO mice.

The results of panels E and F show mostly background binding and a lack of convincing peaks.

We thank the reviewer for this insightful comment. The original panels E and F were generated from post-peak calling data using MACS2 (i.e. narrowPeak format), which highlights enriched regions while filtering out background signal. While this approach is suitable for identifying peak locations, we acknowledge that it may have limited the viewer ability to assess the broader ChIP-seq signal landscape, especially in terms of signal-to-noise ratio and background distribution.

To address this concern and provide a more comprehensive representation of the ChIP-seq data, we have re-generated the genome browser tracks using the original BAM files and converted them into BigWig format. These revised tracks include both the specific signal and background, allowing for a more faithful and context-rich visualization of the data. This change significantly enhances the interpretability of peak regions and allows the reviewer and readers to more fully appreciate the signal distribution across the genome. We believe this presentation better reflects the underlying data and improves transparency and reproducibility, as well as the clarity and quality of the visualization.

For the reviewer's convenience, the revised version of the previously labeled Figure 7E-F, now updated as Figure 7F-G, is presented below:

F

G

Figure 7. In mOLs, Tfii-i binds to RE of *Mbp* and *Sox10*. (F) Tfii-i binding at the *Mbp* genomic locus. The upper track displays Tfii-i binding in mOLs, with consensus peaks highlighted in gray (peaks 75 and 62), TSS is marked in green. The second, third, and fourth tracks show mOL-specific H3K4me3, H3K27ac, and H3K36me3 binding data, respectively. The genome annotation track indicates known *Mbp* enhancers M3 and M5. The bottom track presents mOL-specific predicted cREs and their putative interactions with target genes, visualized as arcs. (G) Tfii-i binding at *Sox10* genomic locus. The upper track displays Tfii-i binding in mOLs, with consensus peaks highlighted in gray (peaks 44 and 69), the TSS is marked in green. The second, third, and fourth tracks show mOL-specific H3K4me3, H3K27ac, and H3K36me3 binding data, respectively. The genome track indicates known *Sox10* enhancers (U1-U5, D6-D7). The bottom track presents mOL-specific predicted cREs. Tfii-i peaks 44 and 69, located within intronic regions of the *Micall1* gene, align with these cREs and are predicted to physically interact with the *Sox10* promoter (highlighted in blue).

To further validate the specificity and reproducibility of the identified ChIP-seq peaks, we have performed ChIP-qPCR experiments, targeting three representative genomic loci with evident Tfii-i binding in the ChIP-seq data: two putative enhancers near *Sox10* (peaks 69 and 44) and the *Mbp* promoter (peak 75). As a negative control (labeled as “Blank” in the graph below), we included a genomic region with no evidence of Tfii-i binding in our ChIP-seq data. The ChIP-qPCR results (presented in new Supplementary Figure 12C) confirmed significantly reduced Tfii-i occupancy at the target loci in *Gtf2i*-KO samples, while no difference was observed at the negative control locus across genotypes. These findings independently confirm our ChIP-seq results and provide robust support for the presence and specificity of the reported Tfii-i binding events.

For the reviewer’s convenience, the new Supplementary Figure 12C is presented below:

Supplementary figure 12. Supplementary ChIP-seq results and validation. (C) ChIP-qPCR validation of ChIP-seq results (control $n=5$, *Gtf2i*-KO $n=4$). Primers targeting consensus peaks 75 (*Mbp* locus), 44, and 69 (*Sox10* locus) confirmed reduced Tfii-i binding in *Gtf2i*-KO group compared to control group at peak 75 (Student’s t-test, $P = 0.035$), peak 44 (Student’s t-test, $P = 0.013$), and peak 69 (Student’s t-test, $P = 0.023$). No significant change was observed at a negative control region (Student’s t-test, $P = 0.511$).

Lastly, we wish to correct an error in the original manuscript: the q-value threshold used for MACS2 peak calling was 0.01, not 0.05 as previously indicated. This correction has been made in the revised Methods section. We hope that the improved visualizations and independent validation experiments satisfactorily address the reviewer’s concerns and substantiate the robustness and biological relevance of the ChIP-seq data presented in original panels E and F (now panels F and G).

An enrichment of SOX10 and OLIG2 motifs is interesting, but the examples for *Mbp* and *Sox10* genes do not show colocalization with any of the known SOX10 and OLIG2 binding sites nor with any of the known enhancers of these genes (see H3K27ac track). An enrichment of these binding sites would normally predict a more substantial association of GTF2I with H3K27ac. Overall, as detailed below, there is little evidence to conclude that (as stated): "These findings suggest that Tfii-i may function in a synergistic transcriptional role alongside Olig2 and Sox10 in OLs."

We thank the reviewer for this thoughtful and important comment. We acknowledge that the Tfii-i binding peaks shown in the original panels E and F exhibit limited colocalization with known *Sox10* or *Olig2* binding sites at the *Mbp* and *Sox10* loci, and with previously characterized enhancers in these regions. We also acknowledge that at these specific loci, Tfii-i shows minimal overlap with H3K27ac-marked regions, which are indicative of active enhancers.

Nevertheless, **we believe this localized observation does not negate the broader biological relevance of the genomic data. To further explore potential functional interactions, we performed a genome-wide intersection of our Tfii-i ChIP-seq dataset with previously published ChIP-seq datasets for Olig2 and Srf (Figure 7D). This analysis revealed a substantial degree of colocalization between Tfii-i and Olig2 binding sites genome-wide, which is consistent with the motif enrichment results shown in Figure 7C.** Notably, this global pattern suggests that Tfii-i may contribute to transcriptional regulation in mOLs through co-occupancy at a subset of Olig2-bound regulatory regions, even if it does not directly bind canonical enhancers at *Mbp* and *Sox10*.

In support of this, **we also identified a specific Tfii-i peak (peak 44) located near a putative *Sox10* enhancer, which overlaps with H3K27ac signal derived specifically from purified mOLs.** While this does not constitute conclusive evidence of enhancer activity, it provides a candidate locus and suggests that Tfii-i binding may, in some cases, align with active regulatory elements.

Taken together, while we acknowledge that the data do not demonstrate co-occupancy of Tfii-i binding at known *Sox10* or *Olig2* binding sites within the *Mbp* and *Sox10* genomic loci, or at canonical *Mbp* and *Sox10* enhancers, we note several supporting findings for other genomic loci. **The observed genome-wide colocalization with Olig2, along with motif enrichment and selected overlaps with H3K27ac signals, supports a model in which Tfii-i may function with a broader Olig2-associated transcriptional network in mOLs.** In response to the reviewer's comment, we have revised the manuscript text to more cautiously interpret these findings and avoid overstating the evidence for direct synergistic activity at specific enhancer loci.

For the reviewer’s convenience, the revised text and panels C-D in revised Figure 7 are presented below:

‘Motif analysis of Tfii-i consensus peaks in mOLs within introns, exons, TTS, and promoters (i.e. across the gene body) revealed enrichment of Olig2 and Sox10 motifs, with the Olig2 motif being particularly prominent, alongside known Tfii-i binding motifs^[1, 2] (Figure 7C, Supplementary Figure 12A). To further investigate this potential interaction, we performed colocalization analysis of Tfii-i peaks in mOLs using publicly available ChIP-seq datasets. This analysis demonstrated substantial overlap between Tfii-i and Olig2 binding sites, as well as with Srf, another TF expressed in OLs^[3], whereas overlap with the neuronal TF Neurod1 was minimal (Figure 7D). Olig2 is a core regulator of OL differentiation and myelination, functioning through a variety of cofactors and binding partners^[4-11]. Collectively, these findings suggest that Tfii-i may play a broad role in a genome-wide transcriptional network coordinated by Olig2 in mOLs.’

Figure 7. In mOLs, Tfii-i binds to RE of *Mbp* and *Sox10*. (C) Motif analysis within a 100 bp window centered on Tfii-i peaks in mOLs reveals enrichment of Olig2 and Sox10 binding motifs. (D) Colocalization analysis of Tfii-i consensus peaks in mOLs shows high overlap with Srf and Olig2 binding.

The genome browser sessions are not optimal since the principal transcription start site of *Mbp* (downstream of M3 and M5) is not clearly designated in panel E. The farthest upstream exon is for the golli isoform of *Mbp*, which is a relatively minor transcript.

We agree with the reviewer that the original genome browser panels were suboptimal in clearly indicating the location of the principal TSS of *Mbp*. **In the revised version, we have refined the browser tracks to focus specifically on the *Mbp* gene and some of its well-characterized enhancers (e.g., M3 and M5). To further improve clarity, we have added graphical annotations to explicitly highlight the principal TSS associated with the classic *Mbp* isoforms relevant to myelination.** We believe these modifications improve the figure's interpretability and directly address the reviewer's concern.

F

Figure 7. In mOLs, Tfi-i binds to RE of *Mbp* and *Sox10*. (F) Tfi-i binding at the *Mbp* genomic locus. The upper track displays Tfi-i binding in mOLs, with consensus peaks highlighted in gray (peaks 75 and 62), TSS is marked in green. The second, third, and fourth tracks show mOL-specific H3K4me3, H3K27ac, and H3K36me3 binding data, respectively. The genome annotation track indicates known *Mbp* enhancers M3 and M5. The bottom track presents mOL-specific predicted cREs and their putative interactions with target genes, visualized as arcs.

In addition, the Sox10 locus is showing putative regulatory elements downstream of SOX10, although a number of studies have principally localized enhancers upstream of the SOX10 gene, which is omitted in this diagram.

We thank the reviewer for this comment and agree that the upstream regulatory landscape of Sox10, including well-characterized enhancers, was underrepresented in the previous version of the figure. **In the revised genome browser panel, we have now included these upstream enhancer regions (U1-U5), and the known downstream enhancers (D6-D7) to provide a more complete view.**

Additionally, we have updated the corresponding text in the manuscript to acknowledge both the presence and the functional relevance of these upstream enhancers, as established in prior studies. We believe these revisions offer a more accurate and comprehensive depiction of the Sox10 regulatory architecture and better contextualize our findings within the framework of existing literature.

The revised text and panel G from Figure 7 are provided below for the reviewer's convenience:

'Although we did not observe direct Tfii-i binding at the Sox10 genomic loci, at known Sox10 enhancers^[12], or at established Olig2 binding sites within the Sox10 locus (Supplementary Figure 12E), we did identify high-confidence Tfii-i peaks at two cis-REs (cREs) predicted to physically interact with the Sox10 promoter region (Figure 7F, peaks 44 and 69).'

G

Figure 7. In mOLs, Tfii-i binds to RE of *Mbp* and *Sox10*. (G) Tfii-i binding at *Sox10* genomic locus. The upper track displays Tfii-i binding in mOLs, with consensus peaks highlighted in gray (peaks 44 and 69), the TSS is marked in green. The second, third, and fourth tracks show mOL-specific H3K4me3, H3K27ac, and H3K36me3 binding data, respectively. The genome track indicates known *Sox10* enhancers (U1-U5, D6-D7). The bottom track presents mOL-specific predicted cREs. Tfii-i peaks 44 and 69, located within intronic regions of the *Micall1* gene, align with these cREs and are predicted to physically interact with the *Sox10* promoter (highlighted in blue).

The suggestion that the TFII-I peaks coincide with predicted enhancer/promoter elements is not supported by reference to oligodendrocyte data.

We appreciate the reviewer's comment and the opportunity to clarify this point. **With the exception of the embryonic Tfii-i panel, included in the revised manuscript (Supplementary Figure 12D-E), all genome browser tracks presented are derived from mOL-specific datasets.**

Specifically, **the PTMH histone modification data (H3K4me3, H3K27ac, and H3K36me3) were obtained from the mature mOLs cluster characterized in the study by the Gonçalo-Branco group^[13].** The candidate cis-regulatory elements (cREs) displayed in the same browser panels were also derived from this study and were computationally predicted based on the mOL-specific chromatin profiles.

In addition, **the Olig2 and Rad21 ChIP-seq datasets shown in Supplementary Figures 12F-G were originated from the same study and are also specific to the mOL lineage.** Therefore, our interpretation that Tfii-i peaks localize near predicted enhancer and promoter elements is firmly based on chromatin and transcription factor binding data that are directly relevant to mOLs.

In addition to the IGV tracks, we conducted a quantitative overlap analysis using bedtools intersect, incorporating PTMH histone modification data from the mOL cluster characterized by the Gonçalo-Branco group^[13]. As shown in **Figure 7E**, this analysis revealed that approximately 13% of consensus Tfii-i peaks overlap with H3K27ac-marked regions, indicative of active enhancers, in genomic loci beyond *Mbp* and *Sox10*. Notably, we also observed substantial overlap with H3K36me3-enriched regions, which are associated with transcriptional elongation. These findings support a broader role for Tfii-i in transcriptional regulation within mOLs.

To address the reviewer's concern, we have revised the manuscript text to more explicitly describe the source, specificity, and relevance of these datasets to the mOL context. The revised text and an example for the improved readability taken from Figure 7F (highlighted in a red squares) is attached below, for the reviewer's convenience:

'To dissect the specific contribution of Tfii-i to the transcriptional program of mOLs, we integrated our ChIP-seq data with a cell type-specific database of post-translational histone modifications (PTMH), which includes datasets derived specifically from mOLs^[13].'

'These predictions were previously assessed *in-silico* by applying the Cicero^[14] tool on mOL-derived single-cell CUT&Tag data^[13].'

Figure 7. In mOLs, Tfii-i binds to RE of *Mbp* and *Sox10*. (E) Overlap analysis of Tfii-i peaks with **mOL-specific** PTMH reveals predominant localization within genomic regions enriched for H3K36me3, a mark associated with active transcription.

Figure 7. In mOLs, Tfii-i binds to RE of *Mbp* and *Sox10*. (F) Tfii-i binding at the *Mbp* genomic locus. The upper track displays Tfii-i binding in mOLs, with consensus peaks highlighted in gray (peaks 75 and 62), TSS is marked in green. The second, third, and fourth tracks show **mOL-specific** H3K4me3, H3K27ac, and H3K36me3 binding data, respectively. The genome annotation track indicates known *Mbp* enhancers M3 and M5. The bottom track presents mOL-specific predicted cREs and their putative interactions with target genes, visualized as arcs.

Since there is a previous analysis of OLIG2 binding sites (in reference 160) and GTF2I binding (ref. 113), these should be shown in these browser sessions as comparisons.

We thank the reviewer for this thoughtful suggestion. **While we agree that comparing our Tfii-i ChIP-seq dataset with previously published datasets may provide additional context, we would like to emphasize the significant biological limitations associated with a direct comparison to the Tfii-i data from reference 113.**

Specifically, **the dataset in reference 113 was generated from whole embryonic brain tissue at E13.5**, which differs markedly from the system used in our study, **purified cortical mOLs differentiated *in-vitro* from postnatal day 1-derived progenitors**. These two experimental contexts **differ in developmental stage, cellular heterogeneity, and tissue origin, all of which are known to profoundly influence transcription factor binding profiles**. As such, a direct comparison between these datasets could be highly confounded by these variables and potentially misleading in interpreting the function or specificity of Tfii-i binding in the OL lineage.

Nonetheless, **to address the reviewer's comment, to provide transparency, and to acknowledge the context-dependent nature of transcription factor binding, we have included a genome browser panel (now presented in Supplementary Figure 12D-E), a Venn diagram (now presented in Supplementary Figure 12F), and overlap analysis (now presented in Supplementary Figure 12G), that compare our mOL-specific Tfii-i ChIP-seq data with the embryonic dataset from reference 113.** This allows readers to appreciate the biological differences while still observing potential overlap, with appropriate caution in interpretation. Supplementary Figure 12F-G are provided below for the reviewer's convenience:

D

Tfii-i binding in mOLs at *Mbp* locus
E

Tfii-i binding in mOLs at *Sox10* locus
F

Overlap with embryonic Tfii-i binding

G

Overlap with OL-related TFs binding, embryonic Tfii-i binding, and architectural proteins binding

Supplementary figure 12. Supplementary ChIP-seq results and validation. (D) Tfii-i binding at *Mbp* genomic locus. The upper track shows Tfii-i binding in mOLs, with consensus peaks 75 and 62 highlighted in gray. The second, third, and fourth tracks show mOL-specific ChIP-seq data for H3K4me3, H3K27ac, and H3K36me3, respectively. The fifth and sixth tracks show Olig2 and Rad21 binding in mOLs^[13], and the seventh track shows embryonic Tfii-i binding data^[1]. The genome annotation track indicates known *Mbp* enhancers M3 and M5. The bottom track presents mOL-specific predicted cREs and their putative gene interactions, visualized as arcs. **(E)** Tfii-i binding at *Sox10* genomic locus. The upper track shows Tfii-i binding in mOLs, with consensus peaks 44 and 69 highlighted in gray. The second, third, and fourth tracks show mOL-specific ChIP-seq data for H3K4me3, H3K27ac, and H3K36me3, respectively. The fifth and sixth tracks show Olig2 and Rad21 binding in mOLs, and the seventh track shows embryonic Tfii-i binding data^[1]. The genome annotation track indicates known *Sox10* enhancers (U1-U5, D6-D7). The bottom track presents predicted mOL-specific cREs, with Tfii-i peaks 44 and 69 located within intronic regions of the *Micall1* gene. These peaks align with cREs predicted to interact with the *Sox10* promoter, visualized as arcs and highlighted in blue. **(F)** Venn diagram showing the overlap between consensus Tfii-i binding sites identified in mOLs and previously published embryonic Tfii-i ChIP-seq data^[1]. **(G)** Colocalization analysis of Tfii-i peaks in mOLs reveals substantial overlap with Srf and Olig2 binding, and limited overlap with embryonic Tfii-i, CTCF and Rad21 binding. *ns* - non-significant, $p < 0.05$.

We fully agree that comparing Olig2 binding with our mOL-derived Tfii-i data is highly relevant. As shown in the motif analysis (Figure 7C) and the new overlap analysis (Figure 7D), our genome-wide data support the potential for transcriptional cooperation between these factors in the OL lineage. Furthermore, the minimal overlap with Neurod1 (Figure 7D) supports the specificity of our Tfii-i binding data to mOLs. While these findings are preliminary and merit further investigation, we have addressed the reviewer's request by including the Olig2 ChIP-seq track, sourced from the same mOL-specific dataset as our histone mark data, in the revised genome browser panels (Supplementary Figure 12D-E). Notably, despite the broader co-occupancy observed genome-wide, we did not detect co-binding of Tfii-i and Olig2 at the *Mbp* and *Sox10* loci.

Figure 7. In mOLs, Tfii-i binds to RE of *Mbp* and *Sox10*. (C) Motif analysis within a 100 bp window centered on Tfii-i peaks in mOLs revealed enrichment of Olig2 and Sox10 binding motifs. (D) Colocalization analysis of Tfii-i consensus peaks in mOLs shows high overlap with Srf and Olig2 binding.

Some basic information is lacking such as whether the sequences at the GTF2I binding sites are conserved in other species (e.g. mouse/human) and whether GTF2I binding is similar to the patterns in ref. 113.

We thank the reviewer for this insightful comment. We agree that evaluating sequence conservation and motif enrichment provides important context to the interpretation of transcription factor binding sites and understanding the evolutionary roles of transcription factors.

To address this, **we assessed sequence conservation at two representative Tfii-i binding sites from our dataset: peak 75 within the *Mbp* promoter and peak 44 within a putative *Sox10* enhancer. Both regions display approximately 70% sequence identity between mouse and human, indicating moderate conservation.** While this level of conservation is somewhat lower than that reported for well-characterized *Sox10*^[12] and *Mbp*^[15, 16] enhancers (typically 85–95%), it nonetheless supports the possibility that these Tfii-i-bound regions may function as regulatory elements. These conservation data are now presented in Supplementary Figure 12B.

To evaluate binding sequence similarity, we performed an additional motif analysis focused on a 20 bp window centered around the peak summits. This higher-resolution analysis revealed enrichment for motifs such as BMAL1 and c-Myc, both of which are E-box-related sequences previously implicated as Tfii-i binding motifs in other biological systems^[1, 2]. These results are consistent with the known DNA-binding preferences of Tfii-i and further support the specificity of the identified peaks in our ChIP-seq data. The outcome of this refined motif analysis is provided in Supplementary Figure 12A.

Together, these additional analyses reinforce the biological relevance of Tfii-i binding sites in mOLs and support for their potential role as functional regulatory elements, and their consistency with previously published data. We have updated the manuscript to incorporate and discuss these findings accordingly. Revised text and Supplementary Figure 12A-B are provided below for the reviewer's convenience:

'Motif analysis of Tfii-i consensus peaks in mOLs within introns, exons, TTS, and promoters (i.e. across the gene body) revealed enrichment of Olig2 and Sox10 motifs, with the Olig2 motif being particularly prominent, alongside known Tfii-i binding motifs^[1, 2] (Figure 7C, Supplementary Figure 12A).'

'Among the high-confidence Tfii-i peaks, notable binding enrichment was observed at relatively evolutionarily conserved regions (Supplementary Figure 12B) within the loci of *Mbp* (Figure 7F) and *Sox10* (Figure 7G, Supplementary Figure 12C), both of which are critical for mOL function.'

A

Motif analysis (20 bp window)

B

Evolutionary conservation ratios at Tfii-i peaks in mOLs

Peak #	Peak position in mouse genome (mm10) and length	Conserved fragment length	Conservation to human (%)
75 (Mbp)	Chr18:82555802-82556185 383 bp	104 bp	70.2
44 (Sox10 putative cRE)	Chr15:79110627-79111064 437 bp	174 bp	68.4

Supplementary figure 12. Supplementary CHIP-seq results and validation. (A) Motif analysis of Tfii-i peaks within 20 bp window centered on peak summits reveals enrichment of E-box-related motifs, including BMAL1 and c-Myc. Tfii-i has previously been shown to bind to E-box motifs^[2, 17]. **(B)** Sequences at peaks 75 and 44 are relatively evolutionarily conserved between mouse and human genomes. A 104 bp fragment within peak 75 (*Mbp* locus) shares 70.2% identity with the hg19 genome, while a 174 bp fragment within peak 44 (*Sox10* putative cRE) shares 68.4% identity. Analyses and visualizations were performed using the ECR browser^[18].

Also, the colocalization of GTF2I with CTCF binding as reported in ref. 113 is not assessed here, which is important to assess consistency with previous findings.

We thank the reviewer for raising this important point. **As previously noted, direct comparisons between our Tfii-i ChIP-seq dataset, generated from postnatal, *in-vitro*-differentiated mOLs, and the embryonic whole-brain dataset from Kopp et al. (reference 113) are inherently limited due to substantial differences in developmental stage, cellular composition, and tissue context.** Despite these constraints, we acknowledge that evaluating colocalization with CTCF, a well-established chromatin architectural protein, provides valuable insight into the broader functional landscape of Tfii-i binding in mOLs.

To explore this, we performed colocalization analyses comparing our mOL-specific Tfii-i peaks with both the embryonic Tfii-i ChIP-seq peaks from Kopp et al.^[1] and available CTCF ChIP-seq datasets. The results of these analyses are now presented in Supplementary Figure 12C. As expected, we observed limited overlap between our mOL-derived Tfii-i peaks and those of embryonic Tfii-i or CTCF, consistent with the distinct developmental and cellular contexts in which the datasets were generated.

We interpret these findings as biologically meaningful and consistent with a context-dependent role for Tfii-i. Specifically, they support a model in which Tfii-i may function in association with architectural proteins such as CTCF during early development, while adopting a more direct transcriptional regulatory role in lineage-committed cells as mOLs. This is also evident from the low overlap scores of Tfii-i mOL-derived peaks with the architectural protein Rad21, as appears in Supplementary Figure 12G. We have included a discussion of these results and their implications in the revised manuscript. Relevant text adjustments and Supplementary Figure 12G are provided below for the reviewer's convenience:

'Previous Tfii-i ChIP-seq data from E13.5 neural tissue revealed strong colocalization with the architectural protein CTCF^[19], supporting a potential role in chromatin organization during early neural development. In contrast, our data show limited overlap with embryonic Tfii-i binding (Supplementary Figure 12F-G), as well as with CTCF binding and the architectural protein Rad21 (Supplementary Figure 12G). This discrepancy suggests a context-dependent role for Tfii-i that evolves during differentiation, potentially shifting from a chromatin architectural function during early development in embryonic neural progenitors to a more direct transcriptional regulatory role in mOLs.'

G

Overlap with OL-related TFs binding, embryonic Tfii-i binding, and architectural proteins binding

Supplementary figure 12. Supplementary CHIP-seq results and validation. (G) Colocalization analysis of Tfii-i peaks in mOLs reveals substantial overlap with Srf and Olig2 binding, and limited overlap with embryonic Tfii-i, CTCF and Rad21 binding. *ns* - non-significant, $p < 0.05$.

Finally, the addition of the 3CqPCR experiment in panels G and H should have multiple negative control primer sets in the same locus rather than a single negative control on a different chromosome. In general, analysis of 3C data requires multiple negative controls that are lacking here.

We thank the reviewer for raising this important point regarding the need for additional negative controls in the 3C-qPCR experiments. **In response to this valid concern, we have conducted additional experiments incorporating more rigorous internal controls to strengthen the reliability and interpretability of our results (presented in revised Figure 7H).**

Specifically, we generated new 3C libraries from whole cortex tissue of P30 control mice. These libraries were **thoroughly validated for restriction enzyme digestion efficiency, ligation efficiency, and overall quality in accordance with established 3C-qPCR protocols^[20]**. We then analyzed interaction frequencies at multiple genomic sites within the *Sox10* locus, focusing on regions corresponding to Tfii-i ChIP-seq peaks 44 and 69, both located near putative cREs. To ensure appropriate background correction, we included multiple negative control primer sets targeting adjacent non-peak regions within the same *Sox10* locus, in addition to the inter-chromosomal *Ercc3* site commonly used as a normalization control in 3C-qPCR assays^[20]. As an internal positive control, we observed high interaction frequencies at D7, a well-characterized *Sox10* enhancer^[12], supporting the reliability of our 3C-qPCR data. This approach allowed us to assess local specificity of interactions and control for non-specific ligation events.

In parallel, **we also utilized residual material from the original mOL-derived 3C libraries, prepared from the same biological samples used in our ChIP-seq experiment, to replicate key findings. Consistent with the cortex-derived data, these mOL-derived 3C libraries demonstrated elevated interaction frequencies between the *Sox10* promoter and the regions corresponding to peaks 44 and 69, relative to nearby non-peak controls.**

Together, these results strengthen our conclusion that the Tfii-i-bound regions at peaks 44 and 69 are spatially proximate to the *Sox10* promoter and may function as regulatory elements. While we acknowledge that further functional characterization would be required to definitively establish enhancer activity, the improved 3C-qPCR data provide additional support for a regulatory relationship between Tfii-i and *Sox10* in mOLs.

The revised 3C-qPCR figure (Figure 7H) and its related main-text adjustments are provided below for the reviewer's convenience:

'To experimentally assess these promoter–cREs interactions, we performed a chromatin conformation capture assay followed by qPCR (3C-qPCR) on cells derived from the cortex of P30 control mice and from an mOL-enriched primary culture. Interaction frequencies between the *Sox10* promoter and candidate cREs, as well as other loci within the *Sox10* region, were quantified and normalized to the interaction frequency at the *Erc3* locus, a commonly used reference in 3C-qPCR assays^[20] (Figure 7H). Consistent with classical 3C-qPCR profiles, we observed high interaction frequencies at proximal genomic loci near the *Sox10* promoter (e.g., the 5 Kb region), reflecting efficient proximal ligation. These frequencies declined at more distal sites (e.g., 35 Kb), but showed a local increase at the positions of the putative cREs (peaks 69 and 44), before declining again at more distant regions (65 Kb and 85 Kb). This interaction pattern was observed in data derived from both whole cortex and mOL-enriched primary culture samples. Notably, we also detected elevated interaction frequency at D7, a previously characterized downstream enhancer of *Sox10*^[12]. These findings demonstrate that the *Sox10* promoter interacts with the specific predicted cREs. While this does not constitute conclusive evidence of enhancer activity, it provides a candidate locus and suggests that Tfii-i binding may, in some cases, align with active regulatory elements. Enhancer-promoter interactions were previously shown to play a crucial role in OL transcriptional program^[21, 22]. As such, this result supports the notion that Tfii-i may contribute to *Sox10* transcriptional regulation.'

H

Figure 7. In mOLs, Tfi-i binds to RE of *Mbp* and *Sox10*. (H) 3C-qPCR analysis (n=3/4 for whole cortex and n=4 pooled into a single sample for mOLs, control only). Elevated interaction frequencies were observed between *Sox10* promoter and the regions corresponding to consensus peaks 44 and 69 (highlighted in gray), compared to adjacent regions within the same locus. Interaction frequencies were normalized to those at the *Ercc3* region and further normalized to the 35 Kb data point for each dataset.

Based on these considerations, I would recommend deleting Figure 7 in its entirety.

We appreciate the reviewer's critical evaluation of Figure 7 and understand that the original version may have lacked sufficient clarity and supporting evidence. However, **we believe that the substantial additions and refinements made in the current revision, including improved genome browser visualizations, enhanced motif overlap, and conservation analysis, incorporation of mOL-specific datasets, ChIP-qPCR validation, and expanded 3C-qPCR experiments with appropriate internal controls, collectively reinforce the technical rigor and biological relevance of the data presented in the revised figure.**

These revisions address the primary concerns raised and provide multiple layers of evidence supporting our conclusions regarding Tfii-i's role in mOL transcriptional regulation. **Given the novelty of Tfii-i function in mOLs, and the broader interest in transcription factor cooperation in lineage-specific contexts, we believe that the data in the revised Figure 7 offer valuable insight to both Tfii-i and mOL regulatory research communities.**

For the reviewer's convenience, we have attached below the revised Figure 7 and Supplementary Figure 12 in their entirety:

A Tfii-i consensus peaks in mOLs

B Genomic distribution of Tfii-i consensus peaks in mOLs

C Motif analysis – Tfii-i peaks in mOLs (gene body)

D Colocalization of Tfii-i peaks in mOLs with other TFs

E Tfii-i consensus peaks overlap to mOL-specific PTMH

F Tfii-i binding in mOLs at *Mbp* locus

G Tfii-i binding in mOLs at *Sox10* locus

H Experimental cREs interactions at *Sox10* genomic loci

Figure 7. In mOLs, Tfi-i binds to RE of *Mbp* and *Sox10*. (A) Heatmap showing genome-wide Tfi-i binding intensity from ChIP-seq data in mOLs derived from control and *Gtf2i*-KO mice. (B) Genomic distribution of Tfi-i consensus peaks in control mOLs. (C) Motif analysis within a 100 bp window centered on Tfi-i peaks in mOLs reveals enrichment of Olig2 and Sox10 binding motifs. (D) Colocalization analysis of Tfi-i consensus peaks in mOLs shows high overlap with Srf and Olig2 binding. (E) Overlap analysis of Tfi-i peaks with mOL-specific PTMH reveals predominant localization within genomic regions enriched for H3K36me3, a mark associated with active transcription. (F) Tfi-i binding at the *Mbp* genomic locus. The upper track displays Tfi-i binding in mOLs, with consensus peaks highlighted in gray (peaks 75 and 62), TSS is marked in green. The second, third, and fourth tracks show mOL-specific H3K4me3, H3K27ac, and H3K36me3 binding data, respectively. The genome annotation track indicates known *Mbp* enhancers M3 and M5. The bottom track presents mOL-specific predicted cREs and their putative interactions with target genes, visualized as arcs. (G) Tfi-i binding at *Sox10* genomic locus. The upper track displays Tfi-i binding in mOLs, with consensus peaks highlighted in gray (peaks 44 and 69), the TSS is marked in green. The second, third, and fourth tracks show mOL-specific H3K4me3, H3K27ac, and H3K36me3 binding data, respectively. The genome track indicates known *Sox10* enhancers (U1-U5, D6-D7). The bottom track presents mOL-specific predicted cREs. Tfi-i peaks 44 and 69, located within intronic regions of the *Micall1* gene, align with these cREs and are predicted to physically interact with the *Sox10* promoter (highlighted in blue). (H) 3C-qPCR analysis (n=3/4 for whole cortex and n=4 pooled into a single sample for mOLs, control only). Elevated interaction frequencies were observed between *Sox10* promoter and the regions corresponding to consensus peaks 44 and 69 (highlighted in gray), compared to adjacent regions within the same locus. Interaction frequencies were normalized to those at the *Ercc3* region and further normalized to the 35 Kb data point for each dataset.

A

B

Evolutionary conservation ratios at Tfii-i peaks in mOLs

Peak #	Peak position in mouse genome (mm10) and length	Conserved fragment length	Conservation to human (%)
75 (Mbp)	Chr18:82555802-82556185 383 bp	104 bp	70.2
44 (Sox10 putative cRE)	Chr15:79110627-79111064 437 bp	174 bp	68.4

C

D

E

F

Overlap with embryonic Tfii-i binding

G

Overlap with OL-related TFs binding, embryonic Tfii-i binding, and architectural proteins binding

Supplementary figure 12. Supplementary ChIP-seq results and validation. (A) Motif analysis of Tfii-i peaks within 20 bp window centered on peak summits reveals enrichment of E-box-related motifs, including BMAL1 and c-Myc. Tfii-i has previously been shown to bind to E-box motifs^[2, 17]. (B) Sequences at peaks 75 and 44 are relatively evolutionarily conserved between mouse and human genomes. A 104 bp fragment within peak 75 (*Mbp* locus) shares 70.2% identity with the hg19 genome, while a 174 bp fragment within peak 44 (*Sox10* putative cRE) shares 68.4% identity. Analyses and visualizations were performed using the ECR browser^[18]. (C) ChIP-qPCR validation of ChIP-seq results (control $n=5$, *Gtf2i*-KO $n=4$). Primers targeting consensus peaks 75 (*Mbp* locus), 44, and 69 (*Sox10* locus) confirmed reduced Tfii-i binding in *Gtf2i*-KO group compared to control group at peak 75 (Student's t-test, $P = 0.035$), peak 44 (Student's t-test, $P = 0.013$), and peak 69 (Student's t-test, $P = 0.023$). No significant change was observed at a negative control region (Student's t-test, $P = 0.511$). (D) Tfii-i binding at *Mbp* genomic locus. The upper track shows Tfii-i binding in mOLs, with consensus peaks 75 and 62 highlighted in gray. The second, third, and fourth tracks show mOL-specific ChIP-seq data for H3K4me3, H3K27ac, and H3K36me3, respectively. The fifth and sixth tracks show Olig2 and Rad21 binding in mOLs^[13], and the seventh track shows embryonic Tfii-i binding data^[1]. The genome annotation track indicates known *Mbp* enhancers M3 and M5. The bottom track presents mOL-specific predicted cREs and their putative gene interactions, visualized as arcs. (E) Tfii-i binding at *Sox10* genomic locus. The upper track shows Tfii-i binding in mOLs, with consensus peaks 44 and 69 highlighted in gray. The second, third, and fourth tracks show mOL-specific ChIP-seq data for H3K4me3, H3K27ac, and H3K36me3, respectively. The fifth and sixth tracks show Olig2 and Rad21 binding in mOLs, and the seventh track shows embryonic Tfii-i binding data^[1]. The genome annotation track indicates known *Sox10* enhancers (U1-U5, D6-D7). The bottom track presents predicted mOL-specific cREs, with Tfii-i peaks 44 and 69 located within intronic regions of the *Micall1* gene. These peaks align with cREs predicted to interact with the *Sox10* promoter, visualized as arcs and highlighted in blue. (F) Venn diagram showing the overlap between consensus Tfii-i binding sites identified in mOLs and previously published embryonic Tfii-i ChIP-seq data^[1]. (G) Colocalization analysis of Tfii-i peaks in mOLs reveals substantial overlap with Srf and Olig2 binding, and limited overlap with embryonic Tfii-i, CTCF and Rad21 binding. *ns* - non-significant, $p < 0.05$.

Since the genomic analysis is not too convincing, there should be a more explicit consideration of models in which the SOX10 overexpression itself may account for the entire phenotype (or also the depletion of GPR37 as a negative regulator of myelination). More explicit statements should be provided as to how this phenotype compares with the phenotypes of SOX10 overexpression or GPR37 loss-of-function to assess whether these changes could largely account for the phenotype shown here.

We thank the reviewer for this valuable suggestion. **In response, we have added two paragraphs to the Discussion section, explicitly comparing the myelination phenotype observed in our *Gtf2i*-KO model with reported effects of *Sox10* overexpression and *Gpr37* loss-of-function.** While both factors may contribute to the phenotype, we emphasize key distinctions in timing, cellular context, and anatomical distribution, supporting the conclusion that *Gtf2i* loss in myelinating glia leads to effects beyond either mechanism alone. The revised text is provided below:

'Myelin, OLs, and axon health are tightly regulated by one another^[23-28]. *Sox10* expression levels have been shown to decrease following exposure to cuprizone^[29], a copper chelator known to induce demyelination^[30, 31]. Although postnatal *Sox10* overexpression has not been extensively studied *in-vivo*, current evidence suggests it can enhance remyelination and support myelin maintenance. For instance, overexpression of *Sox10* in the hippocampus following de-myelination by cuprizone, improved behavioral deficits, myelin ultrastructural properties, and normalized myelin sheath-related protein expression levels^[32]. Furthermore, in another study of a cuprizone demyelination model, upon the introduction of *Sox10* through exosomes, OPC differentiation to OLs was promoted and an increase in *Mbp* protein level was observed^[29]. Additionally, in the embryonic chick spinal cord, overexpression of *Sox10* was shown to promote the expression of myelin-related genes^[5]. In SCs derived from the rat sciatic nerve, *Sox10* overexpression similarly enhanced myelination^[33]. Furthermore, in the oligodendroglial cell line, Oli-neu, *Sox10* overexpression increased the expression of *Mbp* and other OL-related genes^[34]. Collectively, these findings suggest that elevated *Sox10* expression may contribute to remyelination and support myelin maintenance. They also support the notion that increased *Sox10* levels could underline, at least in part, the myelination phenotype observed following *Gtf2i* deletion in myelinating glial cells.

Gpr37 has been identified as a negative regulator of OL differentiation and myelination^[35, 36]. Its expression begins at the late OPC stage^[36], which partially aligns with the timing of *Gtf2i* deletion using the *CnpCre* line^[37]. Although reduced *Gpr37* expression in the CNS myelin of *Gtf2i*-KO mice may contribute to the enhanced myelination phenotype, this change alone is unlikely to fully account for the observed effects. Notably, *Gpr37* depletion has been shown to result in precocious myelination through increased differentiation of pre-myelinating OLs into mature, myelin-producing OLs^[36]. However, in our *Gtf2i*-KO model, we did not observe accelerated OL

differentiation across multiple developmental time points. Moreover, Gpr37 expression is minimal in the SN^[36], suggesting it cannot explain the hypermyelination observed in the PNS of *Gtf2i*-KO mice. Therefore, while both increased *Sox10* and decreased Gpr37 expression may contribute to the myelination phenotype, the overall effects likely reflect a combination of these transcriptional and molecular changes, alongside direct regulatory roles of Tfi-i in gene expression programs within myelinating glial cells.'

Another notable omission is the well-established role of GTF2I in DNA repair, and indeed sites of DNA repair can be enriched in H3K36me3, but no mention of that is provided in the manuscript.

We thank the reviewer for this important observation. **We have now addressed this point in the Discussion section by acknowledging the well-established role of GTF2I in DNA repair and by highlighting the potential relevance of its colocalization with H3K36me3 in mOLs.** The revised paragraph is provided below:

'GTF2I has been implicated in the cellular response to DNA damage and in DNA repair pathways^[38-40]. In parallel, H3K36me3, traditionally associated with transcriptionally active gene bodies^[41, 42], has also been shown to accumulate at sites of DNA damage and facilitate the recruitment of DNA repair machinery^[43, 44]. Our findings indicate that Tfi-i binding in mOLs substantially colocalizes with regions marked by H3K36me3. While DNA repair was not the primary focus of this study, this observation raises the possibility that Tfi-i may also contribute to DNA repair processes in mOLs'.

Other comments:

- p. 16: is the "myelin fraction" purified myelin, or does it have intact cells?

We thank the reviewer for raising this issue. These experiments were performed on the purified myelin fraction, i.e. – no intact cells. We've now highlighted this in the revised manuscript to avoid confusion.

- p. 17: please correct: "1.375+/-0.0"

We thank the reviewer for pointing out this technical mistake, which we have now rectified.

- Figure 5: Figure legend should state how protein and mRNA data are normalized.

We thank the reviewer for this comment. We've now added more clarity in this regard in the figure legend.

'Protein levels were normalized to β -tubulin IV. mRNA levels were normalized to *Gapdh*.'

- figure 6: there are several references to "one-sample t-test" here and elsewhere, should this be 1-sided t-test?

We thank the reviewer for this observation. In Figure 6, we indeed used a one-sample t-test, not a one-sided t-test. These experiments were performed in primary, differentiated mOLs cultures, in which each imaging slide contained a matched pair of conditions: one control (infected with mCherry-expressing AAV) coverslip and one iCre-mCherry-infected coverslip to induce the deletion of *Gtf2i*. To account for potential technical variability across slides (e.g., staining intensity, different technical conditions), we normalized the fluorescence intensity values for each iCre-infected sample to the corresponding control on the same slide. As a result, all control samples are normalized to a value of 1, and each iCre value represents a fold change relative to its matched control.

Given this setup, we performed a one-sample t-test comparing the normalized values from the iCre group against the theoretical mean of 1. This test is appropriate when assessing whether the experimental condition significantly deviates from a defined reference value, which is, in this case, the baseline level set by the control group. We have clarified this point in the methods section of the revised manuscript to avoid confusion.

It is noted that the manuscript employed a different antibody from a previous publication (ref. 113) with much better results of GTF2i binding.

We thank the reviewer for their attention to details in reviewing our manuscript.

Also, since the read mapping from mouse cells was done in the mouse genome, why was the LiftOver tool used in the ChIP-seq data analysis?

The read mapping of the ChIP-seq experiment was performed on the mm39 genome while the PTMH, Olig2, Rad21, and embryonic Tfii-i data were mapped to the mm10 genome. As such, we utilized the LiftOver tool to visualize all datasets on the same genome (mm10).

Primer sets for 3C-qPCR are not provided.

We apologize for the inconvenience. We have provided the 3C-qPCR primers sets in Supplementary Table 3 which was not made available due to a technical error. The updated primer sets are now readily available in supplementary table 3.

References

1. Kopp, N.D., et al., *Functions of Gtf2i and Gtf2ird1 in the developing brain: transcription, DNA binding and long-term behavioral consequences*. Human Molecular Genetics, 2020. **29**(9): p. 1498-1519.
2. Makeyev, A.V., et al., *Diversity and Complexity in Chromatin Recognition by TFII-I Transcription Factors in Pluripotent Embryonic Stem Cells and Embryonic Tissues*. PLOS ONE, 2012. **7**(9): p. e44443.
3. Iram, T., et al., *SRF transcriptionally regulates the oligodendrocyte cytoskeleton during CNS myelination*. Proceedings of the National Academy of Sciences, 2024. **121**(12): p. e2307250121.
4. Sock, E. and M. Wegner, *Using the lineage determinants Olig2 and Sox10 to explore transcriptional regulation of oligodendrocyte development*. Developmental Neurobiology, 2021. **81**(7): p. 892-901.
5. Liu, Z., et al., *Induction of oligodendrocyte differentiation by Olig2 and Sox10: Evidence for reciprocal interactions and dosage-dependent mechanisms*. Developmental Biology, 2007. **302**(2): p. 683-693.
6. Chen, X., et al., *Myelin Deficits Caused by Olig2 Deficiency Lead to Cognitive Dysfunction and Increase Vulnerability to Social Withdrawal in Adult Mice*. Neuroscience Bulletin, 2020. **36**(4): p. 419-426.
7. Wedel, M., et al., *Transcription factor Tcf4 is the preferred heterodimerization partner for Olig2 in oligodendrocytes and required for differentiation*. Nucleic Acids Research, 2020. **48**(9): p. 4839-4857.
8. Wegener, A., et al., *Gain of Olig2 function in oligodendrocyte progenitors promotes remyelination*. Brain, 2014. **138**(1): p. 120-135.
9. Küspert, M., et al., *Olig2 regulates Sox10 expression in oligodendrocyte precursors through an evolutionary conserved distal enhancer*. Nucleic Acids Research, 2010. **39**(4): p. 1280-1293.
10. Li, H. and W.D. Richardson, *The evolution of Olig genes and their roles in myelination*. Neuron Glia Biology, 2009. **4**(2): p. 129-135.
11. Wegner, M., *A Matter of Identity: Transcriptional Control in Oligodendrocytes*. Journal of Molecular Neuroscience, 2008. **35**(1): p. 3-12.
12. Werner, T., et al., *Multiple conserved regulatory elements with overlapping functions determine Sox10 expression in mouse embryogenesis*. Nucleic Acids Research, 2007. **35**(19): p. 6526-6538.
13. Bartosovic, M., M. Kabbe, and G. Castelo-Branco, *Single-cell CUT&Tag profiles histone modifications and transcription factors in complex tissues*. Nature Biotechnology, 2021. **39**(7): p. 825-835.
14. Pliner, H.A., et al., *Cicero Predicts cis-Regulatory DNA Interactions from Single-Cell Chromatin Accessibility Data*. Mol Cell, 2018. **71**(5): p. 858-871.e8.
15. Dionne, N., et al., *Functional organization of an Mbp enhancer exposes striking transcriptional regulatory diversity within myelinating glia*. Glia, 2016. **64**(1): p. 175-194.
16. Dib, S., et al., *Regulatory modules function in a non-autonomous manner to control transcription of the mbp gene*. Nucleic Acids Research, 2010. **39**(7): p. 2548-2558.

17. Roy, A.L., et al., *Cooperative interaction of an initiator-binding transcription initiation factor and the helix–loop–helix activator USF*. *Nature*, 1991. **354**(6350): p. 245-248.
18. Ovcharenko, I., et al., *ECR Browser: a tool for visualizing and accessing data from comparisons of multiple vertebrate genomes*. *Nucleic Acids Research*, 2004. **32**(suppl_2): p. W280-W286.
19. Ong, C.-T. and V.G. Corces, *CTCF: an architectural protein bridging genome topology and function*. *Nature Reviews Genetics*, 2014. **15**(4): p. 234-246.
20. Hagège, H., et al., *Quantitative analysis of chromosome conformation capture assays (3C-qPCR)*. *Nature Protocols*, 2007. **2**(7): p. 1722-1733.
21. Kim, D., et al., *A principled strategy for mapping enhancers to genes*. *Scientific Reports*, 2019. **9**(1): p. 11043.
22. Cheng, N., et al., *STAG2 promotes the myelination transcriptional program in oligodendrocytes*. *eLife*, 2022. **11**: p. e77848.
23. Nave, K.-A. and J.L. Salzer, *Axonal regulation of myelination by neuregulin 1*. *Current Opinion in Neurobiology*, 2006. **16**(5): p. 492-500.
24. Fruttiger, M., et al., *Crucial Role for the Myelin-associated Glycoprotein in the Maintenance of Axon-Myelin Integrity*. *European Journal of Neuroscience*, 1995. **7**(3): p. 511-515.
25. Griffiths, I., et al., *Axonal Swellings and Degeneration in Mice Lacking the Major Proteolipid of Myelin*. *Science*, 1998. **280**(5369): p. 1610-1613.
26. Nave, K.A., *Myelination and support of axonal integrity by glia*. *Nature*, 2010. **468**(7321): p. 244-52.
27. Nave, K.A., *Myelination and the trophic support of long axons*. *Nat Rev Neurosci*, 2010. **11**(4): p. 275-83.
28. Traka, M., et al., *Oligodendrocyte death results in immune-mediated CNS demyelination*. *Nat Neurosci*, 2016. **19**(1): p. 65-74.
29. He, J., et al., *Exosome-specific loading Sox10 for the treatment of Cuprizone-induced demyelinating model*. *Biomedicine & Pharmacotherapy*, 2024. **171**: p. 116128.
30. Franco-Pons, N., et al., *Behavioral deficits in the cuprizone-induced murine model of demyelination/remyelination*. *Toxicology Letters*, 2007. **169**(3): p. 205-213.
31. Matsushima, G.K. and P. Morell, *The Neurotoxicant, Cuprizone, as a Model to Study Demyelination and Remyelination in the Central Nervous System*. *Brain Pathology*, 2001. **11**(1): p. 107-116.
32. Shao, Y., et al., *Effect of Sox10 on remyelination of the hippocampus in cuprizone-induced demyelinated mice*. *Brain and Behavior*, 2020. **10**(6): p. e01623.
33. Fujiwara, S., et al., *SOX10 Transactivates S100B to Suppress Schwann Cell Proliferation and to Promote Myelination*. *PLOS ONE*, 2014. **9**(12): p. e115400.
34. He, D., et al., *Chd7 cooperates with Sox10 and regulates the onset of CNS myelination and remyelination*. *Nature Neuroscience*, 2016. **19**(5): p. 678-689.
35. Qian, Z., et al., *Osteocalcin attenuates oligodendrocyte differentiation and myelination via GPR37 signaling in the mouse brain*. *Science Advances*, 2021. **7**(43): p. eabi5811.

36. Yang, H.-J., et al., *G protein-coupled receptor 37 is a negative regulator of oligodendrocyte differentiation and myelination*. Nature Communications, 2016. **7**(1): p. 10884.
37. Lappe-Siefke, C., et al., *Disruption of Cnp1 uncouples oligodendroglial functions in axonal support and myelination*. Nat Genet, 2003. **33**(3): p. 366-74.
38. Chen, L., et al., *Spatiotemporal 7q11.23 protein network analysis implicates the role of DNA repair pathway during human brain development*. Scientific Reports, 2021. **11**(1): p. 8246.
39. Roy, A.L., *Role of the multifunctional transcription factor TFII-I in DNA damage repair*. DNA Repair, 2021. **106**: p. 103175.
40. Tanikawa, M., et al., *Role of multifunctional transcription factor TFII-I and putative tumour suppressor DBC1 in cell cycle and DNA double strand damage repair*. British Journal of Cancer, 2013. **109**(12): p. 3042-3048.
41. Barski, A., et al., *High-Resolution Profiling of Histone Methylations in the Human Genome*. Cell, 2007. **129**(4): p. 823-837.
42. Bannister, A.J., et al., *Spatial Distribution of Di- and Tri-methyl Lysine 36 of Histone H3 at Active Genes **. Journal of Biological Chemistry, 2005. **280**(18): p. 17732-17736.
43. Pfister, Sophia X., et al., *SETD2-Dependent Histone H3K36 Trimethylation Is Required for Homologous Recombination Repair and Genome Stability*. Cell Reports, 2014. **7**(6): p. 2006-2018.
44. Li, F., et al., *The Histone Mark H3K36me3 Regulates Human DNA Mismatch Repair through Its Interaction with MutS*. Cell, 2013. **153**(3): p. 590-600.

POINT-BY-POINT RESPONSE

The revised manuscript has addressed some of the critiques raised in the first round regarding the rigor of the genomic analysis of Gtf2i binding. In particular, the 3C analysis in Figure 7H has much more convincing controls in this round. However, I remain concerned regarding the binding analysis by ChIP-seq in the remainder of Figure 7.

We thank the reviewer for their additional evaluation of our manuscript and for acknowledging the improvements made in the revised manuscript, particularly with respect to the 3C analysis in Figure 7H. We appreciate their recognition of our efforts to enhance the rigor and clarity of our genomic analyses.

Below, we address the remaining concerns regarding the ChIP-seq binding analysis presented in Figure 7.

The revised manuscript has done some further comparisons as requested with previous data sets of SOX10 and OLIG2 binding sites, along with a previous analysis of Gtf2i from another publication. However, there remain some internal contradiction since there is an apparent colocalization of Gtf2i with OLIG2 and SOX10, but this is not observed in the two loci shown: *Mbp* and *Sox10*. If there are Gtf2i regulated genes where such colocalization is evident, it should be shown.

We thank the reviewer for highlighting this contradiction and for their thoughtful suggestion. We agree with the reviewer that colocalization of Tfi-i, Olig2, and Sox10 is not observed at the *Mbp* and *Sox10* loci shown in Figure 7. To address this, we have added Supplementary Figure 12D–E (shown below at the end of this letter, for convenience), which highlights two representative genomic loci where Tfi-i, Olig2, and Sox10 binding overlap specifically in mOLs.

Moreover, the colocalization of Gtf2i with H3K36 methylation may simply be due to the broad distribution of H3K36 methylation within gene bodies, and there are only discrete sites of Gtf2i binding within these broad regions that constitute the overlap. I would recommend that the statistical significance of this overlap should be calculated. The significance of the overlap with H3K27ac should also be calculated, since this modification is typically more restricts, and is associated with enhancer elements.

We thank the reviewer for raising this important point. We agree that a statistical assessment of Tfii-i colocalization with specific histone modifications in mOLs would strengthen the interpretation of our findings. To assess the enrichment of Tfii-i binding within specific histone modification landscapes, we performed an overlap analysis between our Tfii-i consensus peaks and genomic regions marked by histone modifications (H3K27ac, H3K36me3, H3K4me3, H3K27me3) from mOLs. Using bedtools intersect, we calculated the number of overlaps. **To evaluate statistical significance, we applied Fisher's exact test, incorporating both the number and size of histone-marked regions to estimate expected overlaps under a random distribution model.** This analysis revealed significant enrichment of Tfii-i binding within H3K36me3-marked regions, as shown in the revised Supplementary Figure 12G shown below. These findings support the interpretation that Tfii-i preferentially associates with transcriptionally active chromatin in mOLs.

Finally, the response provided for the apparent wide disparity of Gtf2i binding patterns among biological replicates in the revised Figure 7A is not adequate. It is still apparent that many very strong peaks are not commonly shared across biological replicates, which is unexpected. At the very least, probably it should be stated how many peaks are called for each replicate, and how many are shared across all 5, since it is stated that the analysis requires consensus peaks across biological replicates. Typically, one would expect strong peaks to be shared across replicates, and the lack of replication of strong peaks is evident in the revised Figure 7A. To this point, I would recommend showing peak calling for all 5 replicates for a single locus (e.g. Sox10) (or perhaps in a single chromosome) in a supplemental figure to provide a better illustration of inter-replicate variability, which is a key issue in interpreting the data.

We thank the reviewer for emphasizing this important point. We agree that reporting key metrics such as the number of peaks called per replicate, the number of shared peaks across replicates, and a visual illustration of inter-replicate variability would enhance the interpretability of the ChIP-seq data presented in Figure 7.

To address this, we have added new supplementary panels (Supplementary Figure 12A–C, shown below) that provide: (A) the number of peaks called in each of the five individual biological replicates; (B) the number and proportion of overlapping peaks across different combinations of replicates; and (C) browser tracks from a representative genomic region showing peak calls from all five replicates. This region includes the *Sox10* locus and demonstrates variability in peak presence across replicates. We believe these additions provide greater transparency regarding replicate consistency and allow for more informed interpretation of the consensus peak analysis.

For the reviewer's convenience, the entire revised Supplementary Figure 12 is presented below:

A

Peak calling per replicate

Sample	Control 1	Control 3	Control 4	Control 5	Control 6
# of peaks	247621	198351	134244	250514	188152

B

Tfii-I peaks shared across replicates

Number of peaks	Shared across
26	5 samples
485	4 samples
9959	3 samples

C

Inter-replicate variability of Tfii-i binding in mOLs at the Sox10 genomic locus

D

Overlap of Tfii-I, Olig2, and Sox10 binding in mOLs

E

Overlap of Tfii-I, Olig2, and Sox10 binding in mOLs

F

Motif analysis (20 bp window)

G

Tfii-i binding overlap with PTMH in mOLs

Histone	Odds ratio	P-value
H3K36me3	2.054213	0.028819
H3K27ac	3.083854	0.143524
H3K4me3	1.510268	0.488894
H3K27me3	0	1

H

Evolutionary conservation ratios at Tfii-i peaks in mOLs

Peak #	Peak position in mouse genome (mm10) and length	Conserved fragment length	Conservation to human (%)
75 (Mbp)	Chr18:82555802-82556185 383 bp	104 bp	70.2
44 (Sox10 putative cRE)	Chr15:79110627-79111064 437 bp	174 bp	68.4

I

J

K

L

Overlap with embryonic Tfii-i binding

M

Overlap with OL-related TFs binding, embryonic Tfii-i binding, and architectural proteins bindings

Supplementary Figure 12. Supplementary ChIP-seq results and validation. (A) Table summarizing the number of peaks called in each of the five individual biological replicates. (B) Table summarizing the number of overlapping peaks across different combinations of replicates. (C) IGV track showing the inter-replicate variability of peak calling across all five biological replicates at the *Sox10* genomic locus. The upper track shows Tffii-i binding in mOLs (bed format), with consensus peaks 44 and 69 highlighted in gray. Other tracks peak calling per each replicate in the presented genomic locus. (D-E) Tffii-i, Olig2, and Sox10 binding overlap in mOLs. The upper track shows Tffii-i binding in mOLs with consensus peaks highlighted in gray. The second track shows Olig2 binding in mOLs^[4], while the third track shows Sox10 binding in mOLs (bed format, from rat origin). (D) *Tfb2m* and *Cnst* genomic locus, (E) *Rbm39* locus. (F) Motif analysis of Tffii-i peaks within 20 bp window centered on peak summits reveals enrichment of E-box-related motifs, including BMAL1 and c-Myc. Tffii-i has previously been shown to bind to E-box motifs^[1,2]. (G) Tffii-i colocalization analysis with mOLs-specific PTMH reveals significant enrichment of Tffii-i binding within H3K36me3-marked regions in mOLs (two-sided Fisher's exact test). (H) Sequences at peaks 75 and 44 are relatively evolutionarily conserved between mouse and human genomes. A 104 bp fragment within peak 75 (*Mbp* locus) shares 70.2% identity with the hg19 genome, while a 174 bp fragment within peak 44 (*Sox10* putative cRE) shares 68.4% identity. Analyses and visualizations were performed using the ECR browser^[3]. (I) ChIP-qPCR validation of ChIP-seq results (control $n=5$, *Gtf2i*-KO $n=4$). Primers targeting consensus peaks 75 (*Mbp* locus), 44, and 69 (*Sox10* locus) confirmed reduced Tffii-i binding in *Gtf2i*-KO group compared to control group at peak 75 (two-sided t-test, $P=0.035$), peak 44 (two-sided t-test, $P=0.013$), and peak 69 (two-sided t-test, $P=0.023$). No significant change was observed at a negative control region (two-sided t-test, $P=0.511$). (J) Tffii-i binding at *Mbp* genomic locus. The upper track shows Tffii-i binding in mOLs, with consensus peaks 75 and 62 highlighted in gray. The second, third, and fourth tracks show mOL-specific ChIP-seq data for H3K4me3, H3K27ac, and H3K36me3, respectively. The fifth and sixth tracks show Olig2 and Rad21 binding in mOLs^[4], and the seventh track shows embryonic Tffii-i binding data^[5]. The genome annotation track indicates known *Mbp* enhancers M3 and M5. The bottom track presents mOL-specific predicted cREs and their putative gene interactions, visualized as arcs. (K) Tffii-i binding at *Sox10* genomic locus. The upper track shows Tffii-i binding in mOLs, with consensus peaks 44 and 69 highlighted in gray. The second, third, and fourth tracks show mOL-specific ChIP-seq data for H3K4me3, H3K27ac, and H3K36me3, respectively. The fifth and sixth tracks show Olig2 and Rad21 binding in mOLs, and the seventh track shows embryonic Tffii-i binding data^[5]. The genome annotation track indicates known *Sox10* enhancers. The bottom track presents predicted mOL-specific cREs, with Tffii-i peaks 44 and 69 located within intronic regions of the *Mical1* gene. These peaks align with cREs predicted to interact with the *Sox10* promoter, visualized as arcs and highlighted in blue. (L) Venn diagram showing the overlap between consensus Tffii-i binding sites identified in mOLs and previously published embryonic Tffii-i ChIP-seq data^[5]. (M) Colocalization analysis of Tffii-i peaks in mOLs reveals substantial overlap with Srf and Olig2 binding, and limited overlap with embryonic Tffii-i, CTCF and Rad21 binding. Data are presented as mean values \pm SEM. *ns* - non-significant, $p < 0.05$. Source data are provided as a Source Data file.

While I remain concerned regarding the quality and usefulness of the ChIP-seq data, the added analyses and modified figures do now allow readers to assess how these data conform (or do not conform) to previous studies of GTF2i, SOX10 and OLI2 binding, along with the known regulatory elements of Sox10 and Mbp. The remainder of the manuscript in figures 1-6 provides a fairly convincing analysis, and appropriate changes were made to the Discussion.

We thank the reviewer for their constructive feedback, which has helped us improve the clarity and rigor of our manuscript, with focus on the genomic analysis. We are encouraged by the reviewer's overall positive assessment of the revised manuscript and appreciate their acknowledgment of the improvements made to the genomic analysis and the Discussion, as well as their supportive comments on the overall strength of the manuscript.

There seems to be a formatting issue with Supplemental Figure 12.

We apologize for the formatting issue and thank the reviewer for their keen attention to details. This has been rectified in the revised manuscript.

References

1. Makeyev, A.V., et al., *Diversity and Complexity in Chromatin Recognition by TFII-I Transcription Factors in Pluripotent Embryonic Stem Cells and Embryonic Tissues*. PLOS ONE, 2012. **7**(9): p. e44443.
2. Roy, A.L., et al., *Cooperative interaction of an initiator-binding transcription initiation factor and the helix–loop–helix activator USF*. Nature, 1991. **354**(6350): p. 245-248.
3. Ovcharenko, I., et al., *ECR Browser: a tool for visualizing and accessing data from comparisons of multiple vertebrate genomes*. Nucleic Acids Research, 2004. **32**(suppl_2): p. W280-W286.
4. Bartosovic, M., M. Kabbe, and G. Castelo-Branco, *Single-cell CUT&Tag profiles histone modifications and transcription factors in complex tissues*. Nature Biotechnology, 2021. **39**(7): p. 825-835.
5. Kopp, N.D., et al., *Functions of Gtf2i and Gtf2ird1 in the developing brain: transcription, DNA binding and long-term behavioral consequences*. Human Molecular Genetics, 2020. **29**(9): p. 1498-1519.